# A Survey and Datasheet Repository of Publicly Available US Criminal Justice Datasets

**Miri Zilka**[*]
University of Cambridge
Cambridge, UK
mz477@cam.ac.uk

**Bradley Butcher**[*]
University of Sussex
Brighton, UK
b.butcher@sussex.ac.uk

**Adrian Weller**
University of Cambridge;
The Alan Turing Institute
UK
aw665@cam.ac.uk

## Abstract

Predictive tools are becoming widely used in police, courts, and prison systems worldwide. Criminal justice is thus an increasingly important application domain for machine learning and algorithmic fairness. A few benchmark datasets have received significant attention—e.g., COMPAS [1]—but often without proper consideration of the domain context [2]. We conduct a survey of publicly available criminal justice datasets, highlight their potential uses, discuss context, and identify limitations and gaps in the current landscape. We provide datasheets [3] for 15 datasets, and make them available via a public repository. We compare the surveyed datasets across several dimensions, including size, population coverage, and potential use, highlighting possible concerns. We hope this work provides a useful starting point for researchers looking for appropriate datasets related to criminal justice, and wish to further grow the repository in a broader community effort.

## 1 Introduction

Data is at the core of performance, safety, and fairness of predictive models [4, 5]. Yet data and their documentation are often overlooked and undervalued in machine learning (ML) [3, 6, 7]. Despite the increasing number of public datasets, ML researchers tend to focus on a few popular benchmarks [8, 9]. The lack of diversity in the modeling tasks in the ML literature is—at least in part—due to the difficulty of identifying datasets appropriate for particular research questions. Dataset surveys can serve as a helpful entry point in such a search. While dataset surveys exist in computer vision [10, 11], facial recognition [12], and algorithmic fairness [9, 13], several critical domains remain wanting.

Criminal justice is one such domain. Its relevance for ML researchers is underscored by the growing number of algorithmic tools used for high-stakes decision making in police, courts, and prison systems worldwide [14–19]. While some work suggests that the use of such tools can improve efficiency, consistency, and objectivity [20, 21], there are serious concerns. A growing body of work highlights the risks posed by biased algorithmic systems [22–25], including the use of data that encodes inequalities [14, 26, 27], and the potential for harmful feedback loops [14, 26]. Moreover, criminal justice datasets such as COMPAS [1], and the UCI Communities and Crime dataset [28], have become a common testbed ML methods, often without proper consideration of their context [2]. COMPAS is a particular staple in algorithmic fairness, with over 2000 citations to the original ProPublica article [1].

Failing to consider domain context can result in drawing misleading conclusions [2]. Criminal justice in particular poses several unique challenges. Firstly, identifying ground truth is often difficult. It is known that crime *observed* by law enforcement is often a poor proxy for the total of committed crimes [29, 30]. The probability a crime will become known to law enforcement can further vary

---

[*]Authors contributed equally to this research.

36th Conference on Neural Information Processing Systems (NeurIPS 2022) Track on Datasets and Benchmarks.

depending on the demographic attributes of an offender such as sex, race, and age [31–34]. Secondly, the available data tend to reveal only part of the picture. For example, we may know the offense, but might not posses essential information regarding the victim, or the defendant's criminal history.

Pertinent information is sometimes publicly available. However, using primary sources often requires significant effort. The relevant information is rarely held within a single agency, and linking datasets in a reliable manner is not easy. Even using existing datasets can be challenging, as data cookbooks can be hundreds of pages long, and often do not spell out potential uses or caveats. Data documentation frameworks [3, 7, 35] aim to standardize and improve the quality and usability of dataset documentation. Unfortunately, criminal justice dataset documentation rarely follows these guidelines. Researchers thus often spend significant time digesting documentation, gaining important knowledge and insight about potential uses and limitations in the process. A mechanism to share this knowledge with the community more easily would be highly valuable [36].

We create a datasheet repository for datasets related to the criminal justice domain. For each dataset, we provide broad context, outline potential uses, and discuss existing gaps and limitations. We hope this will facilitate dataset discovery, and encourage informed use of available criminal justice datasets within the ML community. We highlight the following contributions:

- A survey of publicly available datasets within the criminal justice domain.
- A model of data flow in the US criminal justice system, contextualizing the surveyed datasets.
- Detailed datasheets [3] for 15 of the surveyed datasets, made available via a public repository.

## 2   Dataset survey method

We conducted a survey of publicly available datasets related to criminal justice. We used the 'Discover Data' feature within the National Archive of Criminal Justice Data (NACJD) [37]. We further included several datasets that are popular in existing literature, but are not available via the NACJD. We used the following inclusion criteria: (i) US origin; (ii) tabular only; (iii) records only from after 2000; and (iv) publicly available.[2] We looked for datasets that report on a single instance level (e.g., crime, person). Datasets reporting aggregate data were included only when analogous information was not available from a non-aggregated source. Our search is not necessarily comprehensive, but we aimed to identify a variety of datasets with a diverse set of potential uses.

We selected 15 datasets for which to create datasheets.[3] We prioritized datasets we consider useful for the ML community; we chose based on several factors including size, population coverage, richness of variables, documentation quality, and the proportion of individual level over aggregate statistics. We also aimed for a diverse collection of datasets, covering various parts of the criminal justice system.

For each dataset, we completed a corresponding datasheet by the following process:

1. We used the template provided in [3] to complete a draft datasheet for each dataset.
2. Reviewing the drafts, we revised and tailored the template to criminal justice, adjusting, omitting, and adding several questions (e.g., 'Was the data directly observable?' was replaced by 'Was the information self-reported?'); we considered suggestions made in [7, 36] when crafting our revision. The questions in [3] are domain agnostic, most being suitable for our purposes. However, criminal justice datasets are prone to particular biases (e.g., regarding ethnicity), which we have tried to capture in our modified template. We anticipate continuing refinement of the template based on community feedback.
3. We revised all the datasheets according to the new template. The updated template and all datasheets are included as supplemental material. They can be downloaded from the project website, or a corresponding public repository.[4]

Using the completed datasheets, we extract the following information for each dataset: (i) instance type (survey response, criminal charge, etc.), (ii) stage (police, courts, etc.), (iii) dataset size, (iv) year collected, (v) geographical coverage, (vi) demographics, (vii) potential uses, (viii) dataset maintenance, and (ix) license for use. We then identify existing themes and gaps in the dataset landscape.

---

[2]Access restrictions may apply. We only create datasheets for datasets which do not require access approval.

[3]The Chicago Police Department dataset already had a datasheet prepared by its creators.

[4]The link to the project website is `https://criminaljustice-datasheets.github.io/`.

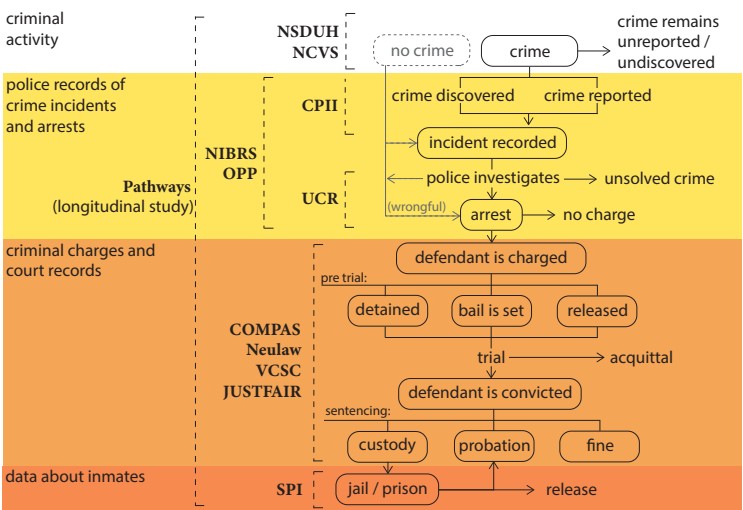

Figure 1: An illustration of the criminal justice pipeline, visualizing possible sequences of events after a crime has been registered. Dataset position alongside the pipeline indicates which section is covered by its data. The colors delineate separation between the different stages (e.g., police, courts), and highlight the escalation of a case from police investigation to a potential custodial sentence. The grey dotted pathway shows wrongful arrests or convictions. This diagram does not specify all possible pre-trial and sentencing outcomes (see [39] for a detailed diagram). See appendix for additional datasets.

# 3 The criminal justice system

Law enforcement can be conceptualized as a funnel-shaped pipeline (Figure 1), beginning with all potentially criminal activity at the top, then progressing down through stages to sentencing, and incarceration at the bottom [38, 39]. In this section, we give a general description of the criminal justice pipeline, contextualizing the available datasets, and identifying the gaps in the current landscape.

Starting at the top of Figure 1, not all illegal activity becomes known to the law enforcement. There are two main ways a criminal offense can become known: (1) *reported crime*, i.e., police is alerted by a victim or a third party; (2) *discovered crime*, i.e., a result of proactive policing efforts. Proactive efforts range from common 'stop-and-search' activities to 'sting operations'. Once a crime is reported or discovered, an incident is recorded by a law enforcement agency. When the perpetrator is unknown, the agency may dedicate resources to crime investigation which may lead to an arrest of a suspect.

Irrespective of whether an arrest takes place, the agency needs to decide if they will charge anybody with a crime. The period of time between pressing charges and the trial itself is commonly referred to as the pre-trial period. A judge or magistrate will usually decide on the suspect's pre-trial custody status. Depending on the severity of the crime and their criminal history, the suspect can be released outright, released conditionally on bail, or kept in custody. If bail is set, the suspect will only be released if they can raise the required amount. At the trial, a suspect can be found guilty or not guilty, or the case may be dismissed due to other reasons. If the suspect is found guilty, a sentence will be decided by a judge or magistrate. Charges turn into convictions if and when the court finds the suspect guilty.

The sentence may include one or a combination of: (a) a custodial sentence to be spent in jail (if under 12 months) or in prison; (b) a period of probation with specified conditions; (c) a monetary fine or compensation; (d) a caution. Sentencing guidelines may indicate the minimum and maximum terms for the sentence, including limits on the time of imprisonment and the amount that can be set for monetary fines.[5] The minimum sentence may depend on the criminal history of the offender. If the sentence does include an incarceration period, once the minimum period of incarceration has elapsed, the individual may be granted parole, and released on probation before serving their full sentence. For long sentences, a decision about whether to grant parole may take place periodically.

---

[5]Sentencing guidelines can be advisory (e.g., Federal Sentencing Guidelines), which means that the judge does not have to adhere to the recommended sentencing.

# 4 Datasets

We identified 30 datasets, described below. We provide datasheets for 15 of these in the supplementary material. Figure 1 shows the position of each dataset in the criminal justice pipeline. Details on the type of information contained in the datasets—e.g., criminal history, recidivism—are summarized in Table 1. Dataset size, composition, period covered by its records, maintenance status, and license are reported in Table 2. These tables act as an index, with detailed information provided in the datasheets.

## 4.1 Criminal activity

Datasets within this category contain data on criminal activity, including offenses not reported or discovered by law enforcement agencies.

The *National Crime Victimization Survey (NCVS) [40]* is a national annual survey conducted by the Bureau of Justice Statistics. A representative sample of approx. 240,000 people was asked whether they were victims of criminal behavior. NCVS collects information on personal crimes (e.g., rape, assault, and robbery), and property crimes (e.g., burglary and theft). If an incident occurred, the respondents provide information about the offender as well as characteristics of the crime, and whether the crime was reported to the police.

The *National Survey on Drug Use and Health (NSDUH) [41]* is a national annual survey of pprox. 70,000 individuals on their tobacco, alcohol and drug use, and health-related issues. Within the context of criminal justice, respondents provide details about their use of illegal substances and DUI, and whether these resulted in an interaction with the law enforcement. In the case of marijuana, respondents are also asked about their buying habits.

## 4.2 Longitudinal studies

Longitudinal studies follow the same individuals over a period of time. They provide data on both criminal activity and subsequent arrests.

The *Pathways to Desistance Study [42]* is a project that followed 1,354 serious juvenile offenders from adolescence to young adulthood. Participants between the ages of 14 and 18 were enrolled, each with at least one serious offense conviction. Each participant was followed for a period of seven years, with interviews conducted every 6 months for the first 3 years, and every 12 months thereafter. In addition, official arrest and court records were obtained for each participant. Among other topics, participants were asked about their offending, interactions with the justice system, and alcohol and drug use.

Researchers interested in longitudinal studies may also wish to consider: National Longitudinal Survey of Youth [43]; Rochester Youth Development Study [44]; Pittsburgh Youth Study [45]; and the Project on Human Development in Chicago Neighborhoods [46].

## 4.3 Police records on crime incidents and arrests

The *Stanford Open Policing Project [47]* is a source of standardized data on vehicle and pedestrian stops from law enforcement departments. The data is not comprehensive, i.e., not all stops are included, but it contains over 200 million records from the majority of state patrol agencies, and over 50 police departments. Data was obtained via public records requests. The project is maintained by researchers and journalists at Stanford University. Records contain all or some of the following: date and time of the stop, the reason for the stop, demographics of the individual stopped, and outcome (e.g. arrest, warning, and citation).

*National Incident-Based Reporting System (NIBRS) [48]* is the main program of data collection from law enforcement agencies in the US. The FBI collects incident-level data on crime from law enforcement agencies via NIBRS. An incident is defined as a set of offenses committed by one or more offenders at the same time and place. NIBRS reports detailed information on each recorded crime incident. Both offender and victim demographics are reported, as well as relationships between them. Information on whether an arrest was made, and its type (e.g., on view), is recorded. Property and weapons involved in the crime are also reported, as well as the time of day and type of location (e.g., street) in which the crime occurred. Not all law enforcement agencies report to NIBRS, but participation is growing, and there is complete population coverage for some states.

The *Collated Police Incident Index (CPII)* is a collection of 20 data sources with incident-level data from police departments, obtained from [49]. This data is less standardized and comprehensive than NIBRS. However, the crime location information is more fine-grained, which may enable other uses.

The *Uniform Crime Reporting (UCR) [50]* program reports aggregate data on arrests, collected from more than 18,000 law enforcement agencies. Although UCR provides much less detail than NIBRS, there are specific types of offenses (e.g., DUI) that are not included in NIBRS but are in UCR.

Researchers interested in data on police contact may also wish to consider the Police-Public Contact Survey [51], and the NYPD Stop-and-search dataset [52].

## 4.4 Criminal charges and court records

The *Correctional Offender Management Profiling for Alternative Sanctions (COMPAS)* is a predictive tool used by judges and parole officers. The tool produces an automated risk score to predict the probability of re-offending within a specific time frame. In 2016, ProPublica released a study [1] and an accompanying dataset of approx. 10,000 records obtained from Broward County, Florida. The dataset included the COMPAS risk scores, and any reported re-offending in the two year period following the original arrest. The study concluded that the COMPAS tool is racially biased, but follow-up research challenged this conclusion [53]. The COMPAS dataset was extended (not by the original creators) in 2020 to include probation data from the Broward County Clerk's Office [53].

The *NeuLaw Criminal Record Database [54]* contains millions of court records from 1977 to 2014 from Harris County in Texas, New York City, and Miami-Dade County in Florida. Each record reports the defendant's demographics, the offenses with which they are charged, and the progress of the case from charge to disposition. The Harris County data includes anonymized identifiers that enable the construction of a partial criminal record (containing only charges pressed within Harris County).

The *Virginia Pre-Trial Data Project (VCSC) [55]* followed 22,986 adult defendants charged with a criminal offense in October 2017.[6] Each case was tracked until final disposition or December 31, 2018, whichever came sooner. 700 variables are reported for each entry, including demographics, pending and current charges, probation status, release status and bond (when applicable), standardized criminal history, and predicted risk level. Additional charges, arrests and convictions during the follow-up period are also reported.

The *JUdicial System Transparency for Fairness through Archived/Inferred Records (JUSTFAIR) [56]* is a database of criminal sentencing decisions from Federal district courts. There are over 570,000 records from the 2001–2018 period, compiled from several public sources. Each record contains information about defendants' demographics, crimes they are charged with, the corresponding sentences, and the identity of the sentencing judge. The creators plan to extend this dataset to include state-level sentencing in the future.

Researchers interested in sentencing data may also consider the Pennsylvania Sentencing Data [57].[7]

## 4.5 Data about inmates

The *Survey of Prison Inmates (SPI) [58]* is a periodic survey of prison inmates in state and sentenced federal prisons, conducted by the Bureau of Justice Statistics. The latest cohort (2016) included 24,848 prisoner participants from 364 prisons. Each instance reports on the participant's demographics, criminal history, current sentence and corresponding offenses. The survey also includes questions on the individual's socioeconomic status, family, drug and alcohol use, and health.

Researchers interested in data on individuals incarcerated or in probation may also consider: the National Corrections Reporting Program (NCRP) [59]; and surveys that report aggregate statistics, such as the Annual Survey of Jails [60], the Annual Parole Survey, and the Annual Probation Survey [61].

## 4.6 Law enforcement agencies and officers

The *Law Enforcement Management and Administrative Statistics (LEMAS) [62]* is a periodic survey collecting data on law enforcement agencies, conducted by the Bureau of Justice Statistics. The

---

[6]This cohort was not unique in an identifiable manner; it is thus representative of any other month in Virginia.

[7]To gain access to this dataset, researchers need to fill in a data request form and pay a fee.

latest cohort (2016) covers over 3,000 agencies, including all with 100 or more full-time officers, and a sample of smaller agencies. Included is information on the number and demographic composition of officers and civilian employees, salaries, computer systems, and vehicles, as well as information on policies regarding weapons and armor, education, training and community policing. From 2016, a Body-Worn Cameras concerning supplement is included.

The *Chicago Police Department (CPD) Dataset [63]* contains information on approx. 35,000 officers in the CPD. The raw data was obtained via Freedom of Information requests. Each officer instance reports on demographics and period of service. Complaint instances contain information on the allegation, the officer involved, the investigation result, and following sanctions. Information about unit assignments, tactical actions, and officer salaries and awards is also available.

Researchers interested in data on law enforcement agencies and officers may also wish to consider Federal Law Enforcement Officers (CFLEO) [64].

## 4.7 Other topics

The *Profiles of Individual Radicalization in the US (PIRUS) [65]* dataset contains information on over 2,200 violent and non-violent extremists in the US. The dataset covers various ideologies (e.g., far-right, far-left). The data reports on the individuals' demographics, backgrounds, and radicalization processes, as well as the extent of violence they planned to commit, and whether their plan was executed.

Data on mortality within the criminal justice system can be found in the Mortality in Correctional Institutions dataset series [66]. Data on fatal police shootings can be found in the Washington Post's Fatal Force dataset [67].

## 5 Discussion

**The relationship between criminal activity and arrest is difficult to study.** From all the surveyed datasets, only three provided data directly on the underlying criminal activity. The NCVS reports on criminal activity from the victim's perspective, including whether or not they reported the crime to the police. The NSDUH includes self-reported illegal activity, but only in the context of substance use. The Pathways to Desistance study (and other longitudinal studies) contains data on self-reported offending, and whether that offending has led to interactions with law enforcement. This type of information is ideal for understanding the relationship between offending and subsequent arrests. However, this study follows a cohort of youth that have been convicted of a serious offense, not a representative sample of the general population. Due to the challenging nature of collecting information on baseline criminality, this gap in the data landscape is not surprising. Nonetheless, the available information does indicate that the probability an illegal activity will lead to an arrest depends on many factors including the type, location, and time of the crime, as well as the offender and victim's demographics [34, 47, 68].

**Data on recorded incidents is rich but disconnected.** Data on interactions between individuals and the police can be found in several disconnected sources. The Open Policing Project provides data on stops made by the police that may or may not have led to the discovery of weapons, illegal substances or other contraband, and a subsequent arrest. The agency will record an incident only when something is found during the stop. Each agency has a local record of incidents and arrests, and in some jurisdictions, an anonymized version of these records is publicly available (see CPII above). NIBRS is a rich dataset that contains information on offenders and victims, the circumstances of the crime, and whether an arrest was made, for most recorded incidents.[8] Although it significantly improves on the aggregate arrest data available via UCR, several papers have been written on the limitations of NIBRS [70, 71]. We do not know how each incident became known to the agency (e.g., a 911 call, a pedestrian search). We also cannot tell whether an arrestee was charged or convicted of the offense. In addition, none of the incidents and arrests datasets includes a unique ID, which means that they cannot be used to study recidivism or repeat victimization.

---

[8]NIBRS only reports on 'group A' offenses, which excludes some illegal activities (e.g., DUI and trespassing). A full list of group A and group B offenses can be found in [69].

Table 1: Information type and demographic coverage

| Name | Stage | Criminal History | Recidivism | Defendant Demographics | Victim Demographics | Hispanics |
|---|---|---|---|---|---|---|
| NSDUH | crime | no | no | yes | n/a | yes |
| NCVS | crime → arrest | no | no | yes | yes | yes |
| Pathways | crime → 7 years later | yes | yes | yes | yes | yes |
| OPP | incident → arrest (partial[1]) | no | no | partial[1] | n/a | partial[1] |
| CPII | incident | no | no | no | no | no |
| NIBRS | incident → arrest | no | no | yes | yes | partial[1] |
| UCR | arrest | no | no | partial[2] | no | no |
| COMPAS | charge → 2 year later | partial[3] | yes | yes | no | yes |
| Neulaw | charge → disposition | partial[3] | no | yes | no | estimate[4] |
| VCSC | charge → disposition | yes | yes | yes | no | no |
| JUSTFAIR | trial | no | no | yes | no | yes |
| SPI | prison | no | no | yes | no | yes |
| LEMAS | n/a | n/a | n/a | officer demographics | | yes |
| CPD | n/a | n/a | n/a | officer demographics | | yes |
| PIRUS | n/a | yes | no | yes | no | yes |

Table 2: Dataset size, composition, maintenance status and license

| Dataset | Data Instance | Size | Sample | Geographic Resolution | Years | Maintained | License |
|---|---|---|---|---|---|---|---|
| NSDUH | survey responses | ~50K | general population | national | 1979–ongoing | yes | public domain |
| NCVS | survey responses | 240K (persons) | general population | national | 1979–ongoing | yes | public domain |
| Pathways | repeat interviews and official records | 1354 | youth offenders | local (multi) | 2000–2003 | no | public domain[5] |
| OPP | pedestrian and vehicle stops | 200M+ | general population[6] | local (multi) | 2000–2020 | yes | ODC-BY 1.0 |
| CPII | recorded incidents | 20 × (10K - 1M) | general population[6] | local (multi) | 2018 | yes | varies |
| NIBRS | recorded incidents | 3M–7M (p/a) | general population[6] | national (part) | 2011–ongoing | yes | public domain |
| UCR | arrests (aggregate) | ~18K (p/a) | arrestees | national (part) | 1995–ongoing | yes | public domain |
| COMPAS | criminal charges | ~10K | arrestees | local | 2016 | yes | not specified |
| Neulaw | criminal charges | 22.5M+ | arrestees | local | 1977–2014 | no | public domain |
| VCSC | criminal charges | ~23K | arrestees | local | 2017 | yes | not specified |
| JUSTFAIR | court records | ~600K | arrestees | national | 2001–2018 | no | public domain |
| SPI | survey responses | ~ 25K (in 2016) | prisoners | state | 1974–2016[7] | no | public domain |
| LEMAS | survey responses | ~ 3K | police agencies | sub-county | 1987–2016[7] | no | public domain |
| CPD | officers, units, complaints, use-of-force reports | 35K, 105K, 109K, 10.5K | all Chicago PD officers | local | 2017–2019 | no | MIT |
| PIRUS | data on individuals | ~ 2200 | radicalized individuals | national | 1948–2018 | yes | public domain |

[1]Only included in data from some agencies. [2]Only in aggregate form. [3]Only prior offending within the same locality are included. [4]Ethnicity is estimated based on name. [5]Some variables are restricted. [6]Members of the general population that have had contact with law enforcement. [7]Data collection was done in irregular intervals.

**Only one dataset provides full criminal history and data on re-offending.** Four of the publicly available datasets that we listed report on criminal charges, pre-trial and court cases: COMPAS, Neulaw, VCSC and JUSTFAIR. Of these, only VCSC contains information about the defendant's full criminal history, and that in a standardized form which may not be suitable for all research questions. COMPAS and part of the Neulaw dataset contain unique identifiers, so a partial construction of a criminal record is possible. Neither of these datasets contains information regarding the victim, which may affect pre-trial and sentencing outcomes. Recidivism is only reported in VCSC and the COMPAS dataset (both original [1] and revised [53]). Both these datasets report on predicted risk and proven re-offending. VCSC reports on transparent, rule-based risk assessment instruments, and provides the components for calculating the risk score, in addition to the final score. Non-aggregated information on individuals held in custody who are subject to probation is limited. We identified two datasets, SPI and NCRP. For the latter, only selected variables are not restricted.

**Survey based data vs. admin data.** Datasets differ by collection method: (a) *survey datasets* (e.g., NVCS, NSDUH, SPI), where individuals may decline to participate; (b) *administrative datasets* recorded as part of routine police or court work (e.g., NIBRS, VCSC), where individuals have no option to opt-out, or give consent to third-party use. For the latter, we can also consider whether the data is published by an official source. While some datasets are made public directly by law enforcement agencies and governmental bodies such as the Bureau of Justice, others have been compiled by journalists and academics mainly via freedom of information (FOI) requests. Datasets compiled via FOI requests often cover limited geographical areas, and are not always maintained.

**Combining datasets is challenging.** None of the datasets mentioned can be linked at the instance level. We highlight that any future work to enable this would be extremely helpful, though we recognize it would be very challenging. Information may be combined at an aggregate level, extending the possibilities of analysis [34]. Problematically, categorizations (e.g., of age, race) and geographic coverage may not be well aligned, as same type of information may be recorded inconsistently based on locality. For example, ethnicity can be filled in by the officer, the victim, or the accused in different counties; the available documentation does not always clarify this. Due to these varying local practices, Hispanics are reported as White in NIBRS by several contributing agencies.

**Bias may be present at every stage of the pipeline.** Most ML work in criminal justice is focused on racial bias and algorithmic fairness. While there is value in considering bias at every step (e.g., from reported crime to arrest, at sentencing), selection bias may occur earlier in the process: not all who offend are as likely to get stopped, arrested, charged, or to receive a custodial sentence [34, 47, 68, 72, 73]. Disparities occur throughout the criminal justice system [74–76], and affect what entries are captured in the individual datasets as illustrated by the funnel in Figure 1.

While general trends are clear [77, 78], the directionality of disparities is complex on a more fine-grained level. For example, police were more likely to stop white than black individuals for speeding violations, but the opposite was true when the stop was made to check the driver's records [79]. If the stop reason is not reported in the data, similar effects may go unnoticed. When a dataset only contains arrests, the extent of this selection bias cannot be measured. Furthermore, local laws may influence the extent of local disparities. As an example, the different laws regarding crack and powder cocaine, as well as indoor and outdoor drug sales, cause racial disparities in drug arrests in Seattle [80].

Another bias comes from wrongful arrests and convictions. Black individuals are estimated to be more likely to be wrongfully convicted than white individuals for sexual assault (3.5x), murder (7x), and drug crimes (12x) [81], and overall receive longer sentences. This is partly due to prosecutors more often charging black defendants with crimes that carry mandatory minimum sentences [82, 83]. This type of bias is often hard to measure and mitigate within the data. If not carefully considered, these disparities can be reinforced by sociotechnical systems. Contextual awareness can help researchers consider and take these into account, and caveat their results and conclusions accordingly.

**Benefits and risks of using criminal justice datasets in ML research.** Our overarching aim is to promote responsibility in criminal justice ML. The datasets we describe can lead to benefits including:

1. *Development of new methods (e.g., fairness, explainability) grounded in real-world scenarios.* Compared to common benchmarks, the surveyed datasets can allow researchers to tackle challenging, domain-specific, and overall more realistic use-cases. Examples include

predicting rare but critically important labels (e.g., PIRUS), spatial-temporal patterns (e.g., CPII), and finding counterfactuals (e.g., JUSTFAIR). The datasets may provide a hard but particularly useful challenge for algorithmic fairness, as they require carefully disentangling of disproportionality from bias and discrimination. Understanding and addressing such real-world challenges is critical for responsible work in the criminal justice domain, and is much needed in deployed applications [23, 38].

2. *Investigation of the criminal justice system.* While some of the datasets have already been used by criminologists, ML methods have rarely been applied. For this line of research, we highly recommend engaging with the relevant literature and domain experts to ensure proper context and interpretation of results.

3. *Development of new benchmarks inspired by real-world use-cases.*

Increasing visibility of the surveyed datasets can however also lead to *intentional* and *unintentional* misuse. We condemn any deliberate misuse such as training harmful algorithms on the datasets. For unintentional misuse, [2] highlights irresponsible use of recidivism data, and the harms caused by ignoring context. One may wonder if it would be better to simply not draw attention to the datasets. We disagree as informed responsible engagement with the data can lead to significant societal benefits, like uncovering the bias propagated by 'innocuous' technical systems [14, 23]. We believe that transparency and the public nature of the surveyed datasets will promote more open rigorous debate about the limitations, and appropriate use of ML in criminal justice, thus counteracting misuse.

**Linking data sources, privacy and the risk of re-identification.**   Combing several datasets can be very informative (see previous discussion of complementarity of the surveyed datasets). The datasets are however not suitable for data linking on entry-level as re-identification is usually not possible.[9] Since criminal charges are not private information in the US, re-identification is conceivable in a fraction of cases by cross-referencing criminal and press reports. However, this requires considerable deliberate effort, while being unlikely to reveal significant new information. Finally, while we highlighted advantages of data linkage for improving our understanding of the criminal justice system, this has to be balanced with protecting the privacy of involved parties. Our hope is that significant progress can be made without requiring access to sensitive information. An example of such a non-invasive but particularly useful variable is the 'reason for stop'.

**Maintenance, updates, and geographical coverage.**   We hope this work is the beginning of a useful growing project with community feedback and support. We will be accepting GitHub pull requests with contributions to the datasheets, the template, and additional datasets. Major changes will be reviewed and implemented annually. Relative to rest of the globe, the US justice system has been a subject of disproportionate attention, despite not being representative of many territories. However, similar datasets covering other territories (e.g., UK) are often not public. This—combined with the desire to keep the survey manageable—has led us to focus on the US as a starting point. In the future, we intend to expand the coverage of this project to many more geographical locations.

**Limitations.**   The presented information and accompanying datasheets are correct to the best of our knowledge, based on available documentation. We recommend that researchers consult the official documentation when using the datasets. Efforts have been made to include available datasets that answer the inclusion criteria. However, it is impossible to guarantee survey comprehensiveness. We are not the creators of the datasets, and do not provide endorsement or guarantees regarding these datasets.

## 6   Conclusion

We survey publicly available datasets within the criminal justice domain. We provide context by placing the datasets within the criminal justice pipeline. We extract detailed information on 15 diverse datasets, and—through an iterative process—create a datasheet for each. We believe these datasheets will help researchers interested in working with criminal justice data to learn which datasets are available, and make informed decisions regarding their specific needs [36]. The datasheets are available as supplementary material to this paper, and in a public repository. The authors plan to continue to grow the repository of datasheets, and encourage community contributions in order to create a reliable and comprehensive source for datasets related to the criminal justice domain.

---

[9]COMPAS is an exception (includes names). We included it since it is a widely known and used dataset.

## Acknowledgments and Disclosure of Funding

The authors thank Riccardo Fogliato and Jiri Hron for valuable discussion and comments. BB acknowledges support from the European Research Council (ERC), grant agreement No 851538. MZ acknowledges support from the Leverhulme Trust grant ECF-2021-429. AW acknowledges support from a Turing AI Fellowship under grant EP/V025279/1, The Alan Turing Institute, and the Leverhulme Trust via CFI.

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
