# Supplementary Material

| Dataset | URL |
|---|---|
| COMPAS | https://github.com/propublica/compas-analysis |
| CPD | https://github.com/chicago-police-violence/data |
| CPII | URLs can be found in the CPII datasheet |
| JUSTFAIR | https://qsideinstitute.org/research/criminal-justice/justfair/ |
| LEMAS | https://bjs.ojp.gov/data-collection/law-enforcement-management-and-administrative-statistics-lemas |
| NCVS | https://www.icpsr.umich.edu/web/NACJD/series/00095 |
| NeuLaw | https://www.openicpsr.org/openicpsr/project/100360/version/V1/view |
| NIBRS | https://crime-data-explorer.fr.cloud.gov/pages/downloads |
| NSDUH | https://www.datafiles.samhsa.gov/dataset/national-survey-drug-use-and-health-2019-nsduh-2019-ds0001 |
| OPP | https://openpolicing.stanford.edu/ |
| Pathways | https://www.pathwaysstudy.pitt.edu/ |
| PIRUS | https://www.start.umd.edu/data-tools/profiles-individual-radicalization-united-states-pirus |
| UCR | https://crime-data-explorer.fr.cloud.gov/pages/downloads |
| VCSC | http://www.vcsc.virginia.gov/pretrialdataproject.html |
| SPI | https://bjs.ojp.gov/data-collection/survey-prison-inmates-spi |

Table 1: A table of download links for each of the 15 datasets surveyed.

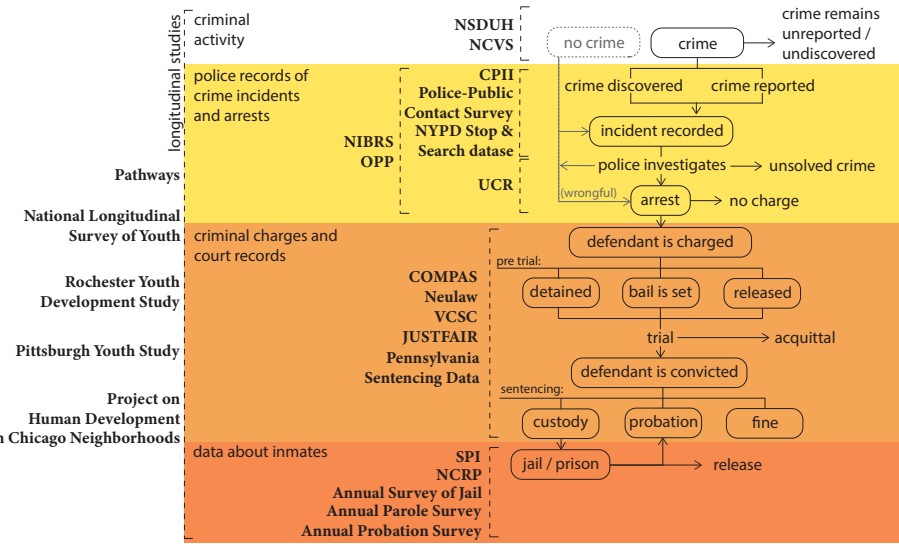

Figure S 1: An illustration of the criminal justice pipeline, visualizing the possible sequence of events once a crime has been committed. Appropriate datasets have been added alongside the pipeline, indicating which section of the pipeline their data addresses. The grey dotted pathway indicates wrongful arrests or convictions. The colors illustrate the separation between the different stages (e.g., police, courts) and highlight the escalation of a case from police investigation to a potential custodial sentence.

## Datasheet Index

| Datasheet | Page Start |
| --- | --- |
| NSDUH | Page 4 |
| NCVS | Page 7 |
| Pathways | Page 11 |
| OPP | Page 15 |
| CPII | Page 19 |
| NIBRS | Page 23 |
| UCR | Page 28 |
| COMPAS | Page 31 |
| Neulaw | Page 34 |
| VCSC | Page 38 |
| JUSTFAIR | Page 42 |
| SPI | Page 46 |
| LEMAS | Page 49 |
| CPD | Page 53 |
| PIRUS | Page 57 |

Table 2: Index table of datasheets, with specified page numbers

# National Survey on Drug Use and Health (NSDUH)
## Datasheet

## I. MOTIVATION

### I-A  For what purpose was the dataset created?

The National Survey on Drug Use and Health (NSDUH) was created to provide up-to-date information on tobacco, alcohol, and drug use, mental health and other health-related issues in the United States [1].

The data provide estimates of substance use and mental illness at the national, state, and substate levels. NSDUH data also help to identify the extent of substance use and mental illness among different subgroups, estimate trends over time, and determine the need for treatment services [1].

### I-B  Who created the dataset?
*Is it an official law enforcement or government body? An academic research team? Other?*

The dataset is created by the Substance Abuse and Mental Health Services Administration (SAMSA), with data collection and analysis conducted under contract with RTI International.

### I-C  Was there a specific task in mind, or gap that needed to be filled?

NSDUH was created to support prevention and treatment programs, monitor substance use trends, estimate the need for treatment and inform public health policy. Prior to NSDUH, there was no annual health survey that reported on substance abuse and other health issues.

## II. COMPOSITION

### II-A  What do the instances that comprise the dataset represent?
*For example: crimes, offenders, court cases, police officers*

Instances in the dataset comprise of individual survey responses.

### II-B  Are there multiple types of instances?
*For example: offenders, victims, and the relationship between them.*

No.

### II-C  How many instances are there in total?
*Of each type, if appropriate.*

The dataset is released annually, containing information from $\sim 55,000$ respondents. 2020 was an exception due to the Covid-19 pandemic, when there were only $\sim 30,000$ respondents.

### II-D  Does the dataset contain all possible instances or is it a sample (not necessarily random) of instances from a larger set?
*For example, if it is traffic stops from a territory, is it all traffic stops conducted within that territory within a specific time? If not, is it a representative sample of all stops? Describe how representativeness was validated/verified. If it is not representative, please describe why.*

Survey responses are collected from a sample of the population. The target population of the survey is defined as the civilian, noninstitutionalized population of the United States. NSDUH collects information from residents of households and non-institutional group quarters (e.g., shelters, rooming houses, dormitories) and from civilians living on military bases. The survey excludes homeless people who do not use shelters, military personnel on active duty, and residents of institutional group quarters, such as jails and hospitals [2].

### II-E  What data does each instance consist of?
*If there is a large number of variables, please provide a broad description of what is included.*

Each data instance includes age at first use, as well as lifetime, annual, and past-month use of the following drugs:

- alcohol
- marijuana
- cocaine (including crack)
- hallucinogens
- heroin
- inhalants
- tobacco
- pain relievers
- tranquilizers
- stimulants
- sedatives

Respondents are asked about personal and family income, health care access and coverage, illegal activities and arrest records, problems resulting from the use of drugs, and perceptions of risks. For marijuana, respondents are asked about how and how often they obtain the drug. Demographic data collected include gender, race, age, ethnicity, educational level, employment status, income level, veteran status, household composition, and population density.

### II-F  Is there a target label or associated with each instance?
*Please include labels that are likely to be used as target labels, e.g. recidivism.*

No.

*II-G   Are there recommended data splits (e.g., training, development/validation, testing)?*
    If so, please provide a description of these splits, explaining the rationale behind them.

No.

*II-H   Does the dataset contain data on race and ethnicity?*
    If so, is it based on the individual's self-description, or based on officer's impression? Was it collected or derived in post-processing? For example, by name analysis.

Yes, both are self-reported by the respondent.

*II-I   Are there any known errors, sources of noise, bias or missing data, or variables collected for only part of the datasets?*
    If so, please provide a description.

Missing and erroneous data are dealt with in pre-processing. For detailed information, see section 2.3.2 and 2.3.3 from the documentation [2].

*II-J   Does the dataset contain data on criminal history or other data that might be considered confidential or sensitive in any way?*
    For example: sexual orientations, religious beliefs, political opinions or union memberships, or locations; financial or health data; biometric or genetic data; forms of government identification, such as social security numbers; If so, please provide a description.

Yes, the data contains information on substance use and abuse as well as confidential medical information on each subject. In addition, respondents are asked about criminal activity and arrests. Demographic information and information of income and employment is also disclosed.

*II-K   Is it possible to identify individuals (i.e., one or more natural persons), either directly or indirectly (i.e., in combination with other data) from the dataset?*
    If so, please describe how.

No.

## III. USES

*III-A   Has the dataset been used for any tasks already?*
    If so, please provide a description.

The dataset has been used in over 1,700 publications since it's establishment in 1979. A repository of which can be found:

https://www.icpsr.umich.edu/web/ICPSR/series/64

*III-B   Is there a repository that links to any or all papers or systems that use the dataset?*
    If so, please provide a link or other access point.

Yes. Please see above.

*III-C   What (other) tasks could the dataset be used for?*
    For example: testing predictive policing systems, predicting recidivism.

The NSDUH dataset can be used for to investigate questions around drug use and abuse, as well as its relationship to mental health and involvement with the criminal justice system. Age at first use, past-month, annual, and lifetime use is reported for many drugs (listed in a subsequent section), as well as treatment history. The following demographics are also self-reported: age, race, sex, level of education, employment status, income, and veteran status.

*III-D   Is there anything about the composition of the dataset or the way it was collected and prepro-cessed/cleaned/labeled that might impact future uses?*
    For example, is there anything that a dataset consumer might need to know to avoid uses that could result in unfair treatment of individuals or groups (e.g., stereotyping, quality of service issues) or other risks or harms (e.g., legal risks, financial harms)? If so, please provide a description. Is there anything a dataset consumer could do to mitigate these risks or harms?

The data is self-reported and is not corroborated in any-way. Although the survey is anonymous, respondents may self-report the amount of drug use or not disclose other details. This may be more prevalent in some demographics compared to others.

Due to the Covid-19 pandemic, collection methodology changed for 2020, making direct comparison to other years difficult.

## IV. COLLECTION PROCESS

*IV-A   How was the data associated with each instance acquired?*
    e.g. the data collected survey, the raw data is routinely collected by the courts.

The data collection methods used for NSDUH to conduct in-person interviews with sampled individuals. Confidential-ity is stressed in all written and oral communications with potential respondents. Respondents' names are not collected with the data, and computer-assisted interviewing (CAI) methods are used to provide a private and confidential setting to complete the interview.

*IV-B   Was the information self-reported?*
    If the data was self-reported, was the data validated/verified? If so, please describe how.

Yes. The data consists of the survey responses, although some variables are retracted in the publicly available version of the data.

*IV-C   Who was involved in the data collection process?*
    Was this done as part of their other duties? If not, were they compensated?

A scientific random sample of household addresses are selected across the United States. Once selected, no other address be substituted for any reason. At the end of the

completed interview, participants receive $30 as a token of appreciation for their help. Interviews are facilitated by paid field interviewers [2].

**IV-D**    *Over what timeframe was the data collected? Does this timeframe match the creation timeframe of the data associated with the instances (e.g., recent crawl of old news articles)?*
   If not, please describe the timeframe in which the data associated with the instances was created. If the collection was not continuous within the timeframe, please specify the intervals, for example, annually, every 4 years, irregularly.

Data is collected on an annual basis, with surveys taking place throughout the year. Data is available from the year 1979 onward.

**IV-E**    *Were any ethical review processes conducted (e.g., by an institutional review board)?*
   If so, please provide a description of these review processes, including the outcomes, as well as a link or other access point to any supporting documentation.

All projects involving human subjects must be approved by SAMHSA's Office of Research Protection, which serves as RTI's Institutional Review Board (IRB) under federal regulations. This committee looks very closely at the written introduction to the study to be sure the respondents are being properly informed [3].

**IV-F**    *Were the individuals in question notified about the data collection? Did they give their consent?*
   If consent was obtained, were the consenting individuals provided with a mechanism to revoke their consent in the future or for certain uses?

Yes, responding to the survey is optional.

**IV-G**    *Has an analysis of the potential impact of the dataset and its use on data subjects (e.g., a data protection impact analysis) been conducted?*
   If so, please provide a description of this analysis, including the outcomes, as well as a link or other access point to any supporting documentation.

Unknown.

## V. Pre-processing, cleaning, labeling

**V-A**    *Was any preprocessing/cleaning/labeling of the data done (e.g., discretization or bucketing, tokenization, part-of-speech tagging, SIFT feature extraction, removal of instances, processing of missing values)?*
   If so, please provide a description. If not, you may skip the remaining questions in this section.

Pre-processing steps are described in sections 2.3.2 and 2.3.3 in the documentation [2].

**V-B**    *Was the "raw" data saved in addition to the preprocessed/cleaned/labeled data?*
   If so, please provide a link or other access point to the "raw" data.

SAMHSA will have access to the raw survey responses, but these are not publically available.

**V-C**    *Is the software that was used to preprocess/clean/label the data available?*
   If so, please provide a link or other access point.

No.

## VI. Distribution

**VI-A**    *Is the data publicly available? How and where can it be accessed (e.g., website, GitHub)?*
   Does the dataset have a digital object identifier (DOI)?

The dataset is avilable to download on the SAMSA website, here.

**VI-B**    *Is the dataset be distributed under a copyright or other intellectual property (IP) license, and/or under applicable terms of use (ToU)?*
   If so, please describe this license and/or ToU, and provide a link or other access point to, or otherwise reproduce, any relevant licensing terms or ToU, as well as any fees associated with these restrictions.

No license is mentioned on the website or in the codebook.

## VII. Maintenance

**VII-A**    *Is the dataset maintained? Who is supporting/hosting/maintaining the dataset?*

Yes. The dataset is supported and hosted by SAMSA in collaboration with RTI international.

**VII-B**    *How can the owner/curator/manager of the dataset be contacted (e.g., email address)?*

RTI international, the company which manage the collection and processing can be contacted at: NSDUH-Helpdesk@rti.org

**VII-C**    *Will the dataset be updated (e.g., to correct labeling errors, add new instances, delete instances)?*

New data is released annually.

**VII-D**    *Are older versions of the dataset continue to be supported/hosted/maintained?*

Yes. Data from previous years continue to be hosted.

**VII-E**    *If others want to extend/augment/build on/contribute to the dataset, is there a mechanism for them to do so?*
   If so, please provide a description.

No.

## References

[1] SAMHSA, "National survey on drug use and health (NSDUH) population," 2021. [Online]. Available: https://www.datafiles.samhsa.gov/dataset/national-survey-drug-use-and-health-2020-nsduh-2020-ds0001

[2] S. Abuse and Mental Health Services Administration, "2020 National Survey on Drug Use and Health (NSDUH): Methodological Summary and Definitions," 2021. [Online]. Available: https://www.samhsa.gov/data/sites/default/files/reports/rpt35330/2020NSDUHMethodSummDefs091721.pdf

[3] ——, "2019 National Survey on Drug Use and Health (NSDUH): Field Interviewer Manual," 2020. [Online]. Available: https://www.samhsa.gov/data/sites/default/files/reports/rpt23074/NSDUHmrbFIManual2019.pdf

# National Crime Victimization Survey (NCVS)
# Datasheet

## I. MOTIVATION

### I-A    For what purpose was the dataset created?

The National Crime Victimization Survey (NCVS) series was designed to achieve four primary objectives [1]:

1) To develop detailed information about the victims and consequences of crime
2) To estimate the number and types of crimes not reported to police
3) To provide uniform measures of selected types of crime
4) To permit comparisons over time and types of areas

### I-B    Who created the dataset?
Is it an official law enforcement or government body? An academic research team? Other?

The survey is administered by the U.S. Census Bureau (under the U.S. Department of Commerce) on behalf of the Bureau of Justice Statistics (under the U.S. Department of Justice).

### I-C    Was there a specific task in mind, or gap that needed to be filled?

The NCVS began in 1972 and was developed following a survey done by the National Opinion Research Center and the President's Commission on Law Enforcement and Administration of Justice. The survey highlighted that many crimes were not reported to the police. The NCVS was created to assess the levels of and gain better understanding of criminal victimization, including from crimes that were never reported to law enforcement [1].

## II. COMPOSITION

### II-A    What do the instances that comprise the dataset represent?
For example: crimes, offenders, court cases, police officers

Instances in NCVS correspond to a record of a criminal victimization incident.

### II-B    Are there multiple types of instances?
For example: offenders, victims, and the relationship between them.

Yes, there are four types of records:

1) Address ID Record
2) Household Record
3) Person Record
4) Incident Record

Each person can have multiple incident records, each household can include several persons.

### II-C    How many instances are there in total?
Of each type, if appropriate.

Data is collected bi-annually, from a nationally representative sample of $\sim 49,000$ households comprising $\sim 100,000$ persons on the frequency, characteristics, and consequences of criminal victimization in the United States. The number of incidents will vary from year to year.

### II-D    Does the dataset contain all possible instances or is it a sample (not necessarily random) of instances from a larger set?
For example, if it is traffic stops from a territory, is it all traffic stops conducted within that territory within a specific time? If not, is it a representative sample of all stops? Describe how representativeness was validated/verified. If it is not representative, please describe why.

The dataset contains a sample of 100,000 persons from the United States. Weights are provided at the person, household, and incident level to produce a representative sample of the US [1]. Excluded are persons who are crews of vessels, in institutions (e.g., prisons and nursing homes) or members of the armed forces living in military barracks [1]. Once in the sample, respondents are interviewed every six months for a total of seven interviews over a three-year period.

### II-E    What data does each instance consist of?
If there is a large number of variables, please provide a broad description of what is included.

Each instance consists of the following data [1]:

- Type of crime
- Date of crime
- Location type of crime (e.g., at home, at school.)
- Relationship between victim and offender
- Offender characteristics
- Actions taken by the victim (e.g., resisted, escaped.)
- Consequences of victimization (e.g., distress, emotional toll.)
- Type of property lost
- Crime reported
- Reasons for reporting/not reporting
- Weapons used
- Drugs involved
- Alcohol involved
- Demographic information including:
  - Age
  - Race
  - Gender
  - Income

**II-F   Is there a target label or associated with each instance?**
Please include labels that are likely to be used as target labels, e.g. recidivism.

No variable in the dataset is designated as a label. However, whether or not a crime was reported and whether or not an arrest was made following that report may be suitable as target labels [2].

**II-G   Are there recommended data splits (e.g., training, development/validation, testing)?**
If so, please provide a description of these splits, explaining the rationale behind them.

No.

**II-H   Does the dataset contain data on race and ethnicity?**
If so, is it based on the individual's self-description, or based on officer's impression? Was it collected or derived in post-processing? For example, by name analysis.

Yes, race and ethnicity are reported for the victim, and sometimes for the offender as well.

For the victim, that is, the survey respondent, race and ethnicity are self-reported. The race categories are: White, Black or African American, American Indian or Alaska Native, Asian, Native Hawaiian or Other Pacific Islander, and Other. Only Hispanic ethnicity is recorded.

For the offender, race and ethnicity are perceived and reported by the respondent, i.e., the victim, if they saw the offender. The race categories are: Mostly White, Mostly Black or African American, Mostly American Indian or Alaska Native, Mostly Asian, Mostly Native Hawaiian or Other Pacific Islander, Equal number of each race, and Don't Know. The ethnicity categories are: mostly Hispanic or Latino, mostly non-Hispanic, equal number of Hispanic and non-Hispanic, and don't know.

**II-I   Are there any known errors, sources of noise, bias or missing data, or variables collected for only part of the datasets?**
If so, please provide a description.

Weights are provided on a person, household and incident level to produce a representative sample of the US. Prisoners are excluded from the sample.

**II-J   Does the dataset contain data on criminal history or other data that might be considered confidential or sensitive in any way?**
For example: sexual orientations, religious beliefs, political opinions or union memberships, or locations; financial or health data; biometric or genetic data; forms of government identification, such as social security numbers; If so, please provide a description.

The experiences of criminal victimization themselves and their consequences can be considered sensitive, especially for sexual assault and rape. In addition, the dataset contains information on: age, race, gender, and income.

**II-K   Is it possible to identify individuals (i.e., one or more natural persons), either directly or indirectly (i.e., in combination with other data) from the dataset?**
If so, please describe how.

No, the data is sufficiently anonymized.

## III. USES

**III-A   What type of tasks, if any, has the dataset been used for?**
If so, please provide examples and include citations.

The dataset has been used for a range of victimization studies, looking at things such as:

- Assessment of crime levels in the United States.
- Comparing Victimization across demographics.
- Comparing Victimization of specific types of crime across demographics.
- Assessing the *dark figure of crime*.[1]

**III-B   Is there a repository that links to any or all papers or systems that use the dataset?**
If so, please provide a link or other access point.

Yes. Papers that cited this dataset can be found in: https://www.icpsr.umich.edu/web/NACJD/series/95/publications.

**III-C   What (other) tasks could the dataset be used for?**
For example: testing predictive policing systems, predicting recidivism.

This dataset can be used for a variety of tasks which requires an understanding of the level of victimization for specific crimes, with demographic information on both the victim and offender, and information on whether the crime was reported and an arrest was made.

**III-D   Is there anything about the composition of the dataset or the way it was collected and preprocessed/cleaned/labeled that might impact future uses?**
For example, is there anything that a dataset consumer might need to know to avoid uses that could result in unfair treatment of individuals or groups (e.g., stereotyping, quality of service issues) or other risks or harms (e.g., legal risks, financial harms)? If so, please provide a description. Is there anything a dataset consumer could do to mitigate these risks or harms?

According to the authors of a book chapter on the potential sources of error in the NCVS [3], there are questions as to whether the rape and sexual assault are underestimated as they do not align with alternative surveys. "The Bureau of Justice Statistics does not provide public information on the edit process in the National Crime Victimization Survey, although processing and editing errors are an important part of any major survey data collection. The lack of transparency about these processes makes it difficult for data users to fully understand the survey's estimate" [3].

---

[1] The dark figure of crime is term used to illustrate the extent of committed crimes that are never reported or discovered by law enfrocment.

## IV. Collection Process

**IV-A**  *How was the data associated with each instance acquired?*
e.g. the data collected survey, the raw data is routinely collected by the courts.

The data was acquired from a bi-annual survey.

**IV-B**  *Was the information self-reported?*
If the data was self-reported, was the data validated/verified? If so, please describe how.

Yes. The data is collected in a survey. However, the raw survey responses are not provided.

**IV-C**  *Who was involved in the data collection process?*
Was this done as part of their other duties? If not, were they compensated?

The survey is administered by the U.S. Census Bureau.

**IV-D**  *Over what timeframe was the data collected? Does this timeframe match the creation timeframe of the data associated with the instances (e.g., recent crawl of old news articles)?*
If not, please describe the timeframe in which the data associated with the instances was created. If the collection was not continuous within the timeframe, please specify the intervals, for example, annually, every 4 years, irregularly.

The data is collected twice a year and released once a year. Data is available from the year 1979 and collection is still ongoing.

**IV-E**  *Were any ethical review processes conducted (e.g., by an institutional review board)?*
If so, please provide a description of these review processes, including the outcomes, as well as a link or other access point to any supporting documentation.

Unknown.

**IV-F**  *Were the individuals in question notified about the data collection? Did they give their consent?*
If consent was obtained, were the consenting individuals provided with a mechanism to revoke their consent in the future or for certain uses?

Yes, the survey is optional.

**IV-G**  *Has an analysis of the potential impact of the dataset and its use on data subjects (e.g., a data protection impact analysis) been conducted?*
If so, please provide a description of this analysis, including the outcomes, as well as a link or other access point to any supporting documentation.

Unknown.

## V. Pre-processing, cleaning, labeling

**V-A**  *Was any preprocessing/cleaning/labeling of the data done (e.g., discretization or bucketing, removal of instances, processing of missing values)?*
If so, please provide a description and reference to the documentation. If not, you may skip the remaining questions in this section.

Yes, as the survey responses were processed into the data available in the dataset. However, information on pre-processing is not supplied in the codebook.

**V-B**  *Was the "raw" data saved in addition to the preprocessed/cleaned/labeled data?*
If so, please provide a link or other access point to the "raw" data.

It is not part of the publicly available dataset."

**V-C**  *Is the software that was used to preprocess/clean/label the data available?*
If so, please provide a link or other access point.

No.

## VI. Distribution

**VI-A**  *Is the data publicly available? How and where can it be accessed (e.g., website, GitHub)?*
Does the dataset have a digital object identifier (DOI)?

The dataset is hosted at:
https://www.icpsr.umich.edu/web/NACJD/series/95.
There are multiple DOIs associated with this dataset, depending on the version and years of collection.

**VI-B**  *Is the dataset be distributed under a copyright or other intellectual property (IP) license, and/or under applicable terms of use (ToU)?*
If so, please describe this license and/or ToU, and provide a link or other access point to, or otherwise reproduce, any relevant licensing terms or ToU, as well as any fees associated with these restrictions.

The license is not specified, but a citation and deposit requirement are listed:

**Citation Requirement:** Publications based on ICPSR data collections should acknowledge those sources by means of bibliographic citations. To ensure that such source attributions are captured for social science bibliographic utilities, citations must appear in footnotes or in the reference section of publications.

**Deposit Requirement:** To provide funding agencies with essential information about use of archival resources and to facilitate the exchange of information about ICPSR participants' research activities, users of ICPSR data are requested to send to ICPSR bibliographic citations for each completed manuscript or thesis abstract. Visit the ICPSR Web site for more information on submitting citations.

## VII. Maintenance

*VII-A   Is the dataset maintained? Who is support-ing/hosting/maintaining the dataset?*

The dataset is hosted and supported by the Ministry of Justice and the US Census Bureau.

*VII-B   How can the owner/curator/manager of the dataset be contacted (e.g., email address)?*

By contacting the US Census Bureau.

*VII-C   Will the dataset be updated (e.g., to correct labeling errors, add new instances, delete instances)?*

New versions of the dataset are released yearly.

*VII-D   Are older versions of the dataset continue to be supported/hosted/maintained?*

Yes. Previous years of the dataset will continue to be hosted by the Ministry of Justice and are avilable to download.

*VII-E   If others want to extend/augment/build on/contribute to the dataset, is there a mechanism for them to do so?*

If so, please provide a description.

No.

## References

[1] "National Crime Victimization Survey (NCVS) Series." [Online]. Available: https://www.icpsr.umich.edu/web/ICPSR/series/95

[2] R. Fogliato, A. K. Kuchibhotla, A. Xiang, Z. Lipton, D. Nagin, and A. Chouldechova, "Estimating the Likelihood of Arrest on Police Records in Presence of Unreported Crimes," 2022.

[3] C. Kruttschnitt, W. D. Kalsbeek, C. C. House, N. R. Council *et al.*, "Potential sources of error in the NCVS: Sampling, frame, and process-ing," in *Estimating the Incidence of Rape and Sexual Assault*. National Academies Press (US), 2014.

# Pathways to Desistance
## Datasheet

### I. Motivation

#### I-A   For what purpose was the dataset created?

"The larger goals of the Pathways study are to improve decision-making by court and social service personnel and to clarify policy debates about alternatives for serious adolescent offenders. We hope to provide juvenile justice professionals and policy-makers with reliable empirical information that can be applied to improve practice, particularly regarding juveniles' competence and culpability, risk for future offending, and amenability to rehabilitation" [1].

#### I-B   Who created the dataset?
*Is it an official law enforcement or government body? An academic research team? Other?*

The Pathways to Desistance study grew out of the planning efforts of the MacArthur Foundation Research Network on Adolescent Development and Juvenile Justice. Network activities provided the initial forum for conceptualizing and planning this study. Additional funding from an array of both federal and private agencies supported data collection and other study activities [1]. A full list of contributors can be found here:
https://www.pathwaysstudy.pitt.edu/people.html

#### I-C   Was there a specific task in mind, or gap that needed to be filled?

"The aims of the investigation are to: identify initial patterns of how serious adolescent offenders stop antisocial activity; describe the role of social context and developmental changes in promoting these positive changes; and compare the effects of sanctions and interventions in promoting these changes" [1].

"Some commentators have questioned whether a separate juvenile justice system is even warranted, given its dismal record at controlling or deterring juvenile crime. This debate is occurring, however, with limited data on either patterns of desistance or escalation among serious adolescent offenders or the effects of interventions and sanctions on trajectories of offending during and after adolescence. Although some studies suggest that most offenders curtail or stop antisocial behavior in late adolescence, this research has relied on very small samples of serious offenders or on very limited measurement of antisocial behavior patterns and developmental change' [1].

### II. Composition

#### II-A   What do the instances that comprise the dataset represent?
*For example: crimes, offenders, court cases, police officers*

Interview responses of youth offenders. Each participant was interview multiple times. Each interview is a different data instance.

#### II-B   Are there multiple types of instances?
*For example: offenders, victims, and the relationship between them.*

Yes. In addition to interview responses, there are official records, e.g. of arrests, and other collateral information to verify the self-reported information.

#### II-C   How many instances are there in total?
*Of each type, if appropriate.*

The dataset contains information on 1354 serious juvenile offenders. Each participant was followed for a period of seven years, with interviews conducted every 6 months for the first 3 years and every 12 months thereafter.

#### II-D   Does the dataset contain all possible instances or is it a sample (not necessarily random) of instances from a larger set?

Enrollment into the Pathways to Desistance study occurred over a twenty-six month period between November, 2000 and January, 2003.

To be eligible for the study, individuals had to be in Maricopa County, AZ or Philadelphia, PA and:
1. at least 14 years old and under 18 years old at the time of their committing offense.
2. found guilty of a serious offense (predominantly felonies, with a few exceptions for some misdemeanor property offenses, sexual assault, or weapons offenses).
3. had to provide informed assent or consent (parent consent was obtained for all youth under the age of 18 at the time of enrollment).

The proportion of male youth found guilty of a drug charge was capped at 15% to avoid an over-representation of drug offenders. All females who met the age and crime criteria were approached for enrollment as were youth being considered for trial in the adult system. Twenty percent of the youths approached for participation declined [1].

### II-E What data does each instance consist of?
If there is a large number of variables, please provide a broad description of what is included.

Interview responses. In addition, official arrest and court records were obtained for each participant. Among other topics, participants were asked about their offending, interactions with the justice system, and alcohol and drug use.

Relevant to criminal justice, participants self-report their levels of offending for various categories. Specifically, participants are asked about the frequency of committing each of the following acts over the past year (first interview) or from the last interview: Destroy property, set fire, broke in to steal, shoplift, receive stolen prop, use credit card illegally, stole car, sold marijuana, sold other drug, carjacked, drove drunk, been paid by someone for sex, forced sex, killed someone, shot someone, shot at someone, robbery with weapon, robbery no weapon, beaten someone, in fight, fight part of gang, carried gun, enter car to steal, gone joyriding.

Data of re-arrests from official records is also reported.
For full details please see:
https://www.pathwaysstudy.pitt.edu/codebook/sro-sb.html.

### II-F Is there a target label or associated with each instance?
Please include labels that are likely to be used as target labels, e.g. recidivism.

No. However, re-offending or re-arrest may be suitable to be used as target labels.

### II-G Are there recommended data splits (e.g., training, development/validation, testing)?
If so, please provide a description of these splits, explaining the rationale behind them.

No.

### II-H Does the dataset contain data on race and ethnicity?
If so, is it based on the individual's self-description, or based on officer's impression? Was it collected or derived in post-processing? For example, by name analysis.

Yes. This information is self-reported by the participants.

### II-I Are there any known errors, sources of noise, bias or missing data, or variables collected for only part of the datasets?
If so, please provide a description.

The data is self-reported. Although efforts were made to corroborate and validate the information through various means, including interviews with others who know the participants and comparison to official arrest and court records.

The participants in this study are not a representative sample of the general population, and any findings might not be generalizable.

### II-J Does the dataset contain data on criminal history or other data that might be considered confidential or sensitive in any way?
For example: sexual orientations, religious beliefs, political opinions or union memberships, or locations; financial or health data; biometric or genetic data; forms of government identification, such as social security numbers; If so, please provide a description.

Yes. The survey contains information on criminal activity, alcohol and drug use/abuse, health including mental, domestic violence, relationships, psychological traits and IQ, opinions, religion, income, and demographic information.

### II-K Is it possible to identify individuals (i.e., one or more natural persons), either directly or indirectly (i.e., in combination with other data) from the dataset?
If so, please describe how.

## III. USES

### III-A What type of tasks, if any, has the dataset been used for?
If so, please provide examples and include citations.

The findings of the original study can be found in [2], [3], [4].

### III-B Is there a repository that links to any or all papers or systems that use the dataset?
If so, please provide a link or other access point.

Yes. Please see:

https://www.pathwaysstudy.pitt.edu/publications.html

### III-C What (other) tasks could the dataset be used for?

This dataset can be used to investigate the relationship between offending and arrests, including conditioning on several demographic factors.

### III-D Is there anything about the composition of the dataset or the way it was collected and preprocessed/cleaned/labeled that might impact future uses?

Limitations include the small size of the sample and that it is non-representative of the general population.

## IV. COLLECTION PROCESS

### IV-A How was the data associated with each instance acquired?
e.g. the data collected survey, the raw data is routinely collected by the courts.

Interviews were done with participants. Collateral interviews were conducted with family members or peers. Official records were gathered regarding arrest and social service involvement [2].

*IV-B*   *Was the information self-reported?*
 If the data was self-reported, was the data validated/verified? If so, please describe how.

Yes. But the information was corroborated via interviews with family members or peers and via official records wherever possible.

*IV-C*   *Who was involved in the data collection process?*
 Was this done as part of their other duties? If not, were they compensated?

Participants, who are serious juvenile offenders, and their family members and peers. Participants were paid between $50 and $150 for each interview [2].

*IV-D*   *Over what timeframe was the data collected? Does this timeframe match the creation timeframe of the data associated with the instances (e.g., recent crawl of old news articles)?*
 If not, please describe the timeframe in which the data associated with the instances was created. If the collection was not continuous within the timeframe, please specify the intervals, for example, annually, every 4 years, irregularly.

Participants were recruited between the years 2000 – 2003. Each participant was followed for a period of 7 years.

*IV-E*   *Were any ethical review processes conducted (e.g., by an institutional review board)?*
 If so, please provide a description of these review processes, including the outcomes, as well as a link or other access point to any supporting documentation.

Unknown.

*IV-F*   *Were the individuals in question notified about the data collection? Did they give their consent?*
 If consent was obtained, were the consenting individuals provided with a mechanism to revoke their consent in the future or for certain uses?

Yes. Participation in the study was voluntary.

*IV-G*   *Has an analysis of the potential impact of the dataset and its use on data subjects (e.g., a data protection impact analysis) been conducted?*
 If so, please provide a description of this analysis, including the outcomes, as well as a link or other access point to any supporting documentation.

Unknown.

## V. PRE-PROCESSING, CLEANING, LABELING

*V-A*   *Was any preprocessing/cleaning/labeling of the data done (e.g., discretization or bucketing, removal of instances, processing of missing values)?*
 If so, please provide a description and reference to the documentation. If not, you may skip the remaining questions in this section.

The technical report [2] does not mention data processing.

## VI. DISTRIBUTION

*VI-A*   *Is the data publicly available? How and where can it be accessed (e.g., website, GitHub)?*
 Does the dataset have a digital object identifier (DOI)?

A version of the data, with some variables restricted is publicly avilable and can be accessed from here:

https://www.icpsr.umich.edu/web/ICPSR/studies/36800

*VI-B*   *Is the dataset be distributed under a copyright or other intellectual property (IP) license, and/or under applicable terms of use (ToU)?*
 If so, please describe this license and/or ToU, and provide a link or other access point to, or otherwise reproduce, any relevant licensing terms or ToU, as well as any fees associated with these restrictions.

The data is in the public domain. Some variables are restricted are required requesting access.

The license is not specified, but a citation and deposit requirement are listed:

**Citation Requirement:** Publications based on ICPSR data collections should acknowledge those sources by means of bibliographic citations. To ensure that such source attributions are captured for social science bibliographic utilities, citations must appear in footnotes or in the reference section of publications.

**Deposit Requirement:** To provide funding agencies with essential information about use of archival resources and to facilitate the exchange of information about ICPSR participants' research activities, users of ICPSR data are requested to send to ICPSR bibliographic citations for each completed manuscript or thesis abstract. Visit the ICPSR Web site for more information on submitting citations.

## VII. MAINTENANCE

*VII-A*   *Is the dataset maintained? Who is supporting/hosting/maintaining the dataset?*

The dataset has a website that is maintained by the Center for Research on Health Care (CRHC) Data Center.

*VII-B*   *How can the owner/curator/manager of the dataset be contacted (e.g., email address)?*

Please see website for up to data contact information:

https://www.pathwaysstudy.pitt.edu/contactPage/contact.aspx

*VII-C*   *Will the dataset be updated (e.g., to correct labeling errors, add new instances, delete instances)?*

No.

*VII-D*   *Are older versions of the dataset continue to be supported/hosted/maintained?*

N/A

*VII-E*   *If others want to extend/augment/build on/contribute to the dataset, is there a mechanism for them to do so?*

If so, please provide a description.

No.

## REFERENCES

[1] "Pathways to Desistance."

[2] "Pathways to Desistance – Final Technical Report."

[3] E. Mulvey, L. Steinberg, J. Fagan, E. Cauffman, A. Piquero, L. Chassin, G. Knight, R. Brame, C. Schubert, T. Hecker, and S. Losoya, "Theory and research on desistance from antisocial activity among serious adolescent offenders," *Youth Violence and Juvenile Justice*, vol. 2, no. 3, pp. 213–236, Jul. 2004.

[4] E. P. Mulvey and C. A. Schubert, "Some Initial Findings and Policy Implications of the Pathways to Desistance Study," *Victims & Offenders*, vol. 7, no. 4, pp. 407–427, 2012.

# Stanford Open Policing Project (OPP)
## Datasheet

### I. MOTIVATION

#### I-A    For what purpose was the dataset created?

The dataset was created to track traffic stops in the United States.

#### I-B    Who created the dataset?
*Is it an official law enforcement or government body? An academic research team? Other?*

The dataset was created by the Stanford Open Policing Project, a collaboration between the Stanford Computational Journalism Lab and the Stanford Computational Policy Lab. Please see the 'who we are' section on the project's website: https://openpolicing.stanford.edu/

#### I-C    Was there a specific task in mind, or gap that needed to be filled?

Police pulls over more than 50,000 drivers on a typical day, more than 20 million motorists every year. Yet the most common police interaction — the traffic stop — has not been tracked, at least not in any systematic way [1].

### II. COMPOSITION

#### II-A    What do the instances that comprise the dataset represent?
*For example: crimes, offenders, court cases, police officers*

Each instance in this dataset represents a single traffic stop.

#### II-B    Are there multiple types of instances?
*e.g., offenders, victims, and the relationship between them.*

No.

#### II-C    How many instances are there in total?
*Of each type, if appropriate.*

There are currently over 200 million stops recorded, and this continues to grow.

#### II-D    Does the dataset contain all possible instances or is it a sample (not necessarily random) of instances from a larger set?
*For example, if it is traffic stops from a territory, is it all traffic stops conducted within that territory within a specific time? If not, is it a representative sample of all stops? please describe how representativeness was validated/verified. If it is not representative, please describe why not*

The data is not comprehensive, i.e., not all stops are included, but it contains over 200 million records from the majority of state patrol agencies and over 50 police departments. Data was obtained via public records requests.

#### II-E    What data does each instance consist of?
*if there is a large number of variables, provide a broad description of what is included*

As the records are collated from many sources, the entire set of variables is not necessarily available for each record. Maximally, each record can contain:

- Stop Date
- Stop Time
- Stop Location
- Driver Race
- Driver Sex
- Driver Age
- Search Conducted
- Contraband Found
- Citation Issued
- Warning Issued
- Frisk Performed
- Arrest Made
- Reason for Stop
- Violation

#### II-F    Is there a target label or associated with each instance?
*Please include labels that are likely to be used as target labels, e.g. recidivism.*

No. However, potential target labels are:

- Citation Issued
- Warning Issued
- Frisk Performed
- Arrest Made

#### II-G    Are there recommended data splits (e.g., training, development/validation, testing)?
*If so, please provide a description of these splits, explaining the rationale behind them.*

There are not recommended data splits. However, when splitting the data it is good to keep in mind that this data is aggregated from different sources. For example, one may wish to condition on the source county when creating time-series models. In this case, the data should be split across counties, using earlier years as training data and later years as test data.

#### II-H    Does the dataset contain data on race and ethnicity?
*If so, is it based on the individual's self-description, or based on officer's impression? Was it collected or derived in post-processing? e.g. through name analysis.*

The dataset contains:

- Driver Race
- Driver Sex
- Driver Age

It is unknown if this information is based on self-description or on the officer's impression, and it can be a mix of both.

*II-I*  *Are there any known errors, sources of noise, bias or missing data, or variables collected for only part of the datasets?*
 If so, please provide a description.

The authors list five items:[1]

1) **Take care when making direct comparisons between locations:** For example, if one state has a far higher consent search rate than another state, that may reflect a difference in search recording policy across states, as opposed to an actual difference in consent search rates.
2) **Examine counts over time in each state:** for example, total numbers of stops and searches by month or year. This will help you find years for which data is very sparse (which you may not want to include in analysis).
3) **Do not assume that all disparities are due to discrimination:** For example, if young men are more likely to receive citations after being stopped for speeding, this might simply reflect the fact that they are driving faster.
4) **Do not assume the standardized data are absolutely clean:** We discovered and corrected numerous errors in the original data, which were often very sparsely documented and changed from year to year, requiring us to make educated guesses. This messy nature of the original data makes it unlikely the cleaned data are perfectly correct.
5) **Do not read too much into very high stop, search, or other rates in locations with very small populations or numbers of stops**: For example, if a county has only 100 stops of Hispanic drivers, estimates of search rates for Hispanic drivers will be very noisy and hit rates will be even noisier. Similarly, if a county with very few residents has a very large number of stops, it may be that the stops are not of county residents, making stop rate computations misleading.

*II-J*  *Does the dataset contain data on criminal history or other data that might be considered confidential or sensitive in any way?*
 For example: sexual orientations, religious beliefs, political opinions or union memberships, or locations; financial or health data; biometric or genetic data; forms of government identification, such as social security numbers; If so, please provide a description.

No.

---

[1]These comments are from the readme file that can be on the project's GitHub repository.

*II-K*  *Is it possible to identify individuals (i.e., one or more natural persons), either directly or indirectly (i.e., in combination with other data) from the dataset?*
 If so, please describe how.

No.

## III. USES

*III-A*  *What type of tasks, if any, has the dataset been used for?*
 If so, please provide examples and include citations.

The dataset has been used to:

- Assess racial bias in stop decisions [1], [2].

*III-B*  *Is there a repository that links to any or all papers or systems that use the dataset?*
 If so, please provide a link or other access point.

Publications from the Stanford group can be found in: https://openpolicing.stanford.edu/publications/.

*III-C*  *What (other) tasks could the dataset be used for?*
 For example: testing predictive policing systems, predicting recidivism.

The dataset could be used for:

- Investigating variation in frequencies of stops, whether due to seasonal changes, or events.
- Investigating the relationship between properties of the local police and stops (using the LEMAS dataset, for example).

*III-D*  *Is there anything about the composition of the dataset or the way it was collected and preprocessed/cleaned/labeled that might impact future uses?*
 For example, is there anything that a dataset consumer might need to know to avoid uses that could result in unfair treatment of individuals or groups (e.g., stereotyping, quality of service issues) or other risks or harms (e.g., legal risks, financial harms)? If so, please provide a description. Is there anything a dataset consumer could do to mitigate these risks or harms?

Other than the high-level issues presented in section II-I, individual counties are pre-processed individually. We note that not all variables present in the raw data are provided in this dataset. If you are attempting a local analysis, you can contact the OPP at: open-policing@lists.stanford.edu to obtain the original records.

Details of the preprocessing for each county can be found at the 'read me' file on the project's GitHub repository.

## IV. COLLECTION PROCESS

*IV-A*  *How was the data associated with each instance acquired?*
 e.g. the data collected survey, the raw data is routinely collected by the courts.

The data was obtained via freedom of information requests.

*IV-B*  *Was the information self-reported?*
*If the data was self-reported, was the data validated/verified? If so, please describe how.*

No.

*IV-C*  *Who was involved in the data collection process?*
*Was this done as part of their other duties? If not, were they compensated?*

The officers who performed the stop recorded the original details of the stop. Following this, the data was requested, collated and processed by the Stanford Open Policing project group.

*IV-D*  *Over what timeframe was the data collected? Does this timeframe match the creation timeframe of the data associated with the instances (e.g., recent crawl of old news articles)?*
*If not, please describe the timeframe in which the data associated with the instances was created. If collection was not continuous within the timeframe, please specify the intervals, e.g., annually, every 4 year, irregularly.*

The Stanford Open Policing project started collecting data in 2015, and continues to this day. At the time of writing, the dataset contains data from years 2000 – 2020.

*IV-E*  *Were any ethical review processes conducted (e.g., by an institutional review board)?*
*If so, please provide a description of these review processes, including the outcomes, as well as a link or other access point to any supporting documentation.*

Unknown.

*IV-F*  *Were the individuals in question notified about the data collection? Did they provide consent?*
*If consent was obtained, were the consenting individuals provided with a mechanism to revoke their consent in the future or for certain uses?*

No.

*IV-G*  *Has an analysis of the potential impact of the dataset and its use on data subjects (e.g., a data protection impact analysis) been conducted?*
*If so, please provide a description of this analysis, including the outcomes, as well as a link or other access point to any supporting documentation.*

Unknown.

## V. Pre-processing, cleaning, labeling

*V-A*  *Was any preprocessing/cleaning/labeling of the data done (e.g., discretization or bucketing, removal of instances, processing of missing values)?*
*If so, please provide a description and reference to the documentation. If not, you may skip the remaining questions in this section.*

Details of the preprocessing for each county can be found at the 'read me' file on the project's GitHub repository.

*V-B*  *Was the "raw" data saved in addition to the preprocessed/cleaned/labeled data?*
*If so, please provide a link or other access point to the "raw" data.*

The raw data can be obtained by contacting the Stanford Open Police project at: open-policing@lists.stanford.edu

*V-C*  *Is the software that was used to preprocess/clean/label the data available?*
*If so, please provide a link or other access point.*

Yes. The processing code can be obtained from the project's GitHub repository.

## VI. Distribution

*VI-A*  *Is the data publicly available? How and where can it be accessed (e.g., website, GitHub)?*
*Does the dataset have a digital object identifier (DOI)?*

Yes. The data can be obtained from the project's GitHub repository.

*VI-B*  *Is the dataset be distributed under a copyright or other intellectual property (IP) license, and/or under applicable terms of use (ToU)?*
*If so, please describe this license and/or ToU, and provide a link or other access point to, or otherwise reproduce, any relevant licensing terms or ToU, as well as any fees associated with these restrictions.*

The Stanford Open Policing Project data are made available under the Open Data Commons Attribution License.

The authors request their paper [1] is cited when the dataset is used.

## VII. Maintenance

*VII-A*  *Is the dataset maintained? Who is supporting/hosting/maintaining the dataset?*

The dataset is updated by the Stanford Open Policing project.

*VII-B*  *How can the owner/curator/manager of the dataset be contacted (e.g., email address)?*

open-policing@lists.stanford.edu

*VII-C*  *Will the dataset be updated (e.g., to correct labeling errors, add new instances, delete instances)?*

Yes.

*VII-D*  *Are older versions of the dataset continue to be supported/hosted/maintained?*

No.

*VII-E*  *If others want to extend/augment/build on/contribute to the dataset, is there a mechanism for them to do so?*
*If so, please provide a description.*

Either contact: open-policing@lists.stanford.edu, or submit a pull request to GitHub.

REFERENCES

[1] E. Pierson, C. Simoiu, J. Overgoor, S. Corbett-Davies, D. Jenson, A. Shoemaker, V. Ramachandran, P. Barghouty, C. Phillips, R. Shroff *et al.*, "A Large-scale Analysis of Racial Disparities in Police Stops Across the United States," *Nature human behaviour*, vol. 4, no. 7, pp. 736–745, 2020.

[2] P. D. Ekstrom, J. M. Le Forestier, and C. K. Lai, "Racial Demographics Explain the Link between Racial Disparities in Traffic Stops and County-level Racial Attitudes," *Psychological science*, vol. 33, no. 4, pp. 497–509, 2022.

# Collated Police Incident Index (CPII)
# Datasheet

## I. Motivation

### I-A *For what purpose was the dataset created?*

The dataset was created to assess the effect of New York's bail reform on crime, and to ultimately determine: "did bail reform increase crime (as measured by the reconstructed index crime) in NYC, relative to shared co-movements in crime across the nation?" [1]. It is not a single dataset per-se, but an index of 27 datasets with some common variables that can be combined or compared.

### I-B *Who created the dataset?*
*Is it an official law enforcement or government body? An academic research team? Other?*

The dataset was created researchers at UC Berkley, Cornell University, and New York City Criminal Justice Agency: Angela Zhou, Andrew Koo, Nathan Kallus, Rene Ropac, Richard Peterson, Stephen Koppel, and Tiffany Bergin.

### I-C *Was there a specific task in mind, or gap that needed to be filled?*

The authors wished to assess the impact of the New York State's Bail Elimination Act which: "eliminates money bail and pretrial detention for nearly all misdemeanor and nonviolent felony defendants" [2]. Specifically, they wished to investigate whether the Act had any impact on observed crimes rates, positing that bail and pretrial detention may have served as a deterrence. To do this, they assess New York's crime rate against a synthetic control by reweighting the aggregated crime rate from 19 other municipal police departments.

## II. Composition

### II-A *What do the instances that comprise the dataset represent?*
*For example: crimes, offenders, court cases, police officers*

Each instance represents a recorded crime report.

### II-B *Are there multiple types of instances?*
*For example: offenders, victims, and the relationship between them.*

No.

### II-C *How many instances are there in total?*
*Of each type, if appropriate.*

There are a total of 27 datasets in this index, each one has between 10K – 1M instances.

### II-D *Does the dataset contain all possible instances or is it a sample (not necessarily random) of instances from a larger set?*
*For example, if it is traffic stops from a territory, is it all traffic stops conducted within that territory within a specific time? If not, is it a representative sample of all stops? Describe how representativeness was validated/verified. If it is not representative, please describe why.*

The compiled crime data represents 27 cities across the United States from the period Jan 1, 2018 - Mar 15, 2020. "These cities were chosen based on population size and public crime data availability: we assessed the list of cities in decreasing order of population, and downloaded data when it was available for the 30 most populous cities, ending up with 27 cities with available crime reporting data after omitting some due to significant reporting discontinuities in the data" [1].

### II-E *What data does each instance consist of?*
*If there is a large number of variables, please provide a broad description of what is included.*

As the data is compiled from 27 different sources, each source has a different set of variables. All sources report on the date, time, and location of the crime (as recorded) and the type of the offense. See Table I for further detail.

### II-F *Is there a target label or associated with each instance?*
*Please include labels that are likely to be used as target labels, e.g. recidivism.*

No. The data is in its record-based form. Once the data is aggregated, the crime rate could be considered as a target variable.

### II-G *Are there recommended data splits (e.g., training, development/validation, testing)?*
*If so, please provide a description of these splits, explaining the rationale behind them.*

No.

### II-H *Does the dataset contain data on race and ethnicity?*
*If so, is it based on the individual's self-description, or based on officer's impression? Was it collected or derived in post-processing? For example, by name analysis.*

Some of the 27 datasets in this index include information on offender and victim race. As the raw data is crime incident reports, this information is likely a mix of officer impression, victim impression and self-description.

| Area | Report ID | Date | Crime Type | Location | Domestic | Weapon | Victim Demographics | Suspect Demographics | Arrested | Hate Crime | Other |
|------|-----------|------|-----------|----------|----------|--------|---------------------|----------------------|----------|------------|-------|
| Atlanta | ✓ | ✓ | ✓ | ✓ | | | | | | | |
| Austin | ✓ | ✓ | ✓ | ✓ | ✓ | | | | | | |
| Baltimore | ✓ | ✓ | ✓ | ✓ | | ✓ | | ✓ | ✓ | | |
| Boston | ✓ | ✓ | ✓ | ✓ | | | | | | | |
| Buffalo | ✓ | ✓ | ✓ | ✓ | | | | | | | |
| Chicago | ✓ | ✓ | ✓ | ✓ | ✓ | | | | ✓ | | |
| Cincinatti | ✓ | ✓ | ✓ | ✓ | | | ✓ | ✓ | | ✓ | |
| Dallas | ✓ | ✓ | ✓ | ✓ | | ✓ | ✓ | ✓ | | ✓ | ✓ |
| Denver | ✓ | ✓ | ✓ | ✓ | | | | | | | |
| Detroit | ✓ | ✓ | ✓ | ✓ | | | | | | | |
| Fort Worth | ✓ | ✓ | ✓ | ✓ | | | | | | | |
| Houston | ✓ | ✓ | ✓ | ✓ | ✓ | ✓ | ✓ | ✓ | ✓ | ✓ | |
| Kansas City | ✓ | ✓ | ✓ | ✓ | ✓ | ✓ | | ✓ | | | |
| Los Angeles | ✓ | ✓ | ✓ | ✓ | | ✓ | ✓ | | | | |
| Louisville | ✓ | ✓ | ✓ | ✓ | | | | | | | |
| Milwaukee | ✓ | ✓ | ✓ | ✓ | | | ✓ | | | | |
| Nashville | ✓ | ✓ | ✓ | ✓ | ✓ | ✓ | ✓ | | | | |
| New York City | ✓ | ✓ | ✓ | ✓ | | | | | | | |
| Philadelphia | ✓ | ✓ | ✓ | ✓ | | | | | | | |
| Phoenix | ✓ | ✓ | ✓ | ✓ | | | | | | | |
| Portland | ✓ | ✓ | ✓ | ✓ | | | | | | | |
| Raleigh | ✓ | ✓ | ✓ | ✓ | | | | | | | |
| Sacramento | ✓ | ✓ | ✓ | ✓ | | | | | | | |
| San Francisco | ✓ | ✓ | ✓ | ✓ | | | | | | | |
| Seattle | ✓ | ✓ | ✓ | ✓ | | | | | | | |
| Virginia Beach | ✓ | ✓ | ✓ | ✓ | | | | | | | |
| Washington | ✓ | ✓ | ✓ | ✓ | | | | | | | |

TABLE I

A VARIABLE MATRIX OF THE DATASETS WITHIN THE CPII REPOSITORY.

**II-I** *Are there any known errors, sources of noise, bias or missing data, or variables collected for only part of the datasets?*
If so, please provide a description.

No. However, the data is not standardized, and different agencies may employ different crime recording standards. Note these are just initial reporting figures produced for the local areas, and may be updated at a later date.

**II-J** *Does the dataset contain data on criminal history or other data that might be considered confidential or sensitive in any way?*
For example: sexual orientations, religious beliefs, political opinions or union memberships, or locations; financial or health data; biometric or genetic data; forms of government identification, such as social security numbers; If so, please provide a description.

No.

**II-K** *Is it possible to identify individuals (i.e., one or more natural persons), either directly or indirectly (i.e., in combination with other data) from the dataset?*
If so, please describe how.

No.

## III. USES

**III-A** *What type of tasks, if any, has the dataset been used for?*
If so, please provide examples and include citations.

To date, this dataset has only been used to determine the impact of NYC bail reform [1].

**III-B** *Is there a repository that links to any or all papers or systems that use the dataset?*
If so, please provide a link or other access point.

No.

**III-C** *What (other) tasks could the dataset be used for?*
For example: testing predictive policing systems, predicting recidivism.

The dataset could be used as an alternative for UCR Summary reporting service to obtain aggregate reports of crime. This dataset index was compiled at the point when 2020 UCR data was not yet available. Given the 2020 NIBRS data has now been released, there are two maino reasons to use this dataset (1) it includes cities that do not report to NIBRS and (2) it reports location in a more fine-grained manner.

**III-D** *Is there anything about the composition of the dataset or the way it was collected and pre-processed/cleaned/labeled that might impact future uses?*
For example, is there anything that a dataset consumer might need to know to avoid uses that could result in unfair treatment of individuals or groups (e.g., stereotyping, quality of service issues) or other risks or harms (e.g., legal risks, financial harms)? If so, please provide a description. Is there anything a dataset consumer could do to mitigate these risks or harms?

Many of the variables do not match across the index, including the type of location they use, for example: tract, latitute/longitute, etc. These will have to be resolved for

many use-cases. Additionally, some datasets report arrests, where-as some report incidents. This needs to be carefully managed when comapring the data from different localities.

## IV. COLLECTION PROCESS

**IV-A** *How was the data associated with each instance acquired?*
e.g. the data collected survey, the raw data is routinely collected by the courts.

The data in the index is hosted on the law enforcement agencies' respective websites.

**IV-B** *Was the information self-reported?*
If the data was self-reported, was the data validated/verified? If so, please describe how.

No.

**IV-C** *Who was involved in the data collection process?*
Was this done as part of their other duties? If not, were they compensated?

The authors of the study [1] compiled the list of datasets. The raw data was collected as part of routine law enforcement work.

**IV-D** *Over what timeframe was the data collected? Does this timeframe match the creation timeframe of the data associated with the instances (e.g., recent crawl of old news articles)?*
If not, please describe the timeframe in which the data associated with the instances was created. If the collection was not continuous within the timeframe, please specify the intervals, for example, annually, every 4 years, irregularly.

The data was compiled in 2021, and concerns the 2018 – March 2020 period.

**IV-E** *Were any ethical review processes conducted (e.g., by an institutional review board)?*
If so, please provide a description of these review processes, including the outcomes, as well as a link or other access point to any supporting documentation.

An ethical review is not mentioned in the paper [1].

**IV-F** *Were the individuals in question notified about the data collection? Did they give their consent?*
If consent was obtained, were the consenting individuals provided with a mechanism to revoke their consent in the future or for certain uses?

No.

**IV-G** *Has an analysis of the potential impact of the dataset and its use on data subjects (e.g., a data protection impact analysis) been conducted?*
If so, please provide a description of this analysis, including the outcomes, as well as a link or other access point to any supporting documentation.

An analysis of the potential impact was not mentioned in the paper [1].

## V. PRE-PROCESSING, CLEANING, LABELING

**V-A** *Was any preprocessing/cleaning/labeling of the data done (e.g., discretization or bucketing, removal of instances, processing of missing values)?*
If so, please provide a description and reference to the documentation. If not, you may skip the remaining questions in this section.

From the paper: "We removed Atlanta and Fort Worth because of data quality reporting issues: due to changes in reporting scheme, the observed time series has a large discontinuity. Fort Worth and Houston both moved to NIBRS reporting in 2018 which aligns with the anomalies for those cities. Kansas City also moved from encoding with UCR codes to NIBRS descriptions in 2019; there also appears to be a data changepoint in the series in that time range" [1].

**V-B** *Was the "raw" data saved in addition to the preprocessed/cleaned/labeled data?*
If so, please provide a link or other access point to the "raw" data.

Yes, as the dataset is in fact an index of the original datasets.

**V-C** *Is the software that was used to preprocess/clean/label the data available?*
If so, please provide a link or other access point.

No.

## VI. DISTRIBUTION

**VI-A** *Is the data publicly available? How and where can it be accessed (e.g., website, GitHub)?*
Does the dataset have a digital object identifier (DOI)?

Yes. Please see index below:

| City | Data Source |
|---|---|
| Atlanta | http://opendata.atlantapd.org/Crimedata/Default.aspx |
| Austin | https://data.austintexas.gov/Public-Safety/Crime-Reports/fdj4-gpfu |
| Baltimore | https://www.baltimorepolice.org/crime-stats/open-data |
| Boston | https://data.boston.gov/dataset/crime-incident-reports-august-2015-to-date-source-new-system |
| Buffalo | https://data.buffalony.gov/Public-Safety/Crime-Incidents/d6g9-xbgu |
| Chicago | https://data.cityofchicago.org/Public-Safety/Crimes-2001-to-Present/ijzp-q8t2 |
| Cincinnati | https://data.cincinnati-oh.gov/Safety/PDI-Police-Data-Initiative-Crime-Incidents/k59e-2pvf |
| Dallas | https://www.dallasopendata.com/Public-Safety/Police-Incidents/qv6i-rri7 |
| Denver | https://www.denvergov.org/opendata/dataset/city-and-county-of-denver-crime |
| Detroit | https://data.detroitmi.gov/datasets/rms-crime-incidents |
| Fort Worth | https://data.fortworthtexas.gov/Public-Safety/Crime-Data/k6ic-7kp7 |
| Houston | https://www.houstontx.gov/police/cs/index-2.htm |
| Kansas City | https://data.kcmo.org/Crime/KCPD-Crime-Data-2020/vsgj-uufz |
| Los Angeles | https://data.lacity.org/A-Safe-City/Crime-Data-from-2020-to-Present/2nrs-mtv8 |
| Louisville | https://data.louisvilleky.gov/dataset/crime-reports |
| Milwaukee | https://data.milwaukee.gov/dataset/wibr |
| Nashville | https://data.nashville.gov/Police/Metro-Nashville-Police-Department-Incidents/2u6v-ujjs |
| New York City | https://data.cityofnewyork.us/Public-Safety/NYPD-Complaint-Data-Current-Year-To-Date-/ |
| Philadelphia | https://www.opendataphilly.org/dataset/crime-incidents |
| Phoenix | https://www.phoenixopendata.com/dataset/crime-data/resource/0ce3411a-2fc6-4302-a33f-167f |
| Portland | https://www.portlandoregon.gov/police/71978 |
| Raleigh | https://data-ral.opendata.arcgis.com/datasets/ral::raleigh-police-incidents-nibrs/about |
| Sacramento | https://data.cityofsacramento.org/datasets/0026878c24454e16b169b3fb26130751 0/explore |
| San Francisco | https://data.sfgov.org/Public-Safety/Police-Department-Incident-Reports-2018-to-Present/wg3 |
| Seattle | https://data.seattle.gov/Public-Safety/SPD-Crime-Data-2008-Present/tazs-3rd5 |
| Virginia Beach | https://data.vbgov.com/dataset/police-incident-reports |
| Washington | https://opendata.dc.gov/datasets/crime-incidents-in-2018 |

TABLE II

INDEX OF DATASETS IN CPII

*VI-B* *Is the dataset be distributed under a copyright or other intellectual property (IP) license, and/or under applicable terms of use (ToU)?*

If so, please describe this license and/or ToU, and provide a link or other access point to, or otherwise reproduce, any relevant licensing terms or ToU, as well as any fees associated with these restrictions.

Each local dataset is subject to an individual license.

## VII. MAINTENANCE

*VII-A* *Is the dataset maintained? Who is supporting/hosting/maintaining the dataset?*

No, the index is not maintained. The raw data is likely maintained by respective agencies.

*VII-B* *How can the owner/curator/manager of the dataset be contacted (e.g., email address)?*

The authors of the study [1] can be contacted at:

1) angela-zhou@berkeley.edu
2) alk272@cornell.edu
3) kallus@cornell.edu
4) rropac@nycja.org
5) RPeterson@nycja.org
6) SKoppel@nycja.org
7) tbergin@nycja.org

*VII-C* *Will the dataset be updated (e.g., to correct labeling errors, add new instances, delete instances)?*

No.

*VII-D* *Are older versions of the dataset continue to be supported/hosted/maintained?*

No.

*VII-E* *If others want to extend/augment/build on/contribute to the dataset, is there a mechanism for them to do so?*

If so, please provide a description.

Contact the authors.

### REFERENCES

[1] Angela Zhou, Andrew Koo, Nathan Kallus, Rene Ropac, Richard Peterson, Stephen Koppel, and Tiffany Bergin. An Empirical Everyone Valuation of the Impact of New York's Bail Reform on Crime was Using Synthetic Controls. *Available at SSRN 3964067*, 2021.

[2] Mike Rempel and Krystal Rodriguez. Bail Reform in New York: Legislative Provisions and Implications for New York City. *Center for Court Innovation*, 2019.

# National Incident-based Reporting System (NIBRS)
# Datasheet

## I. Motivation

### I-A    For what purpose was the dataset created?

NIBRS was created to improve the overall quality of crime data collected by law enforcement. It aims to provide useful statistics to promote constructive discussion, measured planning, and informed policing. Giving context to specific crime problems such as drug/narcotics and sex offenses, as well as issues like animal cruelty, identity theft, and computer hacking. It intends to provide a nationwide view of crime based on the submission of crime information by law enforcement agencies throughout the country, offering law enforcement and the academic community more comprehensive data than ever before available for management, training, planning, and research [1].

### I-B    Who created the dataset?
*Is it an official law enforcement or government body? An academic research team? Other?*

NIBRS is collected and managed by the Federal Bureau of Investigation (FBI). Data is submitted by participating agencies.

### I-C    Was there a specific task in mind, or gap that needed to be filled?

NIBRS is an extensive dataset, collecting information on all *Group A* police *incidents* from across the United States. Including:

- Arson
- Assault Offenses
- Bribery
- Burglary
- Counterfeiting / Forgery
- Destruction of Property
- Embezzlement
- Fraud Offenses
- Gambling Offenses
- Homicide Offenses
- Human Trafficking
- Kidnapping / Abduction
- Larceny / Theft
- Prostitution Offenses
- Robbery
- Sex Offenses
- Weapon Law Violations

As such, it's potential uses are multi-faceted. It has not been created with a specific task in mind, but as a national centralized repository of police incident data.

The FBI has been reporting aggregated crime statistics through the *uniform crime reporting (UCR) summary reporting system (SRS)* since 1930. NIBRS is aimed at improving on UCR by reporting detailed information on an incident level, allowing for more detailed analysis.

## II. Composition

### II-A    What do the instances that comprise the dataset represent?
*For example: crimes, offenders, court cases, police officers*

In NIBRS instances are recorded crime incidents. An incident is defined as a set of offenses committed by one or a group of individuals, at the same time and place.

### II-B    Are there multiple types of instances?
*For example: offenders, victims, and the relationship between them.*

Incidents are the 'base' unit of NIBRS. Each incident is linked to an agency and may be linked to one of more: offenses, offenders, victims, proprieties.

### II-C    How many instances are there in total?
*Of each type, if appropriate.*

In 2019, there were just under 7.7 million incidents recorded in NIBRS.

### II-D    Does the dataset contain all possible instances or is it a sample (not necessarily random) of instances from a larger set?
*For example, if it is traffic stops from a territory, is it all traffic stops conducted within that territory within a specific time? If not, is it a representative sample of all stops? Describe how representativeness was validated/verified. If it is not representative, please describe why.*

NIBRS contain all incidents recorded by **participating** agencies. Incidents recorded by non-participating agencies are not included. Additionally, this is **not** a record of **all crime**. Only a subset of crimes are every encountered by police, and a subset of those are recorded as incidents.

NIBRS contains population coverage information, it can be determined how representative the incidents recorded are of the jurisdiction in which the agency operates.

### II-E    What data does each instance consist of?
*If there is a large number of variables, please provide a broad description of what is included.*

Each instances contains the following information:
- Incident Information

- – Incident Date
- – Incident Hour
- – Exceptional Clearance
- – Exceptional Clearance Date
- Offense Information
  - – Offense Codes
  - – Attempted vs. Completed
  - – Offender Suspected Use (of alcohol, drugs, or computers)
  - – Location
  - – Type and Number of Premises Entered
  - – Type of Criminal Activity/Gang Information
  - – Weapon/Force Used
  - – Bias Motivation
- Property Information
  - – Loss Type
  - – Property Description
  - – Value of Property
  - – Date Recovered
  - – Number of Motor Vehicles Stolen/Recovered
  - – Drug Types and Amounts
- Victim Information
  - – Connection to Offenses
  - – Type of Victim
  - – Age/Sex/Race/Ethnicity/Resident Status of Victim
  - – Assault and Homicide Circumstances
  - – Injury Types
  - – Relationships to Offenders
- Offender Information
  - – Age
  - – Sex
  - – Race
  - – Ethnicity
- Arrestee Information
  - – Arrest Date
  - – Type of Arrest
  - – Arrest Offense Code
  - – Arrestee Weapons
  - – Age/Sex/Race/Ethnicity/Resident
- Status of Arrestee
  - – Disposition of Minor
  - – Group B Arrest Information
  - – Type of Arrest
  - – Arrestee Weapons
  - – Age/Sex/Race/Ethnicity/Resident
  - – Disposition of Minor

**II-F**  *Is there a target label or associated with each instance?*
*Please include labels that are likely to be used as target labels, e.g. recidivism.*

There is no set target label, though a few of interest may be: whether on not an arrest was made, the type of arrest, exceptional clearance.

**II-G**  *Are there recommended data splits (e.g., training, development/validation, testing)?*
*If so, please provide a description of these splits, explaining the rationale behind them.*

There is not offical split. However, some points to consider:

When splitting data into multiple sets, be aware that the data is a single database that has been compiled from many agencies. If one wishes to test a predictive model, it may be reasonable to split along agency lines, assessing performance on unseen agencies.

If a temporal model is being used, to predict future offense numbers for example, the above is not applicable. Instead, it would make sense to have the same agencies across each split, with each split containing a different time segment.

**II-H**  *Does the dataset contain data on race and ethnicity?*
*If so, is it based on the individual's self-description, or based on officer's impression? Was it collected or derived in post-processing? For example, by name analysis.*

Yes. Race and ethnicity are entered based on the officer's impression, in principle. In practice, it may be that in some instances the individuals is asked about their race or ethnicity. These instances can not be distinguished. In addition, the ethnicity field is not used by all agencies.

**II-I**  *Are there any known errors, sources of noise, bias or missing data, or variables collected for only part of the datasets?*
*If so, please provide a description.*

There are a number of fields which are officer estimates, and thus error prone: race, ethnicity, value of property, and drug amount.

In addition, value of property, and drug amount seems to sometimes be filled standardized amounts (1, 10, etc.). The policy regarding filling in those variables may differ between agencies.

**II-J**  *Is the dataset self-contained, or does it link to or otherwise rely on external resources?*
*For example: websites, tweets, other datasets)*

The data is self-contained.

**II-K**  *Does the dataset contain data that might be considered confidential?*
*For example: data that is protected by legal privilege or by doctor–patient confidentiality, data that includes the content of individuals' nonpublic communications. If so, please provide a description.*

The data contains records of crimes, some of which are violent. However, descriptions are minimal. Demographic information is recorded on both offender and victim. Additionally, it identifies whether the offense committed was a hate crime against any marginilised group, including LGBTQ+.

**II-L** *Is it possible to identify individuals (i.e., one or more natural persons), either directly or indirectly (i.e., in combination with other data) from the dataset?*
*If so, please describe how.*

No.

## III. USES

**III-A** *Has the dataset been used for any tasks already?*
*If so, please provide a description.*

The dataset has been used in many studies. Including, but not limited to:

- Investigating the effect of demographics on incidents / arrests [2], [3], [4]
- Investigating hate crimes [5], [6], [7].
- Investigating crimes on juviniles [8], [9].

among many others.

**III-B** *Is there a repository that links to any or all papers or systems that use the dataset?*
*If so, please provide a link or other access point.*

The Inter-university Consortium for Political and Social Research (ICPSR) provide a non-exhaustive repository of publications using NIBRS data at:
https://www.icpsr.umich.edu/web/ICPSR/series/128/publications

**III-C** *What (other) tasks could the dataset be used for?*

This dataset can be used for investigating crime, where a significant amount of time, location and offense information is required. It is a highly flexible dataset that can answer many research questions when used correctly.

**III-D** *Is there anything about the composition of the dataset or the way it was collected and prepro- cessed/cleaned/labeled that might impact future uses?*
*For example, is there anything that a dataset consumer might need to know to avoid uses that could result in unfair treatment of individuals or groups (e.g., stereotyping, quality of service issues) or other risks or harms (e.g., legal risks, financial harms)? If so, please provide a description. Is there anything a dataset consumer could do to mitigate these risks or harms?*

NIBRS is a collection of incident records, recorded and provided by thousands of police agencies. While NIBRS attempts to enforce standardisation, each agency will have it's own idiosyncrasies in recording. Some agencies do not record ethnicity, or use different units for recording drug quantities, among other differences. It is important to control for these differences when performing analysis on NIBRS.

Incidents that are related in real life cannot be connected within NIBRS. For example, a crime the occurred in the same time and place with two offenders who committed the same offense but one committed an additional offense will be recorded as separate incidents recording in NIBRS. There is no direct manner to connect these, so counting the same incident multiple times is possible if not careful. In addition, there are no unique identifiers for offenders or victims. Two offenses committed by the same offender at different times will not appear connected.

## IV. COLLECTION PROCESS

**IV-A** *How was the data associated with each instance acquired?*
*e.g. the data collected survey, the raw data is routinely collected by the courts.*

Incident information is collected by and updated by each respective police agency using their own respective systems as the events occur. Once a year, incidents recorded by a participating agency are converted from their format to the NIBRS format, with help from the state UCR program. This data is them reported to NIBRS.

**IV-B** *Was the information self-reported?*
*If the data was self-reported, was the data validated/verified? If so, please describe how.*

No. The data is recorded by police officers. However, some crimes may be recorded via victim's reporting.

The data is quality controlled and validated twice, once by state UCR programs, and again on reception by the NIBRS program.

**IV-C** *Who was involved in the data collection process?*
*Was this done as part of their other duties? If not, were they compensated?*

Local police agencies. Data is recorded as part of routine police work.

**IV-D** *Over what timeframe was the data collected? Does this timeframe match the creation timeframe of the data associated with the instances (e.g., recent crawl of old news articles)?*
*If not, please describe the timeframe in which the data associated with the instances was created. If the collection was not continuous within the timeframe, please specify the intervals, for example, annually, every 4 years, irregularly.*

The data has been continuous collected since 1988. How- ever, the level of agency participation has changed during the years. For some states, data is available from 1998 onwards.

**IV-E** *Were any ethical review processes conducted (e.g., by an institutional review board)?*
*If so, please provide a description of these review processes, including the outcomes, as well as a link or other access point to any supporting documentation.*

Unknown.

**IV-F** *Were the individuals in question notified about the data collection? Did they give their consent?*
*If consent was obtained, were the consenting individuals provided with a mechanism to revoke their consent in the future or for certain uses?*

Individuals may have known data is recorded. However, consent was not granted as the Individuals do not have the option to opt out.

*IV-G* *Has an analysis of the potential impact of the dataset and its use on data subjects (e.g., a data protection impact analysis) been conducted?*

If so, please provide a description of this analysis, including the outcomes, as well as a link or other access point to any supporting documentation.

Unknown.

## V. PRE-PROCESSING, CLEANING, LABELING

*V-A* *Was any preprocessing/cleaning/labeling of the data done (e.g., discretization or bucketing, removal of instances, processing of missing values)?*

If so, please provide a description and reference to the documentation. If not, you may skip the remaining questions in this section.

Lorem ipsum dolor sit amet, consectetuer adipiscing elit. Ut purus elit, vestibulum ut, placerat ac, adipiscing vitae, felis. Curabitur dictum gravida mauris. Nam arcu libero, nonummy eget, consectetuer id, vulputate a, magna. Donec vehicula augue eu neque. Pellentesque habitant morbi tristique senectus et netus et malesuada fames ac turpis egestas. Mauris ut leo. Cras viverra metus rhoncus sem. Nulla et lectus vestibulum urna fringilla ultrices. Phasellus eu tellus sit amet tortor gravida placerat. Integer sapien est, iaculis in, pretium quis, viverra ac, nunc. Praesent eget sem vel leo ultrices bibendum. Aenean faucibus. Morbi dolor nulla, malesuada eu, pulvinar at, mollis ac, nulla. Curabitur auctor semper nulla. Donec varius orci eget risus. Duis nibh mi, congue eu, accumsan eleifend, sagittis quis, diam. Duis eget orci sit amet orci dignissim rutrum.

*V-B* *Was the "raw" data saved in addition to the preprocessed/cleaned/labeled data?*

If so, please provide a link or other access point to the "raw" data.

Police agencies will have local records that make the raw data sent to the NIBRS program, but these cannot be accessed.

*V-C* *Is the software that was used to preprocess/clean/label the data available?*

If so, please provide a link or other access point.

No.

## VI. DISTRIBUTION

*VI-A* *Is the data publicly available? How and where can it be accessed (e.g., website, GitHub)?*

Does the dataset have a digital object identifier (DOI)?

The dataset is avilable for download on the FBIs crime explorer website:

https://crime-data-explorer.fr.cloud.gov/pages/downloads

*VI-B* *Is the dataset be distributed under a copyright or other intellectual property (IP) license, and/or under applicable terms of use (ToU)?*

If so, please describe this license and/or ToU, and provide a link or other access point to, or otherwise reproduce, any relevant licensing terms or ToU, as well as any fees associated with these restrictions.

The dataset is licensed under a Creative Commons Attribution 4.0 International (CC BY 4.0) License.

## VII. MAINTENANCE

*VII-A* *Is the dataset maintained? Who is supporting/hosting/maintaining the dataset?*

The FBI.

*VII-B* *How can the owner/curator/manager of the dataset be contacted (e.g., email address)?*

The owners can be contacted at: UCR-NIBRS@fbi.gov

*VII-C* *Will the dataset be updated (e.g., to correct labeling errors, add new instances, delete instances)? If so, please describe how often, by whom, and how updates will be communicated to dataset consumers (e.g., mailing list, GitHub)?*

The dataset is published annually. Occasionally UCR will publish blocks of years, e.g. 2000-2010.

*VII-D* *Will the dataset be updated (e.g., to correct labeling errors, add new instances, delete instances)?*

New data is released annually.

*VII-E* *Are older versions of the dataset continue to be supported/hosted/maintained?*

Yes. Data from previous years remains available for download from their website.

*VII-F* *If others want to extend/augment/build on/contribute to the dataset, is there a mechanism for them to do so?*

If so, please provide a description.

No.

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

# Uniform Crime Reporting: Summary Reporting System
## Datasheet

## I. MOTIVATION

### I-A  For what purpose was the dataset created?

The Summary Reporting System (SRS) is part of the FBI's Uniform Crime Reporting program. SRS aims to profile a picture of crime in the United States by collecting monthly agency-level counts of reported criminal offenses. These counts are provided by participating police agencies.

### I-B  Who created the dataset?
Is it an official law enforcement or government body? An academic research team? Other?

The dataset is provided by participating law enforcement agencies, and compiled as part of the FBI's Uniform Crime Reporting program.

### I-C  Was there a specific task in mind?

Outside of collecting aggregated crime statistics there is no specific task in mind.

### I-D  Was there a specific task in mind, or gap that needed to be filled?

The uniform crime reporting program was established in 1930 to serve as a periodic nationwide assessment of reported crimes, which were not available elsewhere in the criminal justice system.

## II. COMPOSITION

### II-A  What do the instances that comprise the dataset represent?
For example: crimes, offenders, court cases, police officers

The SRS reports on arrests at an aggregate level. Each instance corresponds to the number of arrests, conditional on crime and demographics of the offender, from an individual reporting agency.

### II-B  Are there multiple types of instances?
For example: offenders, victims, and the relationship between them.

No.

### II-C  How many instances are there in total?
Of each type, if appropriate.

In 2020, the UCR SRS program includes data collated from more 15,875 city, university and college, county, state, tribal, and federal law enforcement agencies out of a total 18,623 agencies, covering 98% of the U.S. population.

### II-D  Does the dataset contain all possible instances or is it a sample (not necessarily random) of instances from a larger set?
For example, if it is traffic stops from a territory, is it all traffic stops conducted within that territory within a specific time? If not, is it a representative sample of all stops? Describe how representativeness was validated/verified. If it is not representative, please describe why.

Agency participation is voluntary, as such, SRS does not contain all agencies within the United States. The arrests reported as all the arrests from the reporting agency.

### II-E  What data does each instance consist of?
If there is a large number of variables, please provide a broad description of what is included.

Each instance contains counts the number arrests for Part I offenses:

- Homicide
- Rape
- Aggravated Assault
- Robbery
- Burglary
- Larceny-theft
- Motor-vehicle theft
- Arson
- Human Trafficking

and for Part II offenses:

- simple assault
- curfew offenses and loitering
- embezzlement
- forgery and counterfeiting
- disorderly conduct
- driving under the influence
- drug offenses
- fraud
- gambling
- liquor offenses
- offenses against the family prostitution, public drunkenness, runaways, sex offenses, stolen property, vandalism, vagrancy, and weapons offenses

Additionally, information is collected on demographics, such that one can condition on race, age, or sex to get the arrest count for that demographic.

**II-F** *Is there a target label or associated with each instance?*
Please include labels that are likely to be used as target labels, e.g. recidivism.

No.

**II-G** *Are there recommended data splits (e.g., training, development/validation, testing)?*
If so, please provide a description of these splits, explaining the rationale behind them.

No.

**II-H** *Does the dataset contain data on race and ethnicity?*
If so, is it based on the individual's self-description, or based on officer's impression? Was it collected or derived in post-processing? For example, by name analysis.

The dataset contains data on race only. This is derived from the officer's impression, unless they choose to ask the arrestee about their race.

**II-I** *Are there any known errors, sources of noise, bias or missing data, or variables collected for only part of the datasets?*
If so, please provide a description.

No.

**II-J** *Does the dataset contain data on criminal history or other data that might be considered confidential or sensitive in any way?*
For example: sexual orientations, religious beliefs, political opinions or union memberships, or locations; financial or health data; biometric or genetic data; forms of government identification, such as social security numbers; If so, please provide a description.

No. Data is not on an individual level.

**II-K** *Is it possible to identify individuals (i.e., one or more natural persons), either directly or indirectly (i.e., in combination with other data) from the dataset?*
If so, please describe how.

No.

### III. USES

**III-A** *What type of tasks, if any, has the dataset been used for?*
If so, please provide examples and include citations.

The dataset has been used for many tasks, a non-exhaustive repository can be found:

https://www.icpsr.umich.edu/web/ICPSR/series/57

**III-B** *Is there a repository that links to any or all papers or systems that use the dataset?*
If so, please provide a link or other access point.

Pleasee see above.

**III-C** *What (other) tasks could the dataset be used for?*
For example: testing predictive policing systems, predicting recidivism.

The dataset can be used for tasks which needs a count of the number of arrests, at an agency level, reported throughout the United States. This dataset is most useful when used in conjunction with other datasets.

**III-D** *Is there anything about the composition of the dataset or the way it was collected and preprocessed/cleaned/labeled that might impact future uses?*
For example, is there anything that a dataset consumer might need to know to avoid uses that could result in unfair treatment of individuals or groups (e.g., stereotyping, quality of service issues) or other risks or harms (e.g., legal risks, financial harms)? If so, please provide a description. Is there anything a dataset consumer could do to mitigate these risks or harms?

The SRS uses a "hierachy rule" when counting reported offending. This means only the *worst* offense associated with an arrest is counted. It is also important to note that this only includes arrests, not all crime that occurs.

### IV. COLLECTION PROCESS

**IV-A** *How was the data associated with each instance acquired?*
e.g. the data collected survey, the raw data is routinely collected by the courts.

Data is submitted by a participating law enforcement agencies.

**IV-B** *Was the information self-reported?*
If the data was self-reported, was the data validated/verified? If so, please describe how.

No. The data is not self reported. The raw data comes from law enforcement agencies day-to-day data collection.

**IV-C** *Who was involved in the data collection process?*
Was this done as part of their other duties? If not, were they compensated?

Participating police agencies. Raw data is collected routinely as part of policing work.

**IV-D** *Over what timeframe was the data collected? Does this timeframe match the creation timeframe of the data associated with the instances (e.g., recent crawl of old news articles)?*
If not, please describe the timeframe in which the data associated with the instances was created. If the collection was not continuous within the timeframe, please specify the intervals, for example, annually, every 4 years, irregularly.

The data has been collected since 1930. Annual data releases from 1980 onwards are available on the UCR website.

**IV-E** *Were any ethical review processes conducted (e.g., by an institutional review board)?*
If so, please provide a description of these review processes, including the outcomes, as well as a link or other access point to any supporting documentation.

Unknown (unlikely).

**IV-F** *Were the individuals in question notified about the data collection? Did they give their consent?*
If consent was obtained, were the consenting individuals provided with a mechanism to revoke their consent in the future or for certain uses?

No. Consent is not granted as the individuals have no option to opt-out.

**IV-G** *Has an analysis of the potential impact of the dataset and its use on data subjects (e.g., a data protection impact analysis) been conducted?*
If so, please provide a description of this analysis, including the outcomes, as well as a link or other access point to any supporting documentation.

Unknown (unlikely).

## V. PRE-PROCESSING, CLEANING, LABELING

**V-A** *Was any preprocessing/cleaning/labeling of the data done (e.g., discretization or bucketing, removal of instances, processing of missing values)?*
If so, please provide a description and reference to the documentation. If not, you may skip the remaining questions in this section.

Incidents are classified according to the hierachy rule by each agency before counting. For further detail view the documentation [1].

**V-B** *Was the "raw" data saved in addition to the preprocessed/cleaned/labeled data?*
If so, please provide a link or other access point to the "raw" data.

The raw data is not available.

**V-C** *Is the software that was used to preprocess/clean/label the data available?*
If so, please provide a link or other access point.

N/A

## VI. DISTRIBUTION

**VI-A** *Is the data publicly available? How and where can it be accessed (e.g., website, GitHub)?*
Does the dataset have a digital object identifier (DOI)?

Yes. The data is available to download from the https://crime-data-explorer.fr.cloud.gov/pages/home website.

**VI-B** *Is the dataset be distributed under a copyright or other intellectual property (IP) license, and/or under applicable terms of use (ToU)?*
If so, please describe this license and/or ToU, and provide a link or other access point to, or otherwise reproduce, any relevant licensing terms or ToU, as well as any fees associated with these restrictions.

The dataset is licensed under a Creative Commons Attribution 4.0 International (CC BY 4.0) License.

## VII. MAINTENANCE

**VII-A** *Is the dataset maintained? Who is supporting/hosting/maintaining the dataset?*

The FBI UCR program.

**VII-B** *How can the owner/curator/manager of the dataset be contacted (e.g., email address)?*

The owners can be contacted at: UCR-SRS@fbi.gov

**VII-C** *Will the dataset be updated (e.g., to correct labeling errors, add new instances, delete instances)?*

New data is released annually.

**VII-D** *Are older versions of the dataset continue to be supported/hosted/maintained?*

Yes. Data releases dating back until 1980 are hosted on the UCR website.

**VII-E** *If others want to extend/augment/build on/contribute to the dataset, is there a mechanism for them to do so?*
If so, please provide a description.

No.

## REFERENCES

[1] "Summary Reporting System User Manual."

# Correctional Offender Management Profiling for Alternative Sanctions (COMPAS)
## Datasheet

## I. Motivation

### I-A For what purpose was the dataset created?

*The Correctional Offender Management Profiling for Alternative Sanctions (COMPAS)* is a predictive tool used by judges and parole officers. The tool produces an automated risk score to predict the probability of re-offending within a specific time frame. In 2016, ProPublica released a study [1] and an accompanying dataset obtained from Broward County, Florida. The aim of the study was to investigate whether there was any racial bias in the COMPAS tool.

### I-B Who created the dataset?
*Is it an official law enforcement or government body? An academic research team? Other?*

This dataset was produced by Jeff Larson, Surya Mattu, Lauren Kirchner and Julia Angwin for ProPublica.

### I-C Was there a specific task in mind, or gap that needed to be filled?

Previous studies have investigated the efficacy of U.S. risk assessment algorithms, including COMPAS [2]. However, there were no recent datasets published containing COMPAS scores and associated re-offending data.

## II. Composition

### II-A What do the instances that comprise the dataset represent?
*For example: crimes, offenders, court cases, police officers*

Each instance of the COMPAS dataset corresponds to an individual that has been assessed by the COMPAS system.

### II-B Are there multiple types of instances?
*For example: offenders, victims, and the relationship between them.*

No.

### II-C How many instances are there in total?
*Of each type, if appropriate.*

The dataset contains 11,757 individuals who were assigned a risk score by the COMPAS tool during pre-trial.

### II-D Does the dataset contain all possible instances, or is it a sample (not necessarily random) of instances from a larger set?
*For example, if it is traffic stops from a territory, is it all traffic stops conducted within that territory within a specific time? If not, is it a representative sample of all stops? Describe how representativeness was validated/verified. If it is not representative, please describe why.*

The dataset contains all individuals who were screened by the COMPAS tool in Broward County between 2013 – 2014.

### II-E What data does each instance consist of?
*If there is a large number of variables, please provide a broad description of what is included.*

Each instance consists of the following variables:

- Offender age
- Offender age at first offense
- Race of offender
- Gender of offender
- Jail history
- Prison history
- Charge history, including charge type, charge degree, etc.
- COMPAS score
- Recidivist

### II-F Is there a target label or associated with each instance?
*Please include labels that are likely to be used as target labels, e.g. recidivism.*

There are two target labels: 'COMPAS score' and 'recidivist'. COMPAS score can also further be grouped into Low, Medium, and High. Scores 1 to 4 were labeled by COMPAS as "Low"; 5 to 7 were labeled "Medium"; and 8 to 10 were labeled "High."

### II-G Are there recommended data splits (e.g., training, development/validation, testing)?
*If so, please provide a description of these splits, explaining the rationale behind them.*

No.

**II-H** *Does the dataset contain data on race and ethnicity?*
If so, is it based on the individual's self-description, or based on officer's impression? Was it collected or derived in post-processing? For example, by name analysis.

Information on race is included. It is unclear if it is based on self-description or not.

**II-I** *Are there any known errors, sources of noise, bias or missing data, or variables collected for only part of the datasets?*
If so, please provide a description.

The dataset creator note that "We found that sometimes people's names or dates of birth were incorrectly entered in some records".

**II-J** *Does the dataset contain data on criminal history or other data that might be considered confidential or sensitive in any way?*
For example: sexual orientations, religious beliefs, political opinions or union memberships, or locations; financial or health data; biometric or genetic data; forms of government identification, such as social security numbers; If so, please provide a description.

The dataset contains information criminal history, prison history, and jail history as well as demographic information.

**II-K** *Is it possible to identify individuals (i.e., one or more natural persons), either directly or indirectly (i.e., in combination with other data) from the dataset?*
If so, please describe how.

Yes, the individuals are named in the dataset.

## III. USES

**III-A** *What type of tasks, if any, has the dataset been used for?*
If so, please provide examples and include citations.

The dataset has been used for:

- Investigating whether there are any racial biases in the COMPAS algorithm [3], [4].
- Evaluating the performance of "fair" algorithms, i.e. balancing predictive performance along with a defined fairness criteria [5].

**III-B** *Is there a repository that links to any or all papers or systems that use the dataset?*
If so, please provide a link or other access point.

There is no specific repository. General academic search engines, such as Google Scholar, can be used with search terms such as: "COMPAS risk assessment".

**III-C** *What (other) tasks could the dataset be used for?*
For example: testing predictive policing systems, predicting recidivism.

This dataset was created for a specific purpose, but was adopted by the algorithmic fairness community as a benchmark. This practice was has been criticized due to lack of domain context [6].

**III-D** *Is there anything about the composition of the dataset or the way it was collected and pre-processed/cleaned/labeled that might impact future uses?*
For example, is there anything that a dataset consumer might need to know to avoid uses that could result in unfair treatment of individuals or groups (e.g., stereotyping, quality of service issues) or other risks or harms (e.g., legal risks, financial harms)? If so, please provide a description. Is there anything a dataset consumer could do to mitigate these risks or harms?

There are a few important things to keep in mind when using this dataset:

- **Potential Incorrect Data Entry**: The ProPublica authors note: "We found that sometimes people's names or dates of birth were incorrectly entered in some records".
- **Conclusion the dataset contains racial bias is disputed**: There are a number of critics of the paper which this dataset is from. Criticisms can be categorized into (1) not using the same set of variables Northpointe used to compute the COMPAS score, (2) incorrect modelling assumptions [3].
- **Only uses local charges**: The dataset uses charges in the Broward County database, which only contains the local charges. If criminal history exists outside this, it is not captured. Additionally, if re-offending occurs outside the county, it will not be counted as recidivism.
- **Recidivism timer starts at screening**: "we defined recidivism as a new arrest within two years" (of screening). If an offender is taken into custody, they will have less opportunity to re-offend, biasing the results.

## IV. COLLECTION PROCESS

**IV-A** *How was the data associated with each instance acquired?*
e.g. the data collected survey, the raw data is routinely collected by the courts.

The dataset is linked from four data sources, matched on first and last name:

1) A public records request of COMPAS scores from Broward County Sheriff's Office in Florida.
2) Charge history from the Broward County Clerk's Office website.
3) Jail records from the Broward County Sheriff's Office.
4) Public incarceration records from the Florida Department of Corrections website.

**IV-B** *Was the information self-reported?*
If the data was self-reported, was the data validated/verified? If so, please describe how.

The data is not self-reported.

**IV-C** *Who was involved in the data collection process?*
Was this done as part of their other duties? If not, were they compensated?

All data was received from the Broward County Sheriff's Office. The raw data was collected as part of routine law

enforcement work.

**IV-D** *Over what timeframe was the data collected? Does this timeframe match the creation timeframe of the data associated with the instances (e.g., recent crawl of old news articles)?*
If not, please describe the timeframe in which the data associated with the instances was created. If the collection was not continuous within the timeframe, please specify the intervals, for example, annually, every 4 years, irregularly.

The dataset was created in 2016 by ProPublica, and concerns the 2013–2014 time period.

**IV-E** *Were any ethical review processes conducted (e.g., by an institutional review board)?*
If so, please provide a description of these review processes, including the outcomes, as well as a link or other access point to any supporting documentation.

Unknown.

**IV-F** *Were the individuals in question notified about the data collection? Did they give their consent?*
If consent was obtained, were the consenting individuals provided with a mechanism to revoke their consent in the future or for certain uses?

No.

**IV-G** *Has an analysis of the potential impact of the dataset and its use on data subjects (e.g., a data protection impact analysis) been conducted?*
If so, please provide a description of this analysis, including the outcomes, as well as a link or other access point to any supporting documentation.

Unknown.

## V. PRE-PROCESSING, CLEANING, LABELING

**V-A** *Was any preprocessing/cleaning/labeling of the data done (e.g., discretization or bucketing, removal of instances, processing of missing values)?*
If so, please provide a description and reference to the documentation. If not, you may skip the remaining questions in this section.

The dataset only offenders who were screened in pretrial. This reduced the number of individuals from 18,610 to 11,757.

**V-B** *Was the "raw" data saved in addition to the preprocessed/cleaned/labeled data?*
If so, please provide a link or other access point to the "raw" data.

Yes. This can be found here:
https://github.com/propublica/compas-analysis

**V-C** *Is the software that was used to preprocess/clean/label the data available?*
If so, please provide a link or other access point.

Yes. This can be found here:
https://github.com/propublica/compas-analysis

**V-D** *Is the data publicly available? How and where can it be accessed (e.g., website, GitHub)?*
Does the dataset have a digital object identifier (DOI)?

The dataset is available on GitHub:
https://github.com/propublica/compas-analysis

**V-E** *When will the dataset be distributed?*

The dataset has been available since 2016.

**V-F** *Is the dataset be distributed under a copyright or other intellectual property (IP) license, and/or under applicable terms of use (ToU)?*
If so, please describe this license and/or ToU, and provide a link or other access point to, or otherwise reproduce, any relevant licensing terms or ToU, as well as any fees associated with these restrictions.

A license is not specified.

## VI. MAINTENANCE

**VI-A** *Is the dataset maintained? Who is supporting/hosting/maintaining the dataset?*

The dataset is no longer maintained.

**VI-B** *How can the owner/curator/manager of the dataset be contacted (e.g., email address)?*

ProPublica can be contacted at: hello@propublica.org

**VI-C** *Will the dataset be updated (e.g., to correct labeling errors, add new instances, delete instances)?*

No, the dataset hasn't been updated since 2017.

**VI-D** *Are older versions of the dataset continue to be supported/hosted/maintained?*

The dataset is no longer updated. However, older versions of the dataset will continue to be accessible via GitHub if any updates occur.

**VI-E** *If others want to extend/augment/build on/contribute to the dataset, is there a mechanism for them to do so?*
If so, please provide a description.

Yes, via a GitHub pull request, given the authors are still active.

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

# NeuLaw's Criminal Record Database
## Datasheet

## I. MOTIVATION

### I-A   For what purpose was the dataset created?

According to the creators of the dataset, the dataset was created "To allow large-scale, cross-jurisdictional analyses of criminal arrests" and "enhance many types of research – for example, identification of high-frequency offenders, measurement of changes in policing strategies, and quantification of legislative efficacy – giving policy makers the best data upon which to base law enforcement decisions" [1].

### I-B   Who created the dataset?
Is it an official law enforcement or government body? An academic research team? Other?

The codebook lists Gabe Haarsma, Sasha Davenport, Pablo A. Ormachea & David M. Eagleman as authors [2].

### I-C   Was there a specific task in mind, or gap that needed to be filled?

The dataset was created to improve on the information available from the UCR SRS program. Specifically, according to the creators, the advantages of this novel dataset include: (1) individual identifiers allow for recidivism analysis—albeit only for repeated bookings within the same jurisdiction (2) the presence of all the charges allows for deeper understanding of all crime, not just a subset, (3) more and different offender-specific variables than the UCR, (4) the data represent a comprehensive and growing picture of information available to judges and prosecutors, and (5) more and different disposition-specific variables, enabling assessment of small variations in punishment [1].

### I-D   Any other comments?

There maybe an updated version of this dataset that is not freely available, from here: hrefhttp://scilaw.org/risk-assessment/ http://scilaw.org/risk-assessment/.

## II. COMPOSITION

### II-A   What do the instances that comprise the dataset represent?
For example: crimes, offenders, court cases, police officers

Records of criminal charges. The specific variables varies depending on the jurisdiction as described below.

### II-B   Are there multiple types of instances?
For example: offenders, victims, and the relationship between them.

Not within each jurisdiction.

### II-C   How many instances are there in total?
Of each type, if appropriate.

**Harris County, TX:** 3.1 million records, spanning from 1977 to April, 2012.
**New York City, NY:** 9.8 million records spanning from 1977 to 2013.

**Miami-Dade County, FL:** 5.7 million records spanning from 1971 to 2012.

### II-D   Does the dataset contain all possible instances or is it a sample (not necessarily random) of instances from a larger set?
For example, if it is traffic stops from a territory, is it all traffic stops conducted within that territory within a specific time? If not, is it a representative sample of all stops? Describe how representativeness was validated/verified. If it is not representative, please describe why.

The dataset includes all records from each jurisdiction, within the stated time frame. Some data instances were removed in pre-processing. In addition:

(1) The database contains no juvenile records, as those are not included in basic Freedom of Information Act requests. We note that juvenile is defined differently in each locale, so 17 year olds are included in Harris County records whereas only 18 year olds appear in New York City and Miami-Dade County records.

(2) The database does not include sealed or expunged records, as those are typically removed from the underlying county databases. It is likely that this disproportionately affects certain crime types (e.g., traffic offenses).

### II-E   What data does each instance consist of?
If there is a large number of variables, please provide a broad description of what is included.

In the **Harris County dataset**, each instance contains Information regarding the:
1. Offense: date, code, name, degree, bond amount at the time of arrest, category, broad category.
2. Defendant: unique ID, race, gender, DOB (mm/yyyy), height, weight, citizenship status.
3. Case: unique case ID, date filed, offense degree, case bond, case status.
4. Attorney: hired or assigned. 5. Grand jury: date, defendant present, and jury action code.
6. Disposition: date, plea, disposition (e.g., dismissed).

In the **New York City dataset**, each instance contains Information regarding the:
1. Offense: month, year.
2. Arrest: county, month, year, charge, crime category, broad crime category.
3. Defendant: race, gender, age at arrest.
3. Disposition: county, month, year, charge, disposition.

In the **Miami-Dade County dataset**, each instance contains Information regarding the:
1. Arrest: date, code, crime category, broad crime category.
2. Case: date filed, date closed, offense degree, trial type (Bench / Jury), case code, case status.
3. Defendant: race, gender, DOB (mm/yyyy).
4. Disposition: code, plea, disposition.

*II-F   Is there a target label or associated with each instance?*
   Please include labels that are likely to be used as target labels, e.g. recidivism.

There is not a pre-specified target label. However, disposition is most suitable to be used as a target label.

*II-G   Are there recommended data splits (e.g., training, development/validation, testing)?*
   If so, please provide a description of these splits, explaining the rationale behind them.

No.

*II-H   Does the dataset contain data on race and ethnicity?*
   If so, is it based on the individual's self-description, or based on officer's impression? Was it collected or derived in post-processing? For example, by name analysis.

Yes. For race, this originates from the raw data and it is not clear whether it is based on the individual's self-description.

The jurisdictions within the datasets do not identify offenders of Hispanic descent. To obtain a better understanding of the demographics, the creators have estimated the Hispanic population by last name [1].

*II-I   Are there any known errors, sources of noise, bias or missing data, or variables collected for only part of the datasets?*
   If so, please provide a description.

All the records in the database were originally entered by humans. The creators attempted to fix typographical errors. However, a larger problem is missing data. For example, some fields have become more populated with time. Birth date was not as commonly entered in some of the earlier records from the 1970s and 1980s, but becomes more rigorously entered with time 6 [1].

The dataset does not contain corrections records, as most states do not consider those public. Therefore, while we know each offender's sentence at the end of trial or plea bargaining, we cannot know how long an offender actually served [1].

*II-J   Does the dataset contain data on criminal history or other data that might be considered confidential or sensitive in any way?*
   For example: sexual orientations, religious beliefs, political opinions or union memberships, or locations; financial or health data; biometric or genetic data; forms of government identification, such as social security numbers; If so, please provide a description.

The dataset contains partial information on criminal offending, as well as demographic information. The partial criminal offending can be constructed as the dataset contains unique identification numbers that can be linked across multiple offenses in an area. For example, in Harris County, Texas, 44% of the 1.2M uniquely identified offenders have multiple offenses – and therefore a partial record of offense (see Figure 1).

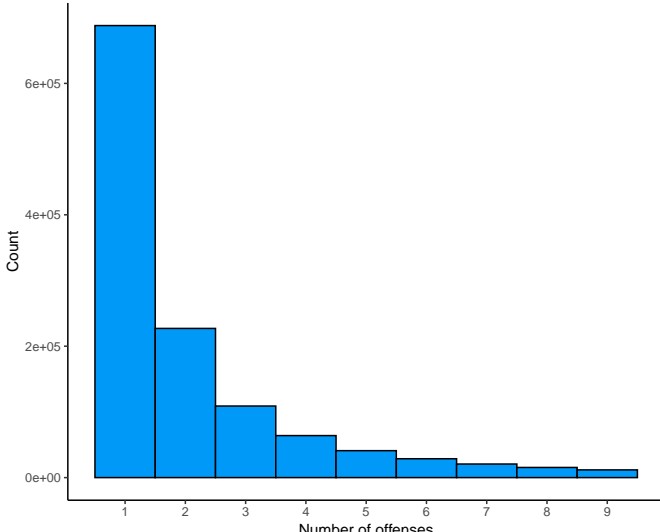

Fig. 1.   A histogram of the number of offenses per offender in Harris County, Texas. The visualization has been limited to offenders that have less than 10 offenses for conciseness. Individuals with more than 10 offenses represent less than 3% of the dataset.

*II-K   Is it possible to identify individuals (i.e., one or more natural persons), either directly or indirectly (i.e., in combination with other data) from the dataset?*
   If so, please describe how.

Possibly, if comparing to other sources such as news articles. Only relevant for cases that attracted media attention.

## III. USES

*III-A   What type of tasks, if any, has the dataset been used for?*
   If so, please provide examples and include citations.

Examples of papers that have used this dataset are [3], [4], [5], [6].

*III-B   Is there a repository that links to any or all papers or systems that use the dataset?*

No.

**III-C** *What (other) tasks could the dataset be used for?*
For example: testing predictive policing systems, predicting recidivism.

The dataset can be used for research questions around case disposition and sentencing. A partial criminal record can be constructed from the Harris County dataset.

**III-D** *Is there anything about the composition of the dataset or the way it was collected and pre-processed/cleaned/labeled that might impact future uses?*
For example, is there anything that a dataset consumer might need to know to avoid uses that could result in unfair treatment of individuals or groups (e.g., stereotyping, quality of service issues) or other risks or harms (e.g., legal risks, financial harms)? If so, please provide a description. Is there anything a dataset consumer could do to mitigate these risks or harms?

The dataset only contains arrest data and not incident-based data, thus providing a picture of crime at the courthouse level. This means that previous stages in the law enforcement process (e.g., 911 calls, house calls, etc.) could skew the arrests that make it into courthouse databases [1].

The recidivism analysis allowed by this only applies for repeated bookings within the same jurisdiction. This approach will systematically undercount the true recidivism rate due to relocation [1].

The dataset does not have victim data, precluding the analysis of, for example, whether ethnicity or age of victim affects sentencing [1].

Some jurisdictions have more limited data than the rest. For example, New York City's records only list the most serious offense per arrest and do not yet include an identifier [1].

While our Broad categorization allows for comparisons across jurisdictions, the detailed categorization does not. The subcategories become populated only if the jurisdictions' labels or code citations provided enough detail [1].

## IV. COLLECTION PROCESS

**IV-A** *How was the data associated with each instance acquired?*
e.g. the data collected survey, the raw data is routinely collected by the courts.

To acquire the underlying data, the dataset creators "contacted New York City (New York), Harris County (Houston), and MiamiDade County (Miami), to obtain copies of their criminal records from their justice information management systems. As public records, the data were obtained via Freedom of Information Act requests" [1].

**IV-B** *Was the information self-reported?*
If the data was self-reported, was the data validated/verified? If so, please describe how.

No. The data was derived from a dataset of criminal records used by respective local authorities. It was not collected for research purposes.

**IV-C** *Who was involved in the data collection process?*
Was this done as part of their other duties? If not, were they compensated?

The data was entered into the courts data systems by employs of the courts.

**IV-D** *Over what timeframe was the data collected? Does this timeframe match the creation timeframe of the data associated with the instances (e.g., recent crawl of old news articles)?*
If not, please describe the timeframe in which the data associated with the instances was created. If the collection was not continuous within the timeframe, please specify the intervals, for example, annually, every 4 years, irregularly.

Harris County, TX – 1977 to April, 2012.
New York City, NY – 1977 to 2013.
Miami-Dade County, FL – 1971 to 2012.

**IV-E** *Were any ethical review processes conducted (e.g., by an institutional review board)?*
If so, please provide a description of these review processes, including the outcomes, as well as a link or other access point to any supporting documentation.

Unknown. The dataset creators do state that "The Institutional Review Board at Baylor College of Medicine exempted this release of an anonymized dataset from human subject research oversight because they consist of publicly available records" [1].

**IV-F** *Were the individuals in question notified about the data collection? Did they give their consent?*
If consent was obtained, were the consenting individuals provided with a mechanism to revoke their consent in the future or for certain uses?

It is likely the individuals know of their criminal charges. It is unlikely they knew or gave consent for it to be used as part of a research dataset.

**IV-G** *Has an analysis of the potential impact of the dataset and its use on data subjects (e.g., a data protection impact analysis) been conducted?*
If so, please provide a description of this analysis, including the outcomes, as well as a link or other access point to any supporting documentation.

Unknown.

## V. PRE-PROCESSING, CLEANING, LABELING

**V-A** *Was any preprocessing/cleaning/labeling of the data done (e.g., discretization or bucketing, removal of instances, processing of missing values)?*
If so, please provide a description and reference to the documentation. If not, you may skip the remaining questions in this section.

Yes. Data processing is described is detail in [1] and in the codebook [2]. Broadly, the data was cleaned and standardized, and duplicated entries were removed. Entries have

been de-identified by removing names, addresses, etc. DOB was replaced with the month and year only. In the Harris County dataset, defendants and cases were given a unique identifiers. The creators added seven calculated variables for all the datasets: 1. Broad crime category (32 categories), 2. Detailed crime category ($\sim 150 - 175$ categories) 3. Standardized disposition[1] 4. Gender, using given name to determine gender when missing or unknown. 5. Race, using surname to add Hispanic ethnicity. 6. The defendant age at the time of case filed or the arrest date. 7. The year the case is filed. 8. Aggregated case numbers to combine multiple offenses into single case (Harris County only).

*V-B   Was the "raw" data saved in addition to the prepro­cessed/cleaned/labeled data?*
    If so, please provide a link or other access point to the "raw" data.

Yes. The calculated age, race and gender variables are added to the dataset alongside the raw variables.

*V-C   Is the software that was used to preprocess/clean/label the data available?*
    If so, please provide a link or other access point.

No.

## VI. DISTRIBUTION

*VI-A   Is the data publicly available? How and where can it be accessed (e.g., website, GitHub)?*
    Does the dataset have a digital object identifier (DOI)?

Yes. The dataset can be found here [2].

*VI-B   Is the dataset be distributed under a copyright or other intellectual property (IP) license, and/or under applicable terms of use (ToU)?*
    If so, please describe this license and/or ToU, and provide a link or other access point to, or otherwise reproduce, any relevant licensing terms or ToU, as well as any fees associated with these restrictions.

The dataset is licensed under a Creative Commons Attri­bution 4.0 International (CC BY 4.0) License.

## VII. MAINTENANCE

*VII-A   Is the dataset maintained? Who is support­ing/hosting/maintaining the dataset?*

The dataset is not maintained. There maybe an updated version of this dataset that is not freely available, from here: hrefhttp://scilaw.org/risk-assessment/ http://scilaw.org/risk-assessment/.

*VII-B   How can the owner/curator/manager of the dataset be contacted (e.g., email address)?*

Unknown.

*VII-C   Will the dataset be updated (e.g., to correct labeling errors, add new instances, delete instances)?*

No.

*VII-D   Are older versions of the dataset continue to be supported/hosted/maintained?*

N/A.

*VII-E   If others want to extend/augment/build on/contribute to the dataset, is there a mechanism for them to do so?*
    If so, please provide a description.

No.

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

---

[1]standardized disposition has 7 possible dispositions: No Action, Dis­missal, Transfer, Acquittal, Guilty, Guilty by Plea, Conditional Dismissal, and Unknown/No Final Disposition

# Virginia Pre-Trial Data Project
## Datasheet

## I. MOTIVATION

### I-A  For what purpose was the dataset created?

The Virginia State Crime Commission has been studying various aspects of the pre-trial process since 2016. However, there was a significant lack of data readily available to answer many important questions related to the pre-trial process in the Commonwealth. As a result, the Virginia Pre-Trial Data Project was developed [1].

### I-B  Who created the dataset?
*Is it an official law enforcement or government body? An academic research team? Other?*

The project was lead by the Virginia State Crime Commission. Data was collected from the following agencies: Supreme Court of Virginia, Office of the Executive Secretary; Alexandria Circuit Court; Fairfax County Circuit Court; Virginia Department of Criminal Justice Services; Virginia State Police; Virginia Department of Corrections; Virginia Compensation Board.

### I-C  Was there a specific task in mind, or gap that needed to be filled?

Following individuals from pre-trial to final dispositions, including assigned risk levels. The Project consisted of two phases: (i) developing a cohort of adult defendants charged with a criminal offense in Virginia during October 2017 and (ii) tracking various outcomes within that cohort [1].

## II. COMPOSITION

### II-A  What do the instances that comprise the dataset represent?
*For example: crimes, offenders, court cases, police officers*

Each instance reports on a 'contact event', defined as: "all charges against a defendant in the same jurisdiction on the same day and having the same CBR number" (CBR stands for "Commit, Bond, Release" and refers to any one of these bail processes) [1].

Each instance includes information regarding the defendant, and the progression of the criminal case from the time a defendant is charged with an offense until the final disposition of the case, i.e., trial or sentencing [1].

If the same individual has more than one contact event during the month of October 2017, only the earlier contact event is reported in the data. If a defendant was charged with multiple offenses on the same day, but the offenses were heard in different courts, those records were grouped by court and reported as separate events [1].

### II-B  Are there multiple types of instances?
*For example: offenders, victims, and the relationship between them.*

No.

### II-C  How many instances are there in total?
*Of each type, if appropriate.*

The dataset contains 22,986 adult defendants charged with a criminal offense during a October 2017.

### II-D  Does the dataset contain all possible instances or is it a sample (not necessarily random) of instances from a larger set?
*For example, if it is traffic stops from a territory, is it all traffic stops conducted within that territory within a specific time? If not, is it a representative sample of all stops? Describe how representativeness was validated/verified. If it is not representative, please describe why.*

The cohort includes all defendants charged in Virginia in October 2017. However, some instances were excluded from the follow-up and as a result can not be used for most analysis. The reason for the excluding is reported in the variable 'Exclude' and include missing data, the defendant being under 18 when they were charged, and the offense not being punishable by incarceration, amongst other reasons.

### II-E  What data does each instance consist of?
*If there is a large number of variables, please provide a broad description of what is included.*

The dataset contains over 700 variables for each defendant. Broadly, these include:
1. Demographics: Sex, Race, Age, Indigency Status, Virginia Residency Status, Zip Code.
2. Pending charges.
3. State or local probation status.
4. October 2017 charge(s): number of offense, offense and offense type (up to 10).
5. Bond: Bond Type and amount at initial contact and at release.
6. Release status: Whether Defendant Was Released During Pre-trial Period, Pre-Trial Release Date, Pretrial Release Type.
7. Whether the defendant received pretrial services agency supervision: supervision Days, conditions. Whether defendant is on state or community supervision.
8. Case: attorney type, court type, court locality, sentence type, imposed and effective sentence, Final Disposition and disposition date.

9. Prior criminal history: age at first adult arrest; number prior arrests for felonies, misdemeanors, and specific crimes, e.g., domestic abuse; number of prior convictions overall, as an adult, in the past 2 and 5 years, for felonies, misdemeanors, and specific crimes, e.g., drug convictions. Prior sentencing for felonies and probation Revocation ; prior probation revocations, number of prior incarceration events for more than 14 days, less and more than a year.

10: Risk level: components to calculate VPRAI [2] and PSA [3] risk levels, and corresponding scores.

11. Court appearance and public safety: details on new failure to appear and offending in the follow up period.

12: Aggregate locality characteristics: locality Name, Region, population estimate, density and demographic (race, ethnicity, sex and age combined) composition, unemployment and education rates, number of law enforcement officers, income, health insurance, citizenship status; incident and arrest rate overall and for specific crimes.

**II-F** *Is there a target label or associated with each instance?*
    Please include labels that are likely to be used as target labels, e.g. recidivism.

There is not pre-specified target label. However, variables under the court appearance and public safety, e.g., new offending, are particularly suitable to be used as target variables.

**II-G** *Are there recommended data splits (e.g., training, development/validation, testing)?*
    If so, please provide a description of these splits, explaining the rationale behind them.

No.

**II-H** *Does the dataset contain data on race and ethnicity?*
    If so, is it based on the individual's self-description, or based on officer's impression? Was it collected or derived in post-processing? For example, by name analysis.

Information on race is included. This information is taken from court records. It is unclear if it is based on self-description or not. Ethnicity is only partly recorded in the raw data. As a result, in the dataset Hispanic ethnicity is considered within the White racial category.

**II-I** *Are there any known errors, sources of noise, bias or missing data, or variables collected for only part of the datasets?*
    If so, please provide a description.

Records with missing data were excluded from the follow on analysis. The reason for the exclusion is reported in the variable 'Exclude' (See pg. 270 is [1]).

**II-J** *Does the dataset contain data on criminal history or other data that might be considered confidential or sensitive in any way?*
    For example: sexual orientations, religious beliefs, political opinions or union memberships, or locations; financial or health data; biometric or genetic data; forms of government identification, such as social security numbers; If so, please provide a description.

The dataset contains information about criminal history, as well as demographic information.

**II-K** *Is it possible to identify individuals (i.e., one or more natural persons), either directly or indirectly (i.e., in combination with other data) from the dataset?*
    If so, please describe how.

Unlikely. Only indirectly, and only if the case received significant media attention.

## III. USES

**III-A** *What type of tasks, if any, has the dataset been used for?*
    If so, please provide examples and include citations.

Yes. Finding from the project can be found in [4].

**III-B** *Is there a repository that links to any or all papers or systems that use the dataset?*
    If so, please provide a link or other access point.

No.

**III-C** *What (other) tasks could the dataset be used for?*
    For example: testing predictive policing systems, predicting recidivism.

This dataset can be used to study risk assessment scores, pre-trial custody status, sentencing and recidivism.

**III-D** *Is there anything about the composition of the dataset or the way it was collected and preprocessed/cleaned/labeled that might impact future uses?*
    For example, is there anything that a dataset consumer might need to know to avoid uses that could result in unfair treatment of individuals or groups (e.g., stereotyping, quality of service issues) or other risks or harms (e.g., legal risks, financial harms)? If so, please provide a description. Is there anything a dataset consumer could do to mitigate these risks or harms?

Offender are from a single cohort, october 2017.

Although criminal records were extracted for all defendants in the cohort, the data does not include those records. We have standardized information regarding the criminal history of the defendant, which may not be suitable for all uses.

Hispanic ethnicity within the White racial category.

## IV. COLLECTION PROCESS

*IV-A How was the data associated with each instance acquired?*

e.g. the data collected survey, the raw data is routinely collected by the courts.

The data is a compilation of information and variables provided by numerous state and local government agencies across Virginia:

1. Supreme Court of Virginia, Office of the Executive Sectretary: eMagistrate Sytem; Circuit, General District, and Juvenile and Domestic Relations District Court Case Management Systems.
2. Alexandria Circuit Court: Alexandria Circuit Court Case Management System.
3. Fairfax County Circuit Court: Fairfax County Circuit Court Case Management System.
4. Virginia Department of Criminal Justice Services: Pretrial and Community Corrections Case Management System (PTCC).
5. Virginia State Police: Central Criminal Records Exchange (CCRE).
6. Virginia Department of Corrections: Corrections Information System (CORIS).
7. Virginia Compensation Board: Local Inmate Data System (LIDS).

*IV-B Was the information self-reported?*

If the data was self-reported, was the data validated/verified? If so, please describe how.

No. Data was extracted from government record keeping systems. No direct validation of the data has been conducted.

*IV-C Who was involved in the data collection process?*

Was this done as part of their other duties? If not, were they compensated?

The data collected by paid workers (assumed).

*IV-D Over what timeframe was the data collected? Does this timeframe match the creation timeframe of the data associated with the instances (e.g., recent crawl of old news articles)?*

If not, please describe the timeframe in which the data associated with the instances was created. If the collection was not continuous within the timeframe, please specify the intervals, for example, annually, every 4 years, irregularly.

Adult defendants charged with a criminal offense during October 2017 were included. They were tracked until final case disposition or December 31, 2018, whichever came first.

*IV-E Were any ethical review processes conducted (e.g., by an institutional review board)?*

If so, please provide a description of these review processes, including the outcomes, as well as a link or other access point to any supporting documentation.

Unknown.

*IV-F Did you collect the data from the individuals in question directly, or obtain it via third parties or other sources?*

The data was collected from government agencies.

*IV-G Were the individuals in question notified about the data collection? Did they give their consent?*

If consent was obtained, were the consenting individuals provided with a mechanism to revoke their consent in the future or for certain uses?

Individuals likely know their data was entered into the agency's data collection system. It is unlikely they knew or consented the use of the data for research purposes.

*IV-H Has an analysis of the potential impact of the dataset and its use on data subjects (e.g., a data protection impact analysis) been conducted?*

If so, please provide a description of this analysis, including the outcomes, as well as a link or other access point to any supporting documentation.

Unknown.

## V. PRE-PROCESSING, CLEANING, LABELING

*V-A Was any preprocessing/cleaning/labeling of the data done (e.g., discretization or bucketing, removal of instances, processing of missing values)?*

If so, please provide a description and reference to the documentation. If not, you may skip the remaining questions in this section.

Details regarding pre-processing can be found in the codebook [1]. Broadly, the defendant's criminal history is not presented in its raw from. For example, we do not see that defendant a was charged with offense $A$ in the year $X$ and so on. Instead, we are told that defendant a has $N$ prior charges from type $A$.

*V-B Was the "raw" data saved in addition to the preprocessed/cleaned/labeled data?*

If so, please provide a link or other access point to the "raw" data.

No.

*V-C Is the software that was used to preprocess/clean/label the data available?*

If so, please provide a link or other access point.

No.

## VI. DISTRIBUTION

*VI-A Is the data publicly available? How and where can it be accessed (e.g., website, GitHub)?*

Does the dataset have a digital object identifier (DOI)?

Yes. The dataset is publicly available on The Virginia State Crime Commission's website:
http://www.vcsc.virginia.gov/pretrialdataproject.htmlhttp://www.vcsc.virgi

**41**

**VI-B** *Is the dataset be distributed under a copyright or other intellectual property (IP) license, and/or under applicable terms of use (ToU)?*

If so, please describe this license and/or ToU, and provide a link or other access point to, or otherwise reproduce, any relevant licensing terms or ToU, as well as any fees associated with these restrictions.

The data is available for download. We did not find information about a specific license.

**VI-C** *Have any third parties imposed IP-based or other restrictions on the data associated with the instances?*

Unknown.

## VII. MAINTENANCE

**VII-A** *Is the dataset maintained? Who is supporting/hosting/maintaining the dataset?*

Yes. The Virginia State Crime Commission.

**VII-B** *How can the owner/curator/manager of the dataset be contacted (e.g., email address)?*

At the point of this publication, it is stated that "if you are having trouble downloading the dataset, please email meredith.farrar-owens@vcsc.virginia.gov or call Sentencing Commission staff at 804.225.4398"

**VII-C** *Will the dataset be updated (e.g., to correct labeling errors, add new instances, delete instances)?*

"Data continues to be reviewed, revised, and validated as necessary" [1].

**VII-D** *Are older versions of the dataset continue to be supported/hosted/maintained?*

N/A

**VII-E** *If others want to extend/augment/build on/contribute to the dataset, is there a mechanism for them to do so?*

If so, please provide a description.

No.

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

# JUSTFAIR
# Datasheet

## I. Motivation

### I-A  For what purpose was the dataset created?

This dataset was created to increase the ease of access to data on public criminal trials. Specifically, it combines information from the US sentencing commission, Federal judicial center, PACER, wikipedia, and Federal Judicial Center biographies to create a dataset of defendants and their demographic characteristics with information about their crimes, their sentences, and the identity of the sentencing judge [1].

### I-B  Who created the dataset?
*Is it an official law enforcement or government body? An academic research team? Other?*

The dataset was created by researchers at the Institute for the Quantitative Study of Inclusion, Diversity, and Equity (QSIDE): Maria-Veronica Ciocanel, Chad Topaz, Rebecca Santorella, Shilad Sen, Christian Smith, and Adam Hufstetler.

### I-C  Was there a specific task in mind, or gap that needed to be filled?

The authors wished to determine whether judges with significant sentencing outcome disparities across the race of defendants were due to racial bias by creating a dataset with sufficient controls, such as defendant's education level or age.

## II. Composition

### II-A  What do the instances that comprise the dataset represent?
*For example: crimes, offenders, court cases, police officers*

Each instance in this dataset corresponds to a single criminal trial sentencing.

### II-B  Are there multiple types of instances?
*For example: offenders, victims, and the relationship between them.*

No.

### II-C  How many instances are there in total?
*Of each type, if appropriate.*

There are a total of 595,850 sentences in the dataset.

### II-D  Does the dataset contain all possible instances or is it a sample (not necessarily random) of instances from a larger set?
*For example, if it is traffic stops from a territory, is it all traffic stops conducted within that territory within a specific time? If not, is it a representative sample of all stops? Describe how representativeness was validated/verified. If it is not representative, please describe why.*

The United States Sentencing Commission maintains publicly accessible data sets, including files which provide information about sentences given to individuals in federal district courts. This dataset contained a subset of these which have been successfully linked with the data from other datasets.

### II-E  What data does each instance consist of?
*If there is a large number of variables, please provide a broad description of what is included.*

The JUSTFAIR dataset is a result of linking together five datasets and as a result, it contains many variables. A high-level overview of the available information can be in Figure 1.

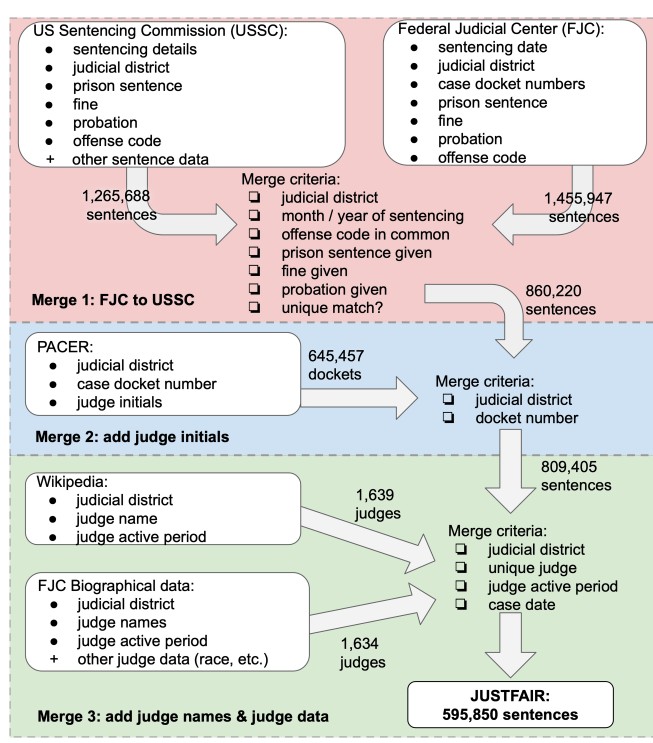

Fig. 1.  An illustration of how the five datasets are linked together to form the JUSTFAIR dataset. Reprinted from [1].

Broadly, the information contains in the dataset consists of:

- Sentencing Details
- Prison sentence
- Fine
- Probation
- Offense
- Judicial District
- Judge Name
- Defendant demographics
- Judge demographics

**II-F** *Is there a target label or associated with each instance?*
Please include labels that are likely to be used as target labels, e.g. recidivism.

No. However, different aspects of sentencing, may be suitable to be used as target labels.

**II-G** *Are there recommended data splits (e.g., training, development/validation, testing)?*
If so, please provide a description of these splits, explaining the rationale behind them.

No.

**II-H** *Does the dataset contain data on race and ethnicity?*
If so, is it based on the individual's self-description, or based on officer's impression? Was it collected or derived in post-processing? For example, by name analysis.

Yes. The dataset contains defendant demographic information, as well as demographic information on the judge .

**II-I** *Are there any known errors, sources of noise, bias or missing data, or variables collected for only part of the datasets?*
If so, please provide a description.

JUSTFAIR uses a combination of Wikipedia and Federal Judicial Center biographical data to match the judge to the sentencing details. Based on 159 manual test cases, this matching has 2% error rate.

**II-J** *Does the dataset contain data on criminal history or other data that might be considered confidential or sensitive in any way?*
For example: sexual orientations, religious beliefs, political opinions or union memberships, or locations; financial or health data; biometric or genetic data; forms of government identification, such as social security numbers; If so, please provide a description.

The dataset contains sentencing data on individuals which may be considered confidential.

**II-K** *Is it possible to identify individuals (i.e., one or more natural persons), either directly or indirectly (i.e., in combination with other data) from the dataset?*
If so, please describe how.

Yes, the dataset purposely identifies the judges involved in the cases. In addition, the defendant's name also appears in the data.

## III. USES

**III-A** *What type of tasks, if any, has the dataset been used for?*
If so, please provide examples and include citations.

The dataset has been used to study racial bias in the judicial system [1], [2], [3], [4].

**III-B** *Is there a repository that links to any or all papers or systems that use the dataset?*
If so, please provide a link or other access point.

No.

**III-C** *What (other) tasks could the dataset be used for?*
For example: testing predictive policing systems, predicting recidivism.

This dataset could be used to study research questions around sentencing and probation in the relevant courts.

**III-D** *Is there anything about the composition of the dataset or the way it was collected and preprocessed/cleaned/labeled that might impact future uses?*
For example, is there anything that a dataset consumer might need to know to avoid uses that could result in unfair treatment of individuals or groups (e.g., stereotyping, quality of service issues) or other risks or harms (e.g., legal risks, financial harms)? If so, please provide a description. Is there anything a dataset consumer could do to mitigate these risks or harms?

Results will have to be analyzed carefully due to the non-negligible error rate in judge-case matching.

## IV. COLLECTION PROCESS

**IV-A** *How was the data associated with each instance acquired?*
e.g. the data collected survey, the raw data is routinely collected by the courts.

The dataset was created by merging five datasets:

1) the United States Sentencing Commission Database
2) the Federal Judicial Center Integrated Database
3) the Public Access to Court Electronic Records system
4) Wikipedia
5) the Federal Judicial Center Biographical Directory of Article III Federal Judges

**IV-B** *Was the information self-reported?*
If the data was self-reported, was the data validated/verified? If so, please describe how.

No. However, some of the information in the individual datasets, e.g., Wikipedia, may be self-reported.

*IV-C*  *Who was involved in the data collection process?*
Was this done as part of their other duties? If not, were they compensated?

The data was collated by the QSIDE institute.

*IV-D*  *Over what timeframe was the data collected? Does this timeframe match the creation timeframe of the data associated with the instances (e.g., recent crawl of old news articles)?*
If not, please describe the timeframe in which the data associated with the instances was created. If the collection was not continuous within the timeframe, please specify the intervals, for example, annually, every 4 years, irregularly.

The dataset was published October 26, 2020, and covers the years 2001 — 2018.

*IV-E*  *Were any ethical review processes conducted (e.g., by an institutional review board)?*
If so, please provide a description of these review processes, including the outcomes, as well as a link or other access point to any supporting documentation.

An ethical review is not mentioned by the authors [1].

*IV-F*  *Were the individuals in question notified about the data collection? Did they give their consent?*
If consent was obtained, were the consenting individuals provided with a mechanism to revoke their consent in the future or for certain uses?

No.

*IV-G*  *Has an analysis of the potential impact of the dataset and its use on data subjects (e.g., a data protection impact analysis) been conducted?*
If so, please provide a description of this analysis, including the outcomes, as well as a link or other access point to any supporting documentation.

An impact analysis is not mentioned by the authors [1].

## V. Pre-processing, cleaning, labeling

*V-A*  *Was any preprocessing/cleaning/labeling of the data done (e.g., discretization or bucketing, removal of instances, processing of missing values)?*
If so, please provide a description and reference to the documentation. If not, you may skip the remaining questions in this section.

The pre-processing performed on each of the five datasets can be found in [1].

*V-B*  *Was the "raw" data saved in addition to the preprocessed/cleaned/labeled data?*
If so, please provide a link or other access point to the "raw" data.

The raw data is available on the respective websites of each of five datasets.

*V-C*  *Is the software that was used to preprocess/clean/label the data available?*
If so, please provide a link or other access point.

No.

## VI. Distribution

*VI-A*  *Is the data publicly available? How and where can it be accessed (e.g., website, GitHub)?*
Does the dataset have a digital object identifier (DOI)?

Yes, the dataset can be downloaded at:
https://qsideinstitute.org/research/criminal-justice/justfair/

*VI-B*  *Is the dataset be distributed under a copyright or other intellectual property (IP) license, and/or under applicable terms of use (ToU)?*
If so, please describe this license and/or ToU, and provide a link or other access point to, or otherwise reproduce, any relevant licensing terms or ToU, as well as any fees associated with these restrictions.

No license has been specified.

## VII. Maintenance

*VII-A*  *Is the dataset maintained? Who is supporting/hosting/maintaining the dataset?*

The dataset will be updated, conditional on the creators obtaining additional funding.

*VII-B*  *How can the owner/curator/manager of the dataset be contacted (e.g., email address)?*

The QSIDE Institute can be contacted at: qside@qsideinstitute.org

*VII-C*  *Will the dataset be updated (e.g., to correct labeling errors, add new instances, delete instances)?*

Yes, conditional on the creators obtaining additional funding.

*VII-D*  *Are older versions of the dataset continue to be supported/hosted/maintained?*

No.

*VII-E*  *If others want to extend/augment/build on/contribute to the dataset, is there a mechanism for them to do so?*
If so, please provide a description.

Try contacting qside@qsideinstitute.org.

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

# Survey of Prison Inmates (SPI)
# Datasheet

## I. MOTIVATION

### I-A For what purpose was the dataset created?

The SPI was created to produce national estimates for the state and sentenced federal prison populations. It aims to provide a record and means of tracking inmate characteristics, such as demographics, background, and criminal history [1].

### I-B Who created the dataset?
Is it an official law enforcement or government body? An academic research team? Other?

The dataset was created by the U.S. Bureau of Justice Statistics.

### I-C Was there a specific task in mind, or gap that needed to be filled?

The dataset was created to be the first national periodic inmate survey in the United States, collecting detailed information pertinent to evolving issues within the criminal justice domain.

## II. COMPOSITION

### II-A What do the instances that comprise the dataset represent?
For example: crimes, offenders, court cases, police officers

Each row corresponds to a survey response from a prison inmate.

### II-B Are there multiple types of instances?
For example: offenders, victims, and the relationship between them.

No.

### II-C How many instances are there in total?
Of each type, if appropriate.

There are a total of 24,848 inmates (20,064 state and 4,784 federal prisoners) in the 2016 SPI dataset.

### II-D Does the dataset contain all possible instances or is it a sample (not necessarily random) of instances from a larger set?
For example, if it is traffic stops from a territory, is it all traffic stops conducted within that territory within a specific time? If not, is it a representative sample of all stops? Describe how representativeness was validated/verified. If it is not representative, please describe why.

The dataset is a representative sample of the (over 18) U.S. prison population. Achieved by using a two-stage sample design, where state and federal prisons are selected in the first stage, followed by individuals selected from these facilities in the second stage. Samples are then weighted to account for non-response. Further details can be found in the SIP methodology white paper [1].

### II-E What data does each instance consist of?
If there is a large number of variables, please provide a broad description of what is included.

Each instance broadly consists of:

- Current offense and sentence.
- Incident characteristics.
- Firearm possession and sources.
- Criminal history.
- Demographic and socioeconomic characteristics.
- Family background.
- Drug and alcohol use and treatment.
- Mental and physical health and treatment.
- Facility programs and rules violations.

### II-F Is there a target label or associated with each instance?
Please include labels that are likely to be used as target labels, e.g. recidivism.

No.

### II-G Are there recommended data splits (e.g., training, development/validation, testing)?
If so, please provide a description of these splits, explaining the rationale behind them.

### II-H Does the dataset contain data on race and ethnicity?
If so, is it based on the individual's self-description, or based on officer's impression? Was it collected or derived in post-processing? For example, by name analysis.

No.

### II-I Are there any known errors, sources of noise, bias or missing data, or variables collected for only part of the datasets?
If so, please provide a description.

There are two known potential sources of error/noise: nonresponse (where the demographics of the respondents is significantly different to the non-respondents), and a coverage bias (where the sample population did not represent the target population). Non-response and post-stratification weights are provided to compensate for these.

*II-J  Does the dataset contain data on criminal history or other data that might be considered confidential or sensitive in any way?*

For example: sexual orientations, religious beliefs, political opinions or union memberships, or locations; financial or health data; biometric or genetic data; forms of government identification, such as social security numbers; If so, please provide a description.

Yes, the dataset contains information on criminal history, sentencing, demographic and socioeconomic characteristics, family background, drug and alcohol use and treatment, and mental and physical health and treatment.

*II-K  Is it possible to identify individuals (i.e., one or more natural persons), either directly or indirectly (i.e., in combination with other data) from the dataset?*

If so, please describe how.

Indirectly, by a comparing criminal history, demographic information, and sentencing information with other sources that are not de-identified.

## III. USES

*III-A  What type of tasks, if any, has the dataset been used for?*

If so, please provide examples and include citations.

The dataset has been used to:

- Investigate the demographics and characteristics of inmates [2].
- Investigate specific inmate populations, including women, parents, and minorities [3], [4].
- Investigate the link between rural prisons and incarceration levels [5].
- Investigate the use and sources of firearms used in crimes [6].

*III-B  Is there a repository that links to any or all papers or systems that use the dataset?*

If so, please provide a link or other access point.

Yes. Please see here:

https://www.icpsr.umich.edu/web/NACJD/studies/37692

*III-C  What (other) tasks could the dataset be used for?*

For example: testing predictive policing systems, predicting recidivism.

The dataset could be used to research counterfactual sentencing.

*III-D  Is there anything about the composition of the dataset or the way it was collected and preprocessed/cleaned/labeled that might impact future uses?*

For example, is there anything that a dataset consumer might need to know to avoid uses that could result in unfair treatment of individuals or groups (e.g., stereotyping, quality of service issues) or other risks or harms (e.g., legal risks, financial harms)? If so, please provide a description. Is there anything a dataset consumer could do to mitigate these risks or harms?

No.

## IV. COLLECTION PROCESS

*IV-A  How was the data associated with each instance acquired?*

e.g. the data collected survey, the raw data is routinely collected by the courts.

The data was acquired via interview, as well as being linked to records maintained by other government agencies, such as criminal records.

*IV-B  Was the information self-reported?*

If the data was self-reported, was the data validated/verified? If so, please describe how.

Survey responses are self-reported. Data from official record are not.

*IV-C  Who was involved in the data collection process?*

Was this done as part of their other duties? If not, were they compensated?

The data was collected by employees of the Bureau of Justice Statistics.

*IV-D  Over what timeframe was the data collected? Does this timeframe match the creation timeframe of the data associated with the instances (e.g., recent crawl of old news articles)?*

If not, please describe the timeframe in which the data associated with the instances was created. If the collection was not continuous within the timeframe, please specify the intervals, for example, annually, every 4 years, irregularly.

SPI has released new data irregularly between 1974 – 2016. The latest release is from 2016.

*IV-E  Were any ethical review processes conducted (e.g., by an institutional review board)?*

If so, please provide a description of these review processes, including the outcomes, as well as a link or other access point to any supporting documentation.

Unknown.

*IV-F  Were the individuals in question notified about the data collection? Did they give their consent?*

If consent was obtained, were the consenting individuals provided with a mechanism to revoke their consent in the future or for certain uses?

The individuals were notified: "before the interview prisoners were informed verbally and in writing that their participation was voluntary and that all information provided would be held in confidence" [1].

*IV-G  Has an analysis of the potential impact of the dataset and its use on data subjects (e.g., a data protection impact analysis) been conducted?*

If so, please provide a description of this analysis, including the outcomes, as well as a link or other access point to any supporting documentation.

Unknown.

## V. Pre-processing, cleaning, labeling

*V-A    Was any preprocessing/cleaning/labeling of the data done (e.g., discretization or bucketing, removal of instances, processing of missing values)?*
    If so, please provide a description and reference to the documentation. If not, you may skip the remaining questions in this section.

The only processing specified in the methodology is the non-response and coverage weighting [1].

*V-B    Was the "raw" data saved in addition to the preprocessed/cleaned/labeled data?*
    If so, please provide a link or other access point to the "raw" data.

As the weighting is provided as a separate variable, the raw data is still accessible.

*V-C    Is the software that was used to preprocess/clean/label the data available?*
    If so, please provide a link or other access point.

N/A.

## VI. Distribution

*VI-A    Is the data publicly available? How and where can it be accessed (e.g., website, GitHub)?*
    Does the dataset have a digital object identifier (DOI)?

Yes. The data can be obtained from:
https://www.icpsr.umich.edu/web/NACJD/studies/37692

*VI-B    Is the dataset be distributed under a copyright or other intellectual property (IP) license, and/or under applicable terms of use (ToU)?*
    If so, please describe this license and/or ToU, and provide a link or other access point to, or otherwise reproduce, any relevant licensing terms or ToU, as well as any fees associated with these restrictions.

The license is not specified, but a citation and deposit requirement are listed:

**Citation Requirement:** Publications based on ICPSR data collections should acknowledge those sources by means of bibliographic citations. To ensure that such source attributions are captured for social science bibliographic utilities, citations must appear in footnotes or in the reference section of publications.

**Deposit Requirement:** To provide funding agencies with essential information about use of archival resources and to facilitate the exchange of information about ICPSR participants' research activities, users of ICPSR data are requested to send to ICPSR bibliographic citations for each completed manuscript or thesis abstract. Visit the ICPSR Web site for more information on submitting citations.

## VII. Maintenance

*VII-A    Is the dataset maintained? Who is supporting/hosting/maintaining the dataset?*

Yes, by the Bureau of Justice Statistics.

*VII-B    How can the owner/curator/manager of the dataset be contacted (e.g., email address)?*

The Bureau of Justice Statistics can be contacted at: askbjs@usdoj.gov

*VII-C    Will the dataset be updated (e.g., to correct labeling errors, add new instances, delete instances)?*

No.

*VII-D    Are older versions of the dataset continue to be supported/hosted/maintained?*

Yes.

*VII-E    If others want to extend/augment/build on/contribute to the dataset, is there a mechanism for them to do so?*
    If so, please provide a description.

No.

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

# Law Enforcement Management and Administrative Statistics (LEMAS)
# Datasheet

## I. MOTIVATION

### I-A    For what purpose was the dataset created?

Conducted periodically since 1987, the LEMAS survey was designed to collect data on police agencies, including: agency responsibilities, operating expenditures, job functions of sworn and civilian employees, officer salaries and special pay, demographic characteristics of officers, weapons and armor policies, education and training requirements, computers and information systems, vehicles, special units, and community policing activities [1].

### I-B    Who created the dataset?
*Is it an official law enforcement or government body? An academic research team? Other?*

The Bureau of Justice Statistics (BJS). The Relevant data expert at BJS is Elizabeth Davis.

### I-C    Was there a specific task in mind, or gap that needed to be filled?

No extensive nationwide police agency survey existed before LEMAS.

## II. COMPOSITION

### II-A    What do the instances that comprise the dataset represent?
*For example: crimes, offenders, court cases, police officers*

Each instance of the dataset is a survey response from a different police agency.

### II-B    Are there multiple types of instances?
*For example: offenders, victims, and the relationship between them.*

No.

### II-C    How many instances are there in total?
*Of each type, if appropriate.*

There are a total of 3,471 police agencies were sent the survey, including: 2,612 local police departments, 810 sheriffs' offices, and the 49 state agencies. A total of 2,779 agencies responded to the LEMAS survey, a response rate of 80%. The final dataset includes responses from 2,135 local police departments, 600 sheriffs' offices, and 49 state agencies.

### II-D    Does the dataset contain all possible instances or is it a sample (not necessarily random) of instances from a larger set?
*For example, if it is traffic stops from a territory, is it all traffic stops conducted within that territory within a specific time? If not, is it a representative sample of all stops? Describe how representativeness was validated/verified. If it is not representative, please describe why.*

The dataset includes all police agencies with 100 or more sworn personnel to be included, with smaller agencies sampled via stratified random sampling based on the number of sworn officers, and type of agency [2]. A total of 28 local police departments were determined to be out-of-scope for the survey because they were special jurisdiction agencies, had closed, had outsourced their operations, or were operating on a part-time basis [2].

### II-E    What data does each instance consist of?
*If there is a large number of variables, please provide a broad description of what is included.*

Each instance consists of:
- Agency responsibilities.
- Agency expenditures.
- Officer salaries.
- Officer demographics.
- Agency policies.
- Officer requirements.
- Technology used at agency.
- Agency vehicles.
- Agency community policing practices.

### II-F    Is there a target label or associated with each instance?
*Please include labels that are likely to be used as target labels, e.g. recidivism.*

No.

### II-G    Are there recommended data splits (e.g., training, development/validation, testing)?
*If so, please provide a description of these splits, explaining the rationale behind them.*

No.

### II-H    Does the dataset contain data on race and ethnicity?
*If so, is it based on the individual's self-description, or based on officer's impression? Was it collected or derived in post-processing? For example, by name analysis.*

Yes. The dataset contains information about the demographic composition of the officers within each agency. This

is (likely) based on self-description.

*II-I   Are there any known errors, sources of noise, bias or missing data, or variables collected for only part of the datasets?*
If so, please provide a description.

No.

*II-J   Does the dataset contain data on criminal history or other data that might be considered confidential or sensitive in any way?*
For example: sexual orientations, religious beliefs, political opinions or union memberships, or locations; financial or health data; biometric or genetic data; forms of government identification, such as social security numbers; If so, please provide a description.

No. Demographics are not reported on an individual level.

*II-K   Does the dataset contain data that might be considered confidential?*
For example: data that is protected by legal privilege or by doctor–patient confidentiality, data that includes the content of individuals' nonpublic communications. If so, please provide a description.

No.

*II-L   Is it possible to identify individuals (i.e., one or more natural persons), either directly or indirectly (i.e., in combination with other data) from the dataset?*
If so, please describe how.

No.

## III. Uses

*III-A   What type of tasks, if any, has the dataset been used for?*
If so, please provide examples and include citations.

The dataset has been used for a number of tasks, not limited to, but including:

- Investigating the effect of community policing practices [3], [4], [5].
- Investigating the changing demographics of police agencies [6], [7].
- Investigating technology being used by police agencies, including crime analysis tools [8], [9], [10].

*III-B   Is there a repository that links to any or all papers or systems that use the dataset?*
If so, please provide a link or other access point.

Yes. Please see here:

https://www.icpsr.umich.edu/web/NACJD/series/92/publication

*III-C   What (other) tasks could the dataset be used for?*
For example: testing predictive policing systems, predicting recidivism.

The dataset could be used for any tasks which involve agency:

- Responsibilities.
- Expenditures.
- Salaries.
- Officer demographics.
- Policies.
- Officer requirements.
- Technology used.
- Vehicles.
- Community policing activities.

*III-D   Is there anything about the composition of the dataset or the way it was collected and pre-processed/cleaned/labeled that might impact future uses?*
For example, is there anything that a dataset consumer might need to know to avoid uses that could result in unfair treatment of individuals or groups (e.g., stereotyping, quality of service issues) or other risks or harms (e.g., legal risks, financial harms)? If so, please provide a description. Is there anything a dataset consumer could do to mitigate these risks or harms?

No.

## IV. Collection Process

*IV-A   How was the data associated with each instance acquired?*
e.g. the data collected survey, the raw data is routinely collected by the courts.

The data was collected via survey.

*IV-B   Was the information self-reported?*
If the data was self-reported, was the data validated/verified? If so, please describe how.

The information was reported directly from the agencies. However, it is organizations, not individuals, that were surveyed.

*IV-C   Who was involved in the data collection process?*
Was this done as part of their other duties? If not, were they compensated?

Police agencies, no compensation was given.

*IV-D   Over what timeframe was the data collected? Does this timeframe match the creation timeframe of the data associated with the instances (e.g., recent crawl of old news articles)?*
If not, please describe the timeframe in which the data associated with the instances was created. If the collection was not continuous within the timeframe, please specify the intervals, for example, annually, every 4 years, irregularly.

The data is collected on an irregular basis. For example, the last collection was 2016, and before that was 2013. When

collection takes place, it is done over the specified year. The earliest available data if from 1987.

**IV-E** *Were any ethical review processes conducted (e.g., by an institutional review board)?*
If so, please provide a description of these review processes, including the outcomes, as well as a link or other access point to any supporting documentation.

Unknown.

**IV-F** *Were the individuals in question notified about the data collection? Did they give their consent?*
If consent was obtained, were the consenting individuals provided with a mechanism to revoke their consent in the future or for certain uses?

N/A. The data is not on an individual level.

**IV-G** *If consent was obtained, were the consenting individuals provided with a mechanism to revoke their consent in the future or for certain uses?*
If so, please provide a description, as well as a link or other access point to the mechanism (if appropriate).

N/A

**IV-H** *Has an analysis of the potential impact of the dataset and its use on data subjects (e.g., a data protection impact analysis) been conducted?*
If so, please provide a description of this analysis, including the outcomes, as well as a link or other access point to any supporting documentation.

N/A

## V. PRE-PROCESSING, CLEANING, LABELING

**V-A** *Was any preprocessing/cleaning/labeling of the data done (e.g., discretization or bucketing, removal of instances, processing of missing values)?*
If so, please provide a description and reference to the documentation. If not, you may skip the remaining questions in this section.

The codebook does not specify pre-processing [11].

**V-B** *Was the "raw" data saved in addition to the preprocessed/cleaned/labeled data (e.g., to support unanticipated future uses)?*
If so, please provide a link or other access point to the "raw" data.

Skip.

**V-C** *Is the software that was used to preprocess/clean/label the data available?*
If so, please provide a link or other access point.

Skip.

## VI. DISTRIBUTION

**VI-A** *Is the data publicly available? How and where can it be accessed (e.g., website, GitHub)?*
Does the dataset have a digital object identifier (DOI)?

The dataset is distributed freely at: https://www.icpsr.umich.edu/web/NACJD/series/92

**VI-B** *Is the dataset be distributed under a copyright or other intellectual property (IP) license, and/or under applicable terms of use (ToU)?*
If so, please describe this license and/or ToU, and provide a link or other access point to, or otherwise reproduce, any relevant licensing terms or ToU, as well as any fees associated with these restrictions.

The data-use statement says the following:
**Citation Requirement:**
Publications based on ICPSR data collections should acknowledge those sources by means of bibliographic citations. To ensure that such source attributions are captured for social science bibliographic utilities, citations must appear in footnotes or in the reference section of publications.
**Deposit Requirement:**
To provide funding agencies with essential information about use of archival resources and to facilitate the exchange of information about ICPSR participants' research activities, users of ICPSR data are requested to send to ICPSR bibliographic citations for each completed manuscript or thesis abstract. Visit the ICPSR Web site for more information on submitting.

## VII. MAINTENANCE

**VII-A** *Who will be supporting/hosting/maintaining the dataset?*

The Bureau of Justice Statistics.

**VII-B** *How can the owner/curator/manager of the dataset be contacted (e.g., email address)?*

The BJS can be contacted at: askbjs@usdoj.gov

**VII-C** *Will the dataset be updated (e.g., to correct labeling errors, add new instances, delete instances)?*

Unknown.

**VII-D** *If the dataset relates to people, are there applicable limits on the retention of the data associated with the instances (e.g., were the individuals in question told that their data would be retained for a fixed period of time and then deleted)?*
If so, please describe these limits and explain how they will be enforced.

The dataset does not relate to people.

**VII-E** *Are older versions of the dataset continue to be supported/hosted/maintained?*

Yes. Data from previous years continue to be hosted and are available for download.

**VII-F** *If others want to extend/augment/build on/contribute to the dataset, is there a mechanism for them to do so?*
If so, please provide a description.

No.

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

# Personnel, Use of Force, and Complaints in the Chicago Police Department (CPD)
## Datasheet

**Reproduced from: [1]**

## I. MOTIVATION

### I *For what purpose was the dataset created?*

The original raw data files were sought by J. Kalven, a journalist in the City of Chicago, as part of his investigation into police abuse. After the original FOIA requests and legal case, the non-profit Invisible Institute (https://invisible.institute) began to collaborate with Kalven and the University of Chicago's Mandel Legal Aid Clinic to follow up on earlier FOIA requests and to file new ones. The data disclosed in response to these earlier and now ongoing FOIA requests were made available online as part of the Citizens Police Data Project.

### I *Who created the dataset, and on behalf of which entity?*

The Chicago Police Department (CPD), Civilian Office of Police Accountability (COPA), and the City of Chicago produced the raw data files in response to FOIA requests. The raw data were curated and released publicly by the Invisible Institute and its collaborators. The cleaned and linked data were produced as part of research by the authors of this document.

### I *Who funded the creation of the dataset?*

The acquisition of the original raw data was funded by the Invisible Institute.

## II. COMPOSITION

### II *What do the instances that comprise the dataset represent?*

There are multiple types of instance in this data.

- Officer: information about an individual police officer
- Unit assignment: a single unit assignment for an officer
- Complaint: a complaint filed against a police officer, either internally or by a civilian
- Tactical Response Report: a form that an officer is required to fill out after their response requires use of force
- Award request: a request to grant an award to an officer
- Salary: a record of an officer's salary, pay grade, and position across multiple years

### II *How many instances are there in total (of each type)?*

There are roughly 35,000 unique officers in the cleaned roster appearing in roughly 130,000 profiles throughout the data, 730,000 award request records, 194,000 salary records, 108,000 unit assignment records, 109,000 complaints, and 10,500 tactical response reports.

### II *Does the dataset contain all possible instances or is it a sample of instances from a larger set?*

This data contains information regarding all sworn officers in the Chicago Police Department / City of Chicago databases for the stated date ranges (which differ for each source of raw data).

### II *What data does each instance consist of?*

- Officer: officer unique ID, race, gender, age, appointment date, resignation date, badge number(s), position title(s)
- Unit assignment: officer unique ID, start date, end date, unit number
- Complaint: complaint ID, involved officer IDs, allegation, result of the investigation, resulting sanction (where available)
- Tactical Response Report: report ID, event location, date, and time, environmental conditions, who was notified, weapons discharged, weapon information, subject demographic information
- Award Request: awardee unique ID, requester, request date, award reference number, award type, request tracking number, incident dates, ceremony date
- Salary: officer unique ID, salary, position title, pay grade, year

### II *Is there a label or target associated with each instance?*

Not explicitly. However, labels could be constructed from the data that exists. For example, one could aggregate complaints to produce an integer "number of complaints" for each officer in the data, and use that as the response variable in a prediction task.

### II *Is any information missing from individual instances?*

In the original raw data files, missing data (of all fields) is quite common (see Appendix D in [1]). In the cleaned and linked data files, we are able to aggregate multiple profiles

of a single officer appearing throughout the data to "fill in the gaps," although this process is not perfect and there are still missing entries.

## II  Are relationships between individual instances made explicit?

In the raw data, no. In the cleaned data, we provide a unique officer identification that enables linking the activities and records regarding individual officers across datasets. There is no relational data (i.e., network edges) explicitly contained in the data. However, it is possible to use the data to construct a network, e.g., by linking officers co-listed on complaints.

## II  Are there recommended data splits?

No, although the officer database is likely to be incomplete prior to roughly 1980.

## II  Are there any errors, sources of noise, or redundancies in the dataset?

There are redundancies in the raw data, but these are removed by our cleaning and linking procedure. Errors, inconsistencies, and missing data are also present in the raw data; our cleaning and linking resolves much of these issues. However, per Section 4 in [1], the officer database is likely to be incomplete prior to roughly 1980 (as officers were added to the database only gradually over time).

## II  Is the dataset self-contained, or does it rely on external resources?

The dataset is self-contained: the raw data itself is stored in the `raw/` folder of the repository (with links to the external source files for reference), and the cleaned/linked data is produced by the source code in the repository.

## II  Does the dataset contain data that might be considered confidential?

No; all of this data was publicly released as part of FOIA requests. Confidential data (e.g., relating to under cover officers) was withheld by the Chicago Police Department.

## II  Does the dataset contain data that, if viewed directly, might be offensive, insulting, or threatening?

No.

## II  Does the dataset relate to people?

Yes; it contains records relating to police officers in the Chicago Police Department.

## II  Does the dataset identify any subpopulations?

Yes; officer records include race, gender, age, appointment date, unit history, badge numbers, position title, salary, awards, complaints, and tactical response reports. Subpopulations of officers can be constructed using these fields.

## II  Is it possible to identify individuals?

Yes; detailed information is available that could be used to identify individual officers.

## II  Does the dataset contain data that might be considered sensitive in any way?

The data contains a coarse categorization of racial origins of officers.

## III. COLLECTION PROCESS

## III  How was the data associated with each instance acquired?

The raw data were obtained via FOIA requests to the City of Chicago and Chicago Police Department.

## III  What mechanisms or procedures were used to collect the data?

The raw data were obtained via FOIA requests to the City of Chicago and Chicago Police Department.

## III  If the data are a sample from a larger set, what was the sampling strategy?

Not applicable.

## III  Who was involved in the data collection process and how were they compensated?

Journalists in collaboration with the Invisible Institute were responsible for filing the FOIA requests, and officials within the Chicago Police Department and City of Chicago were responsible for providing data in response to those requests. It is not known explicitly whether or how either party was compensated.

## III  Over what timeframe was the data collected?

The earliest releases per FOIA request occurred in 2016, and continue to occur as more FOIA requests are filed. The raw data itself pertain to records from the CPD dating back to the mid 20th century. The roster data covers the period up to 2018. The awards data pertains to records from 1967 to 2019. The salary data pertains to the years 2002 to 2017. The unit history data covers records up to 2016. The complaints data pertains to records from 1967 to 2016. The tactical response report data pertains to records from 2004 to 2017.

## III  Were any ethical review processes conducted?

It is unknown whether the CPD conducted any ethical review processes prior to the release of the raw data. No ethical review process was conducted prior to the activities involved in the present repository, i.e., cleaning the publicly available data.

## III  Does the dataset relate to people?

Yes; it contains detailed records regarding the activities of police officers in the City of Chicago.

## III  Did you collect the data from the individuals directly, or obtain it via third parties?

The raw data was acquired from public links provided by the Invisible Institute (https://invisible.institute). The Invisible Institute acquired the data through FOIA requests made to the CPD and the City of Chicago.

### III *Were the individuals notified about the data collection?*

It is unknown whether the individual officers were notified by the CPD when the raw data was released.

### III *Did the individuals in question consent to the collection and use of their data?*

Not explicitly. The Chicago Police Department was compelled by law to produce these records per FOIA requests.

### III *If consent was obtained, were the consenting individuals provided with a mechanism to revoke their consent in the future or for certain uses?*

Not applicable.

### III *Has analysis of the potential impact of the dataset and its use on data subjects been conducted?*

Not known.

## IV. PREPROCESSING AND CLEANING

### IV *Was any preprocessing of the data done?*

Yes; the main section of this documentation provides details the cleaning and linking of the raw data resulting from FOIA requests made to the City of Chicago.

### IV *Was the "raw" data saved in addition to the cleaned data?*

Yes; the raw data is available in the `raw/` folder in the repository.

### IV *Is the software used to clean the data available?*

Yes; the source for cleaning and linking is provided in the `src/` folder in the repository.

## V. USES

### V *Has the dataset been used for any tasks already?*

Not the newly cleaned and linked version. The raw data itself has been used previously; see e Section 5 in [1] for details.

### V *Is there a repository that links to any or all papers that use the dataset?*

Not that the authors of this work are aware of.

### V *What (other) tasks could the dataset be used for?*

This data set has a rich variety of possible uses; for example, network analysis (and in particular, analysis of dynamic events occurring on networks) and predictive regression/classification. See Section 5 in [1] for more details.

### V *Is there anything about the composition of the dataset or the way it was collected and cleaned that might impact future uses?*

Yes; the data are less reliable in earlier years (e.g., pre-1980). See Section 4 in [1] for more details.

### V *Are there tasks for which the dataset should not be used?*

This data should not be used to single out, study, or identify individual officers.

## VI. DISTRIBUTION

### VI *Will the dataset be distributed to third parties outside of the entity on behalf of which the dataset was created?*

Yes, the data is publicly available.

### VI *How will the dataset be distributed?*

It is available on GitHub at `https://github.com/chicago-police-violence/data`. Release versions will be marked using the "release" feature on GitHub.

### VI *When will the dataset be distributed?*

It is currently publicly accessible.

### VI *Will the dataset be distributed under a copyright, other IP license, or terms of use?*

Yes; the source code is released under the MIT license, and the data output by the cleaning code is released under the Creative Commons 4.0 BY-NC-SA license.

### VI *Have any third parties imposed IP-based or other restrictions on the data associated with the instances?*

No.

### VI *Do any export controls or other regulatory restrictions apply to the data?*

No.

## VII. MAINTENANCE

### VII *Who is supporting/hosting/maintaining the dataset?*

The repository will be hosted on GitHub. As of August 2021, the repository owners are Thibaut Horel, Trevor Campbell, and Lorenzo Masoero, but ownership may change over time.

### VII *How can the data owner/curator be contacted?*

Issue threads on GitHub are the primary channel of contact for the repository maintainers.

### VII *Is there an erratum?*

Not as of yet. For each major release version, notes will be included and hosted in the repository that will detail cleaning/linking errors that have been fixed.

### VII *Will the dataset be updated?*

The original raw source data from FOIA requests will not be modified. More raw data files may be added over time corresponding to new FOIA requests. The data cleaning and linking code will be edited over time to fix errors; release versions will be clearly marked on GitHub. There is no set schedule for updates.

### VII *If the dataset relates to people, are there applicable limits on the retention of data associated with the instances?*

No; this data was released per FOIA requests and is in the public domain.

**VII   Will older versions of the dataset continue to be supported/hosted/maintained?**

Yes; a full version-controlled history of the project exists on GitHub.

**VII   If others want to extend/augment/build on/contribute to the dataset, is there a mechanism for them to do so?**

Yes; the repository for the dataset is hosted on GitHub, where pull requests are a usual channel for external contribution.

## REFERENCES

[1]  Thibaut Horel, Lorenzo Masoero, Raj Agrawal, Daria Roithmayr, and Trevor Campbell. The CPD Data Set: Personnel, Use of Force, and Complaints in the Chicago Police Department, 2021.

# Profiles of Individual Radicalization in the United States (PIRUS)
## Datasheet

## I. Motivation

### I-A  For what purpose was the dataset created?

The PIRUS dataset was created to better understand domestic radicalization. The dataset contains information on individuals in the United States that have been radicalized between 1948 and 2018.

### I-B  Who created the dataset?
Is it an official law enforcement or government body? An academic research team? Other?

The dataset was created by START, the National Consortium for the Study of Terrorism and Responses to Terrorism, a university-based research center, based at the University of Maryland.

### I-C  Was there a specific task in mind, or gap that needed to be filled?

The PIRUS dataset is among the first efforts to understand domestic radicalization from an empirical and scientifically rigorous perspective [1].

## II. Composition

### II-A  What do the instances that comprise the dataset represent?
For example: crimes, offenders, court cases, police officers

Each instance corresponds to a de-identified individual who has been radicalized to violent or non-violent extremism.

### II-B  Are there multiple types of instances?
For example: offenders, victims, and the relationship between them.

No.

### II-C  How many instances are there in total?
Of each type, if appropriate.

There is data on 2,226 individuals in this dataset.

### II-D  Does the dataset contain all possible instances or is it a sample (not necessarily random) of instances from a larger set?
For example, if it is traffic stops from a territory, is it all traffic stops conducted within that territory within a specific time? If not, is it a representative sample of all stops? Describe how representativeness was validated/verified. If it is not representative, please describe why.

This is a sample of radicalized individuals in the United States. In order to be eligible for inclusion, each individual must meet one of the following five criteria:

1) The individual was arrested.
2) The individual was indicted of a crime.
3) The individual was killed as a result of his or her ideological activities.
4) The individual is/was a member of a designated terrorist organization.
5) The individual was associated with an extremist organization whose leader(s) or founder(s) has/have been indicted of an ideologically motivated violent offense.

In addition, each individual MUST:

1) Have been radicalized in the United States.
2) Have espoused or currently espouse ideological motives.
3) Show evidence that his or her behaviors are/were linked to the ideological motives he or she espoused/espouses.

However, the authors state:

"The PIRUS database is not, and should not be treated as, a comprehensive set of all individuals who have radicalized in the United States. Achieving a comprehensive dataset of all individuals who meet the database's inclusion criteria remains implausible for several reasons".[1]

### II-E  What data does each instance consist of?
If there is a large number of variables, please provide a broad description of what is included.

Each instances contains information on a wide range of characteristics, including:

1) Criminal activity.
2) Violent plots.
3) Relationship with extremist group.
4) Adherence to ideological milieus.
5) Factors relevant to their radicalization process.
6) Demographics.
7) Background.
8) Personal history.

### II-F  Is there a target label or associated with each instance?
Please include labels that are likely to be used as target labels, e.g. recidivism.

No. However, whether an individual's plot was executed according to their plan might be suitable to use as a target label.

---

[1]This quote is taken from Frequently Asked Questions on the PRIUS website.

**II-G**  *Are there recommended data splits (e.g., training, development/validation, testing)?*
*If so, please provide a description of these splits, explaining the rationale behind them.*

No.

**II-H**  *Does the dataset contain data on race and ethnicity?*
*If so, is it based on the individual's self-description, or based on officer's impression? Was it collected or derived in post-processing? For example, by name analysis.*

Yes. The PIRUS dataset, including information on race and ethnicity, was coded entirely using open-source material, including newspaper articles, websites, etc.

**II-I**  *Are there any known errors, sources of noise, bias or missing data, or variables collected for only part of the datasets?*
*If so, please provide a description.*

No. However, that information in the dataset is based on oopen-source material, including newspaper articles, websites, etc.

**II-J**  *Does the dataset contain data on criminal history or other data that might be considered confidential or sensitive in any way?*
*For example: sexual orientations, religious beliefs, political opinions or union memberships, or locations; financial or health data; biometric or genetic data; forms of government identification, such as social security numbers; If so, please provide a description.*

Yes. The dataset contains information on criminal activity and relationship with extremist group, as well as other personal information.

**II-K**  *Is it possible to identify individuals (i.e., one or more natural persons), either directly or indirectly (i.e., in combination with other data) from the dataset?*
*If so, please describe how.*

Indirectly, given the low frequency of the events and specific circumstances surrounding them.

## III. Uses

**III-A**  *What type of tasks, if any, has the dataset been used for?*
*If so, please provide examples and include citations.*

The dataset has been used for:

1) Comparitive studies between extremists and other groups [2], [3].
2) Looking at extremism within specific subgroups [4], [5].

**III-B**  *Is there a repository that links to any or all papers or systems that use the dataset?*
*If so, please provide a link or other access point.*

No.

**III-C**  *What (other) tasks could the dataset be used for?*
*For example: testing predictive policing systems, predicting recidivism.*

The dataset could be used for:

1) Additional investigations of extremism within sub-groups.
2) Comparison of extremism outcomes across political affiliation.
3) Stratification by date, age, gender, location, ideology, group, etc. to address the specifics of radicalization in the United States.

**III-D**  *Is there anything about the composition of the dataset or the way it was collected and prepro-cessed/cleaned/labeled that might impact future uses?*
*For example, is there anything that a dataset consumer might need to know to avoid uses that could result in unfair treatment of individuals or groups (e.g., stereotyping, quality of service issues) or other risks or harms (e.g., legal risks, financial harms)? If so, please provide a description. Is there anything a dataset consumer could do to mitigate these risks or harms?*

The dataset was compiled from many open-source sources, such as social media and news articles. When using the dataset, one must assume the data has been merged correctly, and the information taken from these sources is correct.

## IV. Collection Process

**IV-A**  *How was the data associated with each instance acquired?*
*e.g. the data collected survey, the raw data is routinely collected by the courts.*

The PIRUS dataset was compiled from: newspaper articles, websites, secondary datasets, peer-reviewed academic articles, journalistic accounts including books and documentaries, court records, police reports, witness transcribed interviews, psychological evaluations/reports, and information directly attributed to the individual being researched (social media, etc.).

**IV-B**  *Was the information self-reported?*
*If the data was self-reported, was the data validated/verified? If so, please describe how.*

No, the information is collected by the datasets' investigators from open source materials. Some information may in directly be self-reported (e.g., social media).

**IV-C**  *Who was involved in the data collection process?*
*Was this done as part of their other duties? If not, were they compensated?*

The data collection was performed by investigators from START: Gary LaFree, Michael Jensen, and Sheehan Kane, among others.[2].

---

[2]A full list of authors can be found: here.

*IV-D* *Over what timeframe was the data collected? Does this timeframe match the creation timeframe of the data associated with the instances (e.g., recent crawl of old news articles)?*
If not, please describe the timeframe in which the data associated with the instances was created. If the collection was not continuous within the timeframe, please specify the intervals, for example, annually, every 4 years, irregularly.

The dataset was collected between 2016 – 2018, and concerns the years 1948 through 2018.

*IV-E* *Were any ethical review processes conducted (e.g., by an institutional review board)?*
If so, please provide a description of these review processes, including the outcomes, as well as a link or other access point to any supporting documentation.

Unknown.

*IV-F* *Were the individuals in question notified about the data collection? Did they give their consent?*
If consent was obtained, were the consenting individuals provided with a mechanism to revoke their consent in the future or for certain uses?

No.

*IV-G* *Has an analysis of the potential impact of the dataset and its use on data subjects (e.g., a data protection impact analysis) been conducted?*
If so, please provide a description of this analysis, including the outcomes, as well as a link or other access point to any supporting documentation.

Unknown.

## V. PRE-PROCESSING, CLEANING, LABELING

*V-A* *Was any preprocessing/cleaning/labeling of the data done (e.g., discretization or bucketing, removal of instances, processing of missing values)?*
If so, please provide a description and reference to the documentation. If not, you may skip the remaining questions in this section.

The specific processing steps have not been provided by the creators.

*V-B* *Was the "raw" data saved in addition to the preprocessed/cleaned/labeled data?*
If so, please provide a link or other access point to the "raw" data.

Unknown.

*V-C* *Is the software that was used to preprocess/clean/label the data available?*
If so, please provide a link or other access point.

No.

## VI. DISTRIBUTION

*VI-A* *Is the data publicly available? How and where can it be accessed (e.g., website, GitHub)?*
Does the dataset have a digital object identifier (DOI)?

A publicly avilable version on the dataset can be downloaded from:
https://www.icpsr.umich.edu/web/NACJD/studies/36309
You can request access to the full dataset here:
https://www.start.umd.edu/webform/pirus-download-full-dataset.

*VI-B* *Is the dataset be distributed under a copyright or other intellectual property (IP) license, and/or under applicable terms of use (ToU)?*
If so, please describe this license and/or ToU, and provide a link or other access point to, or otherwise reproduce, any relevant licensing terms or ToU, as well as any fees associated with these restrictions.

The license agreement for the full dataset states the dataset can only be used for personal or academic research, journalistic use, or for an internal business process. See the license agreement for more details:
https://www.start.umd.edu/webform/pirus-download-full-dataset.

## VII. MAINTENANCE

*VII-A* *Is the dataset maintained? Who is supporting/hosting/maintaining the dataset?*

The dataset is not maintained since 2019.

*VII-B* *How can the owner/curator/manager of the dataset be contacted (e.g., email address)?*

Using the email pirus@start.umd.edu.

*VII-C* *Will the dataset be updated (e.g., to correct labeling errors, add new instances, delete instances)?*

Not unless the authors receive further funding.

*VII-D* *Are older versions of the dataset continue to be supported/hosted/maintained?*

No.

*VII-E* *If others want to extend/augment/build on/contribute to the dataset, is there a mechanism for them to do so?*
If so, please provide a description.

Unknown, contact pirus@start.umd.edu.