# OpenReview forum: "A Survey and Datasheet Repository of Publicly Available US Criminal Justice Datasets"
_NeurIPS.cc/2022/Track/Datasets_and_Benchmarks — NeurIPS 2022 Datasets and Benchmarks _

### Official Review · Reviewer_FZfq · 2022-07-21
**Incomplete picture of criminal justice**

**Rating:** 6
**Confidence:** 4
**Correctness:** They appear to be.
**Clarity:** It is clear

**Strengths:**

This is a good summary of many diverse data sets, collecting information that can be used to make informed choices about how to use this data.

**Weaknesses:**

 However I have some concerns.

My biggest concern is the presumption in their model that a crime was committed. The criminal justice system is imperfect, and not everyone is convicted, nor is everyone who is convicted guilty. Do no data sets address this issue? Even if that is the case, given that the authors wish to identify gaps in the data sets, why is that not represented in Figure 1 outside of the word "acquittal"?

In addition, the model is incomplete -- for example the list of outcomes of pretrial hearings is incomplete, as people may be released but required to wear an ankle monitor (just as an example). I am not an expert in this area, so there may be other outcomes that are missing too that I am unaware of.

Further, the order of operations shown is not accurate to actual criminal justice experiences. For example, many people have to pay fines even if they are not convicted, for example fees for required ankle bracelet surveillance between charging and conviction, or paying back a loan for bail money. My neighbor was forced to pay a fine simply for appearing in court, even though she was not charged in the end.

Another thing that seems to be missing from the model provided is the type of defender (public or private for example) that participants had access to, or even whether, when and how such a person was assigned.

**Additional Feedback:**

A good start. Not sure if this is ready for publication.

**Documentation:**

The documentation provided on their github repository is in the form of PDFs. This seems less than ideal to me. Even putting aside whether those PDFS are accessible, there is the question of difficulty in getting a gestalt picture of everything. Why not put this all in a spreadsheet where it is easy to compare different data sets for example?

Adding after rebuttal: I appreciate the additions to the paper and the HTML versions of the data sheets. My biggest remaining concern is that the website does not link to the paper, have any information about ethics, or highlight what is missing from the data sets (again, such as wrongful arrests and convictions). I would like to see these things added to the website, which may be all people see of this data repository.

**Ethics:**

The treatment of ethics in this paper falls short of what I'd like to see. The authors acknowledge the potential for concern (" We note these datasets could be used for political purposes, or as a means of creating predictive policing tools. We caution against unethical or irresponsible use of the datasets."). However, they do not educate the reader at all about the problems that have been found with how data is used in the past. This is a shame, since a quick glance at their analysis of the data set shows that they did report on uses (and how  a data set should not be used) that they could use to summarize problematic uses or concerns in the paper.

**Relation To Prior Work:**

Not my expertise.

**Summary And Contributions:**

The authors present an analysis of criminal justice related data sets. I think this is an important topic, and I appreciate the degree to which they are exploring pros and cons of these data sets.

Overall, this seems to be a valuable resource. I have some concerns about the completeness of the model however.

---

> ### Author Response · Authors · 2022-08-25
> **Response**
>
> Thanks for the time you’ve taken to read and review our paper! Please see below our comments on the highlighted weaknesses and additional feedback:
>
> Weaknesses:
>
> Funnel: Thank you for this feedback. We have now extended the funnel to include the option of a wrongful arrest and acquittal. Generally, we aimed to create a high-level representation rather than a fine-grained model. We thus believe that inclusion of additional outcomes would make the figure dense and less digestible, making it harder for the reader to appreciate how the individual datasets fit into the criminal justice process. A more detailed funnel would be certainly useful in a study focusing on the workings of the criminal justice system itself, as has been done in [1] for the UK. We have now added a comment stating that the presented funnel is not comprehensive.
>
> Documentation: We have created a simple website for this project (https://criminaljustice-datasheets.github.io/), where it is possible to download the latex files in addition to the pdfs.
>
> Ethics: We accept this suggestion and have used the additional space allowed to significantly expand the ethical discussion, including a longer discussion on past and potential misuses of these and similar datasets.
>
> [1] Miri Zilka, Holli Sargeant, and Adrian Weller. 2022. Transparency, Governance and Regulation of Algorithmic Tools Deployed in the Criminal Justice System: a UK Case Study. In Proceedings of the 2022 AAAI/ACM Conference on AI, Ethics, and Society (AIES '22). Association for Computing Machinery, New York, NY, USA, 880–889. https://doi.org/10.1145/3514094.3534200

---

### Official Review · Reviewer_nQjG · 2022-07-24
**A comprehensive survey of datasources on the US criminal justice system.**

**Rating:** 6
**Confidence:** 3
**Correctness:** I could not find statements that seem…

**Strengths:**

The paper makes an important contribution to the field of criminal justice datasets by collecting and systematizing a variety of available datasets. The survey seems to be exhaustive (within its limited domain of US datasets) and provides a good overview of relevant data sources.

The authors provide datasheets for datasets on 15 datasets which provide a comprehensive and structured resource that answers many of the relevant questions w.r.t. those datasets.

The authors discuss the "funnel" arising from the sequential decisions made in the court system and provide an overview of procedures that typically lead to this funnel. This provides a good understanding of the structure of the different data sources.

The paper discusses a series of relevant limitations to the collected data, raising awareness for potential use cases.


**Weaknesses:**

I think the paper would make an even stronger case if the results were presented in the form of a website/GitHub repository enabling easier navigation. This would also allow other investigators to contribute e.g. via pull requests and allow for updating resources if e.g. errors are found in the datasets or when new datasets become available. I would strongly suggest further work in that direction. In their current version, the datasheets are an artefact for which maintenance and updates are unclear which I would consider a major drawback of the paper.

Improving in this direction could be done by
1) Including a statement regarding maintenance and updates
2) Uploading latex sources that enable updating the datasheets
3) Uploading a latex template for similar datasheets
4) Outlining how people can contribute new datasets to this collection, how datasheets can be updated and the criteria for such changes.
4) Providing a good overview / improving the README in the existing Github repository.



**Additional Feedback:**

I personally am not used to seeing citations in the abstract, but I guess this is a matter of personal/editorial choice.

In my opinion, the funnel could be extended/slightly improved. The current funnel takes the perspective of a "crime" that takes place and then explores how this crime leads to data arising at different stages. I think this slightly distracts from how an individual might perceive this funnel if e.g. an innocent individual is dragged into the funnel erroneously. This might also help with a more systematic assessment of sampling biases arising from this funnel e.g. at which steps individuals are omitted from the next path at which data is collected.

The paper could be further improved by providing a review of (mis-)uses of the different datasets, which could lead the reader to additional resources and provide more in-depth info for the interested reader. I am not sure if the references that are given with each datasheet already cover this.

I understand why the survey focuses on the US criminal justice system, but I simultaneously dislike this fact as it further channels research and resources into the comparatively well-researched area of the US system. Instead of e.g. encouraging research into legal systems in countries that have less resources.This could perhaps be discussed in more detail in the manuscript.

**Clarity:**

The paper is generally well-written and clearly structured. The datasheets present a structured/condensed source of information.

**Documentation:**

The datasets are adequately covered in the different datasets.

**Ethics:**

All datasets included in the paper are already available publicly. I would therefore assume that the collection of such datasets - with the appropriate limitations and potential miss-uses clearly stated - does not raise any ethical concerns.

**Relation To Prior Work:**

The prior work is adequately covered in the paper.

**Summary And Contributions:**

The paper presents a collection of data sources on the US criminal justice system with the goal to provide researchers with an overview of available data sources. Datasets stem from a variety of fields from crime reports to jail/prison sentencing. The authors furthermore provide datasheets for 15 selected datasets which provides a standardized/structured resource for accessing relevant data characteristics.

My major complaint would be the lack of a maintenance schedule/route towards updating datasheets and the collection in general.
This does not only prevent collaboration with other researchers but also results in the paper being an immutable artefact which, in my opinion,  is not adequate for such a collection of datasets.
If this complaint were addressed, I would recommend acceptance of the paper.

---

> ### Author Response · Authors · 2022-08-25
> **Response**
>
> Thanks for the time you’ve taken to read and review our paper! Please see below our comments on the highlighted weaknesses and additional feedback:
>
> Website: We accept this suggestion and have created a website for this project (https://criminaljustice-datasheets.github.io/) where it is possible to download the latex files in addition to the pdfs. We will also be accepting (and encouraging) requests to update the datasheets. We also comment on maintenance and updates in the revised version of the manuscript.
>
> Additional feedback:
>
> Funnel: Thank you for this feedback. We have now extended the funnel to include an option of a wrongful arrest.
>
> (Mis)uses: We have included references that critique the use of the covered datasets, although we do not critique works directly to avoid a “naming-and-shaming”.
>
> US focus: Thank you for highlighting this point. The US system has indeed been a subject of disproportionate attention. However, similar datasets covering different territories (in the UK, for example) are often not public. In addition, we wanted to keep the scope manageable as a starting point. In the future, we would be interested in expanding the coverage of the project to include additional territories. We have expanded the discussion on this point in the revised manuscript.

---

> > ### Comment · Reviewer_nQjG · 2022-08-31
> > **Agree**
> >
> > The authors have addressed most of my concerns in the updated manuscript.
> > I will update my score accordingly.

---

### Official Review · Reviewer_4pZ4 · 2022-07-26
**Review of A Survey and Datasheet Repository of Publicly Available US Criminal Justice Datasets**

**Rating:** 6
**Confidence:** 3

**Strengths:**

1. It is in the best interest of the research community (among others) to broaden which datasets are used when studying criminal justice rather than focusing evaluations on a few datasets (e.g., COMPAS). With this paper, the authors have initiated this process by shining a light on 15 potential new datasets which are already publicly available.
2. The 15 datasets are thoughtfully organized and presented. In particular, I found mapping the datasets onto the pipeline (Figure 1) to be a useful tool for getting quickly acquainted with the datasets, and it nicely complements Tables 1 and 2.
3. In creating the index tables and a new datasheet template tailored to criminal justice datasets, the authors have initiated the important discussion about what metadata should accompany criminal justice datasets and how these metadata questions might be standardized for describing new criminal justice datasets going forward. This is an important discussion for datasets of any field, but is particularly challenging and important for criminal justice datasets in which context is often not properly considered.
4. The paper is well-written and should be easy to understand for a lay machine learning audience.

**Weaknesses:**

**Major points**:
1. Since the authors are bringing 15 criminal justice datasets to the attention of the ML community, it seems important to discuss in the paper why introducing these criminal justice datasets is beneficial to the ML community and to society. To this point, I feel it is important to answer the following questions: How does providing the datasheets improve how these datasets can be used in ML beyond the official documentation? Does introducing these datasets help alleviate some of the existing problems with the use of criminal justice datasets in ML? Or will this just extend current problems to new datasets?
2. In the introduction, the authors state, “We give broad context to the datasets, draw out potential uses, and discuss gaps and limitations.” While the paper does address the first and third points, it does not, in my opinion, adequately address potential uses for these datasets. Questions III-A - D in the datasheets provide some information on potential uses, but for a general machine learning audience it seems important to broadly discuss in the main paper how these 15 datasets should be used by the ML research community. Should they primarily be used to investigate the criminal justice pipeline (as most of the uses in the datasheets seem to indicate)? Should these datasets be used as benchmark datasets for testing out new methods which are not necessarily tailored to criminal justice applications?
3. I found myself quite curious about the updates made to the datasheet template as described in Section 2. This seems like an important contribution of this paper, but it is not highlighted as such. What gaps in the original template did these updates fill? What unique challenges do criminal justice datasets pose to metadata documentation?
4. It remains a bit unclear to me how the authors envision researchers using Section 4, which is nearly three pages dedicated to short descriptions of the 15 datasets. Is the intention that these descriptions provide a quick introduction to a dataset and if interested, one should then go to the datasheet for more information? The combination of Figure 1 and Tables 1 and 2 seems quite useful for this purpose in and of itself. The benefits of including these short descriptions in the main paper vs in the supplementary material is not clear to me.
5. While Section 3 provides necessary context (in particular for Figure 1), I am not well-versed in this area and find it troubling that this section does not have any references.

**Minor points**:
1. In the introduction, one of the paper’s highlighted contributions is stated as reporting on 30 datasets. However, it’s not clear to me that this is really the case. Throughout Section 4, other datasets are mentioned in addition to the main 15, but these are not included in the index tables nor in Figure 1 and are not provided datasheets. While I do see the usefulness in directing readers to other potentially relevant datasets in Section 4, I found myself a bit confused (in particular moving from Sections 1 to 2) as to where the 30 vs 15 datasets were coming into play. Now after reading the entire paper, I don’t feel that this is one of its contributions. I’ve noted this as a minor point because there is actually fairly little mention of the 30 datasets and removing mentions of this would seem to affect very little of the paper.
2. Related to the previous point, it is still a little unclear to me how the authors arrived at the 15 datasets for which they created datasheets, since there seems to have been 30 datasets which met the mentioned inclusion criteria. Why these 15 datasets? Are these 15 datasets useful for an ML audience in particular?
3. Figure 1: just want to verify that the colors in this figure correspond to the stages as mentioned in Table 1? It may be useful to indicate this somewhere on the figure.
4. Section 7 typo: I believe “trough” should be “through”.
5. Table 2: inconsistent capitalization in the geographic resolution and maintained columns.


**Additional Feedback:**

I would be happy to reconsider my score if my major points are addressed in the rebuttal.


**Clarity:**

Yes, overall the paper is well written and straightforward to follow. Please see points above for suggestions regarding re-organization and content clarification in a few places.

**Correctness:**

To the best of my knowledge, the authors have documented the datasets in accordance with the dataset creators’ official documentation. They also note that the official documentation should be consulted before using a dataset.

**Documentation:**

Yes. Datasets appear to be well-documented by the datasheets created by the authors.


**Ethics:**

I do not believe that there are additional ethical concerns beyond those addressed by the authors in Section 6. All of the datasets included in the paper are already publicly available in NACJD.

**Relation To Prior Work:**

Some prior work is discussed in the introduction, and relevant work for each of the datasets is cited. In addition to this, I am also curious to know if other such surveys have been conducted in the past or if this is the first survey of its kind. I.e., have other surveys been conducted of criminal justice datasets? If so, how does this one differ? Have similar surveys been conducted for other disciplines and presented to the ML community in a similar manner (e.g., as in the two two index tables and as a datasheet repository)?

**Summary And Contributions:**

In this paper, the authors surveyed datasets from the National Archive of Criminal Justice Data (NACJD) and compiled a list of datasets arising from a variety of points in the criminal justice pipeline. For 15 of these datasets, the authors created publicly available datasheets using a template that they updated from the original in order to better suit criminal justice datasets. The authors briefly describe each of the 15 datasets in the paper and create two index tables that summarize for each dataset: 1) the type of criminal justice information and demographics covered, and 2) the size, composition, maintenance, and license information. Finally, the authors discuss challenges in working with criminal justice datasets and illustrate these points using examples from the 15 surveyed datasets.

---

> ### Author Response · Authors · 2022-08-24
> **Response (Major)**
>
> Thanks for the time you’ve taken to read and review our paper! Please see below our comments on the highlighted weaknesses and additional feedback:
>
> Major points:
> 1. The official documentation of the covered datasets ranges from a few to hundreds of pages. The NSDUH 2020 codebook, for example, is 845 pages long. The datasheets are intended as a standardized and information-efficient introduction to the datasets, but we fully anticipate users to consult the official documentation when conducting their research. Introducing these datasets helps narrow the large gap between the ML research community and applications of ML and algorithmic approaches within the criminal justice system. Although none of these datasets should be used as training or test data for real-world applications, they allow researchers to consider a more diverse and real-world representative set of predictive challenges within the criminal justice domain. We cannot guarantee that these datasets will not be used by anyone in a careless manner, but aim to inform and help researchers engage with this domain in a more responsible manner.
> 2. Thank you for raising this important point. We have included an expanded discussion on uses as part of a longer ethical discussion in the revised version of the manuscript. Generally, we do not propose use of the discussed datasets as benchmarks, although COMPAS is widely regarded as such. They can, however, be used to investigate new methods and approaches. Specifically, these datasets can be used when researchers wish to demonstrate how their method may be applicable to real-world decision-making scenarios but are 1) not in a position (or don’t want) to develop an application, or 2) do not have access or are not able to publish application-specific data. Most of the datasets can also be used to investigate the criminal justice system itself but we recommend engaging with domain experts to ensure proper context and interpretation of the results.
> 3. Thank you for highlighting this point. It was an interesting thought process and we perhaps didn’t fully appreciate the value of describing it in detail. The main challenge we found is that the datasheets seemed to be designed to be filled by the creators of the datasets. In addition, there is a focus on datasets created for the purpose of being used in ML, and most of the datasets we included were not designed for this process. The questions themselves are domain agnostic and we have found most of them suitable for criminal justice. However, datasets in criminal justice are prone to particular biases (e.g. data collection on ethnicity), which we have tried to include in the template. We anticipate continuing to revise and refine this template based on community feedback.
> 4. We appreciate this point. We did envision this section as an entry point for the specific datasets included in the survey, but also more generally for the type of data that is openly available about the criminal justice system. We have considered moving this section to the supplementary material, but we worry that for the less familiar reader, the points made in the discussion will not be clear without reading a short summary of the datasets before reaching the discussion.
> 5. Thank you for this feedback. We have added references to this section in the revised version of the manuscript.

---

> > ### Author Response · Authors · 2022-08-24
> > **Response (Minor)**
> >
> > Minor points:
> > 1. We appreciate this point. Indeed, we pay much more attention to 15 datasets that we felt are likely to be more valuable for the community. However, we do feel it is a contribution to point out the existence and provide a reference to another 15 datasets that may be of use to some of the readers. We did not add this to Figure 1 or the summary tables as we felt it will make those overly dense and thus less useful. We have added a version of Figure 1 that contains all 30 datasets to the appendix.
> > 2. We prioritised datasets that we think are likely to be the most useful for the ML community. Our choice is based on several concrete factors (size, population coverage, richness of variables, quality of the documentation, proportion of individual level over aggregate statistics) together with an aim to choose a diverse collection of datasets covering various parts of the pipeline (Figure 1). The above has been mentioned (lines 66–70), but we now expanded and clarified our discussion.
> > 3. The colours illustrate the separation between the different stages (police, courts, prison) and highlight the ‘escalation’ going from a police investigation (up) to a prison sentence. We have added a comment regarding this in the text.
> > 4. (+ 5) Thank you for spotting these mistakes!
> >
> > Relation To Prior Work: A recently accepted paper (http://www.dei.unipd.it/~fabrisal/papers/2022_eaamo.pdf) conducts a survey on algorithmic fairness datasets, including creating a datasheet for COMPAS. We cite this work in the revised version of the manuscript. We are not aware of any similar surveys covering the criminal justice domain, or other domain-specific surveys.

---

> > > ### Comment · Reviewer_4pZ4 · 2022-08-29
> > > **Response to authors**
> > >
> > > I thank the authors for their rebuttal comments and revised paper. In particular, I appreciate the expansion of Section 5 to include more discussion on the benefits and risks of increasing the visibility of criminal justice datasets in ML and the addition of citations and caveats for Figure 1.
> > >
> > > A few small notes:
> > > * Echoing Reviewer vac8’s response, the gray path in Figure 1 is a bit hard to read and differentiate from the black paths. I like their suggestion of a dotted line or a more contrasting color.
> > > * There is still inconsistent capitalization in Table 2 (see minor point 5 in my original review).
> > > * I appreciate the authors’ points on Section 4 and agree this seems like a useful entry point for the datasets, but I’m afraid I remain unconvinced that this section (in its current form) is most useful to readers in the main paper rather than in the supplement. I’m still left wondering if there is a more efficient way to introduce the datasets and prepare readers for Section 5 that does not take around a fourth of the paper’s page count. This would allow even further expansion and discussion in other sections.
> > >
> > > Overall, the authors’ revisions have improved the paper, particularly the expanded Section 5, and I feel most of my major concerns have been adequately addressed by the author response. In light of this, I raise my score.

---

> > > > ### Author Response · Authors · 2022-08-29
> > > > **Response**
> > > >
> > > > Thank you very much for taking the time to consider our response and the revised version of the paper! We are pleased you find it improved and thank you again for all your helpful feedback.
> > > >
> > > > On the small notes:
> > > >
> > > > -- We have now turned that pathway to a dotted line in addition to it being grey.
> > > >
> > > > -- Thank you! this is now resolved as well.
> > > >
> > > > -- We do appreciate this point. At the moment we are not sure how this can be resolved but will be happy to accept suggestions. We may be able to condense this section somewhat in favour of the discussion in the final version.
> > > >
> > > > Thank you again!

---

### Official Review · Reviewer_6Woc · 2022-07-27
**Important and well-done paper**

**Rating:** 8
**Confidence:** 4

**Strengths:**

This is a well-written paper on an important topic. Papers like this will become increasingly important in the realm of machine learning; datasheets for datasets and specialized repositories are essential for responsible data use. The survey is thorough and thoughtfully done.

**Weaknesses:**

I think the paper could benefit from an expanded discussion of related work and misuse of these datasets.

**Additional Feedback:**

I really enjoyed this paper; surveys like this will be an important step for contextual data use in machine learning. With a bit more polishing (including possibly graduating from GitHub) this repository has the potential to be very useful for machine learning researchers. This is a clear accept.

**Clarity:**

The paper is well-written and well-organized. The figure is clean and enhances the clarity of the text.

**Correctness:**

The survey seems complete, including a broad range of criminal justice datasets (including all the major datasets I'm familiar with).

**Documentation:**

The datasheets are well-done, i.e., the datasets are well-documented. The use of GitHub is not ideal in terms of guaranteed repository availability, but this is a minor concern. Polishing the repository itself for better navigability would be helpful future work.

**Ethics:**

The paper itself is likely to improve ethical considerations in ML use of criminal justice data. However, as mentioned above, the paper could benefit from a more detailed discussion of current ethical concerns in this realm.

**Relation To Prior Work:**

I think an expanded discussion of related work would improve this paper. I am not familiar with any other survey papers for criminal justice data in particular, but a paragraph regarding the need/use/creation of such data surveys and specialized data repositories would better place this work in context.
In addition, a bit more discussion on criminal justice data misuse in ML would fit in well with the rest of the paper and better motivate the survey.

**Summary And Contributions:**

The authors conduct a survey of criminal justice datasets. Their main contributions are a discussion of these datasets, in context of the full criminal justice pipeline, and a public repository containing datasheets for 15 selected datasets.

---

> ### Author Response · Authors · 2022-08-24
> **Response**
>
> Thanks for the time you’ve taken to read and review our paper! We are happy and grateful that you see the value this project can offer to the ML community. We thank you for your helpful suggestions, which we have incorporated: we have used the added 10th page to expand on the ethical discussion, including a longer discussion on potential misuses of the datasets. We have also expanded the introduction and added citations that discuss the importance of data-related work to the ML community.

---

### Official Review · Reviewer_HPfZ · 2022-07-27
**Review of A Survey and Datasheet Repository of Publicly Available US Criminal Justice Datasets**

**Rating:** 6
**Confidence:** 4
**Clarity:** The paper is clearly written.

**Strengths:**


### Utility and quality of the submission: Impact, originality, novelty, relevance to the NeurIPS community will all be considered.

+ This piece reviews an important area of criminal justice datasets. These datasets are used in a large body of fairness literature. The work provides a survey of relevant criminal justice datasets and supplements these datasets with data sheets. This is important and relevant to the NeurIPS community and provides a nice survey while adding additional value in the form of data sheets.
+ There is a nice synthesis of information from the authors' survey in the discussion section. This also adds value to the survey.

### Completeness of the relevant documentation: For datasets, sufficient detail must be provided on how the data was collected and organized, what kind of information it contains, how it should be used ethically and responsibly, as well as how it will be made available and maintained. For benchmarks, best practices on reproducibility should be followed.

+ The main contribution of this work *is* documentation, so it certainly has that.

### Accessibility and accountability: For datasets, there should be a convincing hosting, licensing, and maintenance plan.

+ There is a github repo for maintaining and updating datasheets.


**Weaknesses:**


### Utility and quality of the submission: Impact, originality, novelty, relevance to the NeurIPS community will all be considered.

+ This isn't the most novel or original work since it is completing datasheets for datasets that have examined by a number of works in the past (this piece cites many such related works). However, no other work is as comprehensive in its survey and no work systematically creates datasheets. So concerns of novelty are somewhat minor.
+ The discussion on biases could be strengthened. At the point where it is discussed, it feels like an afterthought. But there are significant problems here beyond just implicit biases. For example, earlier there is discussion on how, for example, "Hispanics are often reported as White." This would lead to significant problems with biases and errors in analyses in downstream models (especially for things like fairness by accounting for demographics). This discussion could be expanded to touch on more of the findings from the authors' exploration that might affect models or analyses.
+ Similarly much of the discussion makes it seem like omitting information is a strictly bad thing. For example, the authors seem to lament that victim information is not included to conduct analyses with. But these datasets require significant care and taking into account privacy considerations. More engagement with these privacy considerations would strengthen the paper.

### Completeness of the relevant documentation: For datasets, sufficient detail must be provided on how the data was collected and organized, what kind of information it contains, how it should be used ethically and responsibly, as well as how it will be made available and maintained. For benchmarks, best practices on reproducibility should be followed.

+ Some of the datasheets are not particularly detailed. If the contribution of the work is the datasheets I would have expected a bit more. For example, for the CPII datasheet the question, "What data does each instance consist of?" has the response: "As the data is compiled from 27 different sources, each source has a different set of variables. All sources report on the date, time, and location of the crime (as recorded) and the type of the offense." I would've expected a table or breakdown here to make this information up front. Similarly on this same page there are some broken references [?]. I would suggest a pass over the supplementary datasheets to add more detail to spots like this and fix broken refs.

### Accessibility and accountability: For datasets, there should be a convincing hosting, licensing, and maintenance plan.

+ If the goal is to raise awareness of these datasheets and the underlying gaps for these datasets, I would suggest creating a project webpage to host them in a way that's more prominent than as pdfs in a github repo. This will raise the impact, make it more accessible, and might get others to update the datasheets as new information is identified (such as for the questions regarding whether the data is being used already).


**Additional Feedback:**


# Overall

While I have some concerns on novelty, overall, I think the paper should be accepted, but only if authors go through the datasheets and elaborate on some of the spots that look a bit skimmed over (as well as fixing broken references). I also suggest that for the sake of impact that the authors create a website to host the datasheets for accessibility.



**Correctness:**

The claims are narrow and limitations are discussed. However, the work does not do enough to caveat the discussion of the criminal justice system. For example:

L91-92, it should be clear that policing efforts are not only encompassed by stop and search or sting operations, there are other mechanisms, for example like auditing in the case of monetary crimes. A simple fix is "The latter includes..."

There are many cases where this is an oversimplification of all the potential processes involved (and as such glazes over many areas which can lead to additional biases/problems with the datasets). However, as this is not a comprehensive survey of the CJ systems in the US or around the world, I suggest that the authors simply add appropriate caveats throughout the paper (and also discuss briefly how things might change for a global perspective).

**Documentation:**

See above, it seems like a website is a better format to host the datasheets than a github repo. This would help with accountability since it is more visible. Combined with an option to pull request updates to the website, this would ensure better maintainability since it distributes the load to the community.

**Ethics:**

No ethical concerns.

**Relation To Prior Work:**

The paper discusses prior work in this area and covers the relevant papers that I would have cited. Some more engagement with [2] could be warranted though since that work specifically discusses the challenges of gathering datasheets for criminal justice datasets.

**Summary And Contributions:**

This work surveys datasets in the criminal justice field that are often used for machine learning or fairness research. It supplements this survey with a catalog of datasheets that are newly generated by the authors (except for 1, where the dataset creators already generated a datasheet). Along with the survey there is a synthesis of knowledge where authors discuss gaps in the datasets and potential downstream analyses.

---

> ### Author Response · Authors · 2022-08-24
> **Response**
>
> Thanks for the time you’ve taken to read and review our paper! Please see below our comments on the highlighted weaknesses and additional feedback:
>
> Discussion on bias: We appreciate and accept this point. We have used the additional space allowed for the revised submission to expand the ethical discussion, including a more detailed discussion on biases generally, and their implication for modelling choices.
>
> Privacy and omitting information: We appreciate this feedback. Purely for the purpose of understanding the workings of criminal justice system (including biases and discrimination), having an as-complete-as-possible picture is beneficial. However—as you say—this can clash with privacy. We have included this point in the revised discussion.
>
> Pass over the datasets: Thank you for pointing out the broken references. We have now gone over all the datasheets again looking for mistakes that we missed before the original submission. We have added a table to the CPII datasheet as you suggested.
>
> Website: We accept this suggestion and have created a simple website for this project (https://criminaljustice-datasheets.github.io/), where it is possible to download the latex files in addition to the pdfs. We will also be accepting (and encouraging) requests to update the datasheets.
>
> Correctness: We accept these suggestions and have made revisions to the manuscript accordingly. A discussion of the US vs. global has also been added to the ethical discussion.
>
> Prior work: We have expanded the discussion of [2] in the discussion.

---

### Official Review · Reviewer_vac8 · 2022-07-27
**The paper proposes an interesting survey on existing US criminal justice datasets. However, it lacks references and their claims on the datasets are not well-enough supported. Finally, major ethical concerns are addressed only superficially by the authors.**

**Rating:** 6
**Confidence:** 3

**Strengths:**

1. The paper formalizes the US justice pipeline as a flowchart, and uses it to link the datasets to relevant parts of the pipeline, which makes it easy for researchers to choose a dataset.
2. A datasheet is associated to each dataset, providing a very practical and useful summary of each dataset.
3. A gap in coverage of the justice pipeline by existing datasets is identified, which could guide the collection of future datasets (although the authors note it is difficult).
4. Political and social implications of the availability of such datasets are quickly discussed.


**Weaknesses:**

1. Although it is central to this kind of data, the ethical discussions are a bit limited. Section 6 only contains one sentence about it, and the datasheets do not seem to contain any informations regarding these questions.
2. The authors provide download links for the datasets, but I would have liked to have a (unified) method that download the datasets automatically, making the use of multiple criminal justice dataset practical for researchers.
3. The criminal justice pipeline described in Figure 1 does not seem to be supported by any reference. I believe there are two possibilities, either it is a well-known pipeline (then I would appreciate to have a link to relevant previous works) or it is a contribution of the paper (in which case reference to e.g., the law, could be nice, but I admit I am no expert of the US criminal justice). It also looks impossible to escape the pipeline after being charged: I would imagine that after probation one can be free, but it does not appear on Figure 1?
4. The authors mention many datasets, but do not include data sheets for each of them. What motivates the choice between the datasets chosen for datasheets and the others?
5. Some datasets contain multiple data records about one given individual. Authors mention that partial criminal path can be reconstructed from these datasets. It is not clear how many of these partial paths can be reconstructed? For all records? 10% of records? 1%?
6. The authors mention that different data collection processes lead to different kinds of "unfairness" in the datasets. They claim that "bias may be present at each step of the pipeline", but fail to propose any (even very basic) statistics to support this claim.
7. I find Figure 1 to be a little to strict: are the authors sure that their association between one dataset and one part of the pipeline are correct? I am worried that someone could find other uses for these datasets, either by applying new methods on it or by combining multiple datasets (that are gathered in the survey).


**Additional Feedback:**

Minor questions: I wonder whether the data sheets contain too many questions or not. But I can not really evaluate whether this is relevant for a researcher who would use this data.

Also, while the red subtitles in the data sheets help to answer the questions to add a new dataset, they mostly contribute to making the link between questions and their answers more difficult to read.

**Clarity:**

The paper's English is good. However, the structure of the paper makes it a little difficult to read. Notably, I would find it more clear if sections of Figure 1 were given titles, and if Section 4 was organized using these titles.

**Correctness:**

I am not an expert in criminal justice, but it seems authors have correctly summarized dataset informations in the provided data sheets and in Tables 1/2. The pipeline presented in figure 1 looks reasonable but I can not fully evaluate its correctness.

**Documentation:**

Documentation of datasets is quite complete. For each of the proposed datasets, the authors detail size of the dataset, data collection, previous uses of the dataset, preprocessing, download link and license. A rather extensive comparison of these datasets on many parameters is provided in Tables 1 and 2.

However ethical concerns remain, see next section.


**Ethics:**

Authors do not really discuss ethics in details, although this is crucial for such types of data. Another of my concern is that combination of the proposed datasets may facilitate new unethical uses of the data, and I would have appreciated more insights from the authors on this point.

Another ethical concern is the possibility of re-indentification, which is more relevant for the dataset owners than for this survey, but that could maybe be discussed in a little more depth.

**Relation To Prior Work:**

Literature on the use of machine learning over criminal justice datasets seem to be well-covered. More references supporting the criminal justice pipeline in Figure 1 would however be appreciated.

**Summary And Contributions:**

This paper is a survey on 15 datasets related to US Criminal Justice. It describes the US Justice pipeline in a flowchart, and sorts the datasets with according to it. The authors propose a datasheet for each dataset, which summarizes relevant information on the dataset (data collection, motivation, uses, distribution and maintainance). They highlight that there are parts of the justice pipeline to which no dataset correspond, and discuss domain-specific difficulties in the data collection process.

---

> ### Author Response · Authors · 2022-08-24
> **Response**
>
> Thanks for the time you’ve taken to read and review our paper! Please see below our comments on the highlighted weaknesses and additional feedback:
>
> 1. We accept this suggestion and have used the additional space allowed to significantly expand the ethical discussion. We now (i) discuss in greater detail the benefits and risks of using the studied datasets within ML, (ii) expand the bias and privacy, and (iii) discuss potential misuses including the combination of datasets.
> 2. We appreciate this suggestion; however, the datasets are distributed under varying licences and other conditions of use. Providing the suggested download method would risk violating several of them. To make things easier as you suggest, we added a table with a download URL for each dataset to the Appendix.
> 3. Figure 1 is based on previously published work of the authors [1]. We now cite the paper and additional supporting sources in the relevant section of the paper.
> 4. We prioritised datasets that we think are likely to be the most useful for the ML community. Our choice is based on several concrete factors (size, population coverage, richness of variables, quality of the documentation, proportion of individual level over aggregate statistics) together with an aim to choose a diverse collection of datasets covering various parts of the pipeline (Figure 1). The above has been mentioned (lines 66–70), but we now expanded and clarified our discussion.
> 5. The only four kinds of individuals in these datasets are those with: 1) single arrest; 2) one arrest in the covered jurisdiction and more elsewhere; 3) multiple arrests in the covered jurisdiction but none elsewhere; and 4) multiple arrests in the covered jurisdiction and more elsewhere. We cannot distinguish between 1) and 2), and between 3) and 4). We can, however, quantify the proportion of the dataset that is covered by 3+4. We will add this information to the relevant datasheets.
> 6. Apologies, but we are not sure if we correctly understand the question. If the enquiry is about evidence of bias in the datasets, we are happy to add references indicating known biases within specific datasets [2,3]. In general, there are references [4,5] showing racial disparities in the likelihood of arrest and incarceration in the US. These disparities are likely to be reflected in the datasets we study. We can add a note of caution citing this specific example.
> 7. Figure 1 is aimed at giving a quick overview of the type of data that is covered by each dataset. While we of course cannot guarantee to anticipate the future, it is hard to imagine using the datasets for purposes other than in Figure 1 without re-identification. Since the datasets are anonymised, re-identification of all or even the majority of records is unlikely. For example, since NIBRS only records arrests but not information about convictions, its utility for a study focusing on court records is limited without large-scale re-identification. We will add a clarifying note to this effect.
>
> Clarity: Thank you for the suggestions! We incorporate all of them in the new revision.
>
> Ethics: We accept the suggestion to expand the ethical discussion and have used the additional 10th page to do so. We now discuss a range of issues ranging from un/intentional misuse to dataset combination. Regarding re-identification, for the majority of datasets introduced, re-identification is highly unlikely, except for high-profile crimes / court cases that have been discussed in detail in the media. There are exceptions to this, notably the COMPAS dataset, where actual names are included. However, this dataset is already widely known and used. It is included in our work mainly to highlight the dataset’s problems, of which re-identification is one.
>
> Additional feedback:
> Regarding too many questions: this is a nuanced point and something we have considered while filling in the datasheets themselves. We have chosen to stay on the side of trying to provide more information rather than less, as it is easier to skip questions than to discover missing information. Since we hope this will be an evolving project with community feedback, this will be revisited and refined in time.
> With regard to the subtitles, we appreciate this feedback. We have added a command to the latex datasheet files, “\subtitlesfalse”, that hides the subtitles.
>
> Refs in next comment.

---

> > ### Author Response · Authors · 2022-08-24
> > **Refs**
> >
> >
> > [1] Miri Zilka, Holli Sargeant, and Adrian Weller. 2022. Transparency, Governance and Regulation of Algorithmic Tools Deployed in the Criminal Justice System: a UK Case Study. In Proceedings of the 2022 AAAI/ACM Conference on AI, Ethics, and Society (AIES '22). Association for Computing Machinery, New York, NY, USA, 880–889. https://doi.org/10.1145/3514094.3534200
> >
> > [2] Riccardo Fogliato, Alice Xiang, Zachary Lipton, Daniel Nagin, and Alexandra Chouldechova. 2021. On the Validity of Arrest as a Proxy for Offense: Race and the Likelihood of Arrest for Violent Crimes. In Proceedings of the 2021 AAAI/ACM Conference on AI, Ethics, and Society (AIES '21). Association for Computing Machinery, New York, NY, USA, 100–111. https://doi.org/10.1145/3461702.3462538
> >
> > [3] Bao, Michelle, et al. "It's COMPASlicated: The Messy Relationship between RAI Datasets and Algorithmic Fairness Benchmarks." arXiv preprint arXiv:2106.05498 (2021).
> >
> > [4] Sutton, John R. "Structural bias in the sentencing of felony defendants." Social Science Research 42.5 (2013): 1207-1221.
> >
> > [5] Bradley Butcher, Chris Robinson, Miri Zilka, Riccardo Fogliato, Carolyn Ashurst, and Adrian Weller. 2022. Racial Disparities in the Enforcement of Marijuana Violations in the US. In Proceedings of the 2022 AAAI/ACM Conference on AI, Ethics, and Society (AIES '22). Association for Computing Machinery, New York, NY, USA, 130–143. https://doi.org/10.1145/3514094.3534184

---

> > > ### Comment · Reviewer_vac8 · 2022-08-29
> > > **Response to authors**
> > >
> > > Thank you for your answer, which addresses all my concerns. I particularly appreciate the additional references and more extensive discussions on dataset choices and ethical concerns. I agree that the collection of these datasets does not create new major concerns on reidentification since record linkage is not so likely (or due to external sources as the medias).
> > >
> > > I will increase my score in light of the authors response.
> > >
> > > Below are some very minor issues/typos that remain:
> > > - References [39] reads "criminal justice system flowchart", it does not look normal.
> > > - Line 346: "... as re-identification is usually possible" -> usually not possible?
> > > - The grey color for the grey path in Figure 1 is not easily distinguishable from the black one (maybe use dotted line or more contrasted color).
> > > - Line 350: "whie" -> "while".

---

> > > > ### Author Response · Authors · 2022-08-29
> > > > **Response**
> > > >
> > > > Thank you very much for taking the time to consider our response and the revised version of the paper! We are pleased you find it improved and thank you again for all your helpful feedback.
> > > >
> > > > Thank you also for spotting these typos! We have now uploaded a revision where these are resolved.

---

### Review · Ethics_Reviewer_kfw9 · 2022-08-25

**Recommendation:** 2

**Ethics Review:**

As multiple reviewers have noted, there is insufficient discussion of the ethical components of such a dataset. In the author response them indicate they have expanded the paper by adding a 10th page to include amore in depth discussion. However, I do not see this 10th page. I encourage the authors to upload their revised version so I can review and give final comments in light of their revisions.

---

> ### Author Response · Authors · 2022-08-27
> **Revised paper uploaded**
>
> Thank you for taking the time to read our paper. We have now uploaded the revised version. For the convenience of the reviewers, revisions are in red. We have attempted to address all the points made by the reviewers. We hope you find the ethical discussion satisfactory. Please let us know if you have any further suggestions, we would be happy to address them.

---

### Meta-Review · Area_Chair_cdpY · 2022-09-09

**Recommendation:** Accept
**Confidence:** 4

**Metareview:**

The paper introduces a set of criminal justice datasets to the machine learning community, surveying 30 datasets and creating datasheets for 15 of them. Reviewers appreciated that the paper raises awareness of these datasets in the ML community and the documentation work that the authors have contributed. There were two main concerns: inadequate discussion of ethics, and lack of detail on how the ML community could work with these datasets. The authors have addressed the first concern in a revision and partially addressed the second concern.

---

### Decision · Program_Chairs · 2022-09-16

Accept